# A co-design framework of neural networks and quantum circuits towards quantum advantage

Weiwen Jiang [1✉], Jinjun Xiong [2✉] & Yiyu Shi [1✉]

Despite the pursuit of quantum advantages in various applications, the power of quantum computers in executing neural network has mostly remained unknown, primarily due to a missing tool that effectively designs a neural network suitable for quantum circuit. Here, we present a neural network and quantum circuit co-design framework, namely QuantumFlow, to address the issue. In QuantumFlow, we represent data as unitary matrices to exploit quantum power by encoding $n = 2^k$ inputs into $k$ qubits and representing data as random variables to seamlessly connect layers without measurement. Coupled with a novel algorithm, the cost complexity of the unitary matrices-based neural computation can be reduced from $O(n)$ in classical computing to $O(polylog(n))$ in quantum computing. Results show that on MNIST dataset, QuantumFlow can achieve an accuracy of 94.09% with a cost reduction of $10.85 \times$ against the classical computer. All these results demonstrate the potential for QuantumFlow to achieve the quantum advantage.

[1] University of Notre Dame, Notre Dame, IN 46556, USA. [2] IBM Thomas J. Watson Research Center, Yorktown Heights, NY 10598, USA.
✉email: wjiang2@nd.edu; jinjun@us.ibm.com; yshi4@nd.edu

Although quantum computers are expected to dramatically outperform classical computers, so far quantum advantages have only been shown in a limited number of applications, from the prime factorization[1] and sampling the output of random quantum circuits[2] to the most recent breakthroughs on Boson Sampling[3]. In this work, we will demonstrate that quantum computers can achieve potential quantum advantage on neural network computation, a very common task in the prevalence of artificial intelligence (AI).

In the past decade, neural networks[4–6] have become the mainstream machine learning models, and have achieved consistent success in numerous AI applications, such as image classification[7–10], object detection[11–14], and natural language processing[15–17]. When the neural networks are applied to a specific field (e.g., AI in medical or AI in astronomy), the high-resolution input images bring new challenges. For example, one 3D-MRI image contains $224 \times 224 \times 10 \approx 5 \times 10^6$ pixels[18] while one Square Kilometer Array (SKA) science data contains $32,768 \times 32,768 \approx 1 \times 10^9$ pixels[19,20]. The large inputs greatly increase the computation in neural network[21], which gradually becomes the performance bottleneck, even consider the advance classical computing hardware[22,23]. Among all computing platforms, the quantum computer is the most promising one to address such challenges[2,24] as a quantum accelerator for neural networks[25–27]. Unlike classical computers with $N$ digit bits to represent 1 N-bit number at one time, quantum computers with $k$ qubits can represent $2^k$ numbers and manipulate them at the same time[28]. Recently, a quantum machine learning programming framework, TensorFlow-Quantum, has been proposed[29]; however, how to exploit the power of quantum computing for neural networks is still remained unknown.

One of the most challenging obstacles to implementing neural network computation on a quantum computer is the missing link between the design of neural networks and that of quantum circuits. The existing works separately design them from two directions. The first direction is to map the existing neural networks designed for classical computers to quantum circuits; for instance, recent works[30–35] map McCulloch-Pitts (MCP) neurons[36] onto quantum circuits. Such an approach has difficulties in consistently mapping the trained model to quantum circuits. For example, it needs a large number of qubits to realize the multiplication of real numbers. To overcome this problem, some existing works[30–33] assume binary representation (i.e., "−1" and "+1") of activation, which cannot well represent data as seen in modern machine learning applications. This has also been demonstrated in work[37], where data in the interval of $(0, 2\pi]$ instead of binary representation are mapped onto the Bloch sphere to achieve high accuracy for support vector machines (SVMs). In addition, some typical operations in neural networks cannot be implemented on quantum circuits, leading to inconsistency. For example, to enable deep learning, batch normalization is a key step in a deep neural network to improve the training speed, model performance, and stability; however, directly conducting normalization on the output qubit is difficult. In consequence, batch normalization is not applied in the existing multi-layer network implementation[31].

The other direction is to design neural networks dedicated to quantum computers, like the tree tensor network (TTN)[38,39]. Such an approach suffers from scalability problems. More specifically, the effectiveness of neural networks is based on a trained model via the forward and backward propagation on large training sets. However, it is too costly to directly train one network by applying thousands of times forward and backward propagation on quantum computers at current stage; in particular, there are limited available quantum computers for public access. An alternative way is to run a quantum simulator on a

classical computer to train models, but the time complexity of quantum simulation is $O(2^k)$, where $k$ is the number of qubits. This significantly restricts the trainable network size for quantum circuits.

In this work, we follow the network/hardware co-design philosophy[40,41] to address the above obstacles by considering the quantum circuit implementation during the design of neural networks and taking network property into consideration when synthesizing quantum circuits. Such a co-design framework, namely QuantumFlow, contains five sub-components (QF-pNet, QF-hNet, QF-FB, QF-Circ, and QF-Map) and they work collaboratively to design neural networks and implement them to quantum computers, as shown in Fig. 1. Based on three layer types (P-LYR, U-LYR, and N-LYR), QF-Nets contains two types of neural networks QF-pNet and QF-hNet, where QF-pNet applies P-LYR for neural computation while QF-hNet applies hybrid P-LYR and U-LYR. The QF-FB is designed to train the QF-Nets on classical computer. The trained QF-Nets can be deployed to quantum circuit QF-Circ for the inference, and then optimized by the QF-Map with the consideration of QF-Nets' property. Furthermore, QF-Map maps the virtual qubits to the physical qubits with the consideration of qubits' error rates. For detailed description of the components, please see Supplementary Note 1.

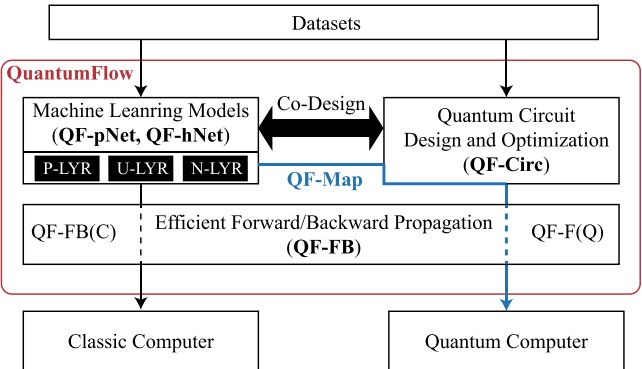

**Fig. 1 Illustration of the proposed end-to-end co-design framework, QuantumFlow.** QuantumFlow provides a missing link between neural network and quantum circuit designs, which is composed of QF-pNet, QF-hNet, QF-FB, QF-Circ, QF-Map that work collaboratively to design neural networks and their quantum implementations.

## Results

This section presents the evaluation results of all five sub-components in QuantumFlow. We first evaluate the effectiveness of QF-Nets (i.e., QF-pNet and QF-hNet) on the commonly used MNIST dataset[42] for the classification task. Then, we show the consistency between QF-FB(C) on classical computers and QF-F (Q) on the Qiskit Aer simulator. Next, we show that QF-Map is a key to achieve quantum advantage. We finally conduct an end-to-end case study on a binary classification test case on IBM quantum processors to test QF-Circ.

**QF-Nets achieve high accuracy on MNIST.** Figure 2a reports the results of different approaches for the classification of handwritten digits on the commonly used MNIST dataset[42]. Results clearly show that with the same network structure (i.e., the same number of layers and the same number of neurons in each layer), the proposed QF-hNet can achieve the highest accuracy than the existing models: (i) multi-level perceptron (MLP) with binary weights for the classical computer, denoted as MLP(C); (ii) MLP with binary inputs and weights designed for the classical

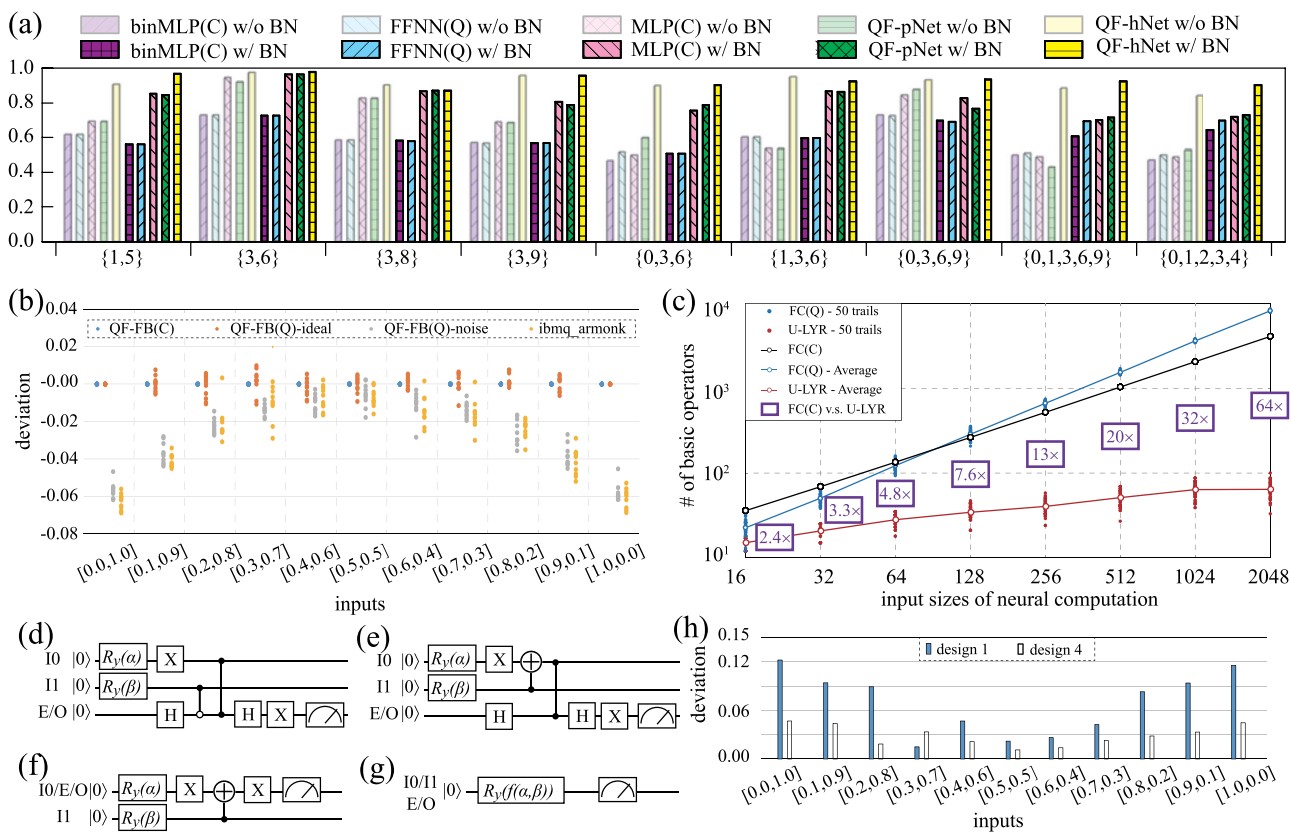

**Fig. 2 Experimental results. a** QF-hNet achieves state-of-the-art accuracy in image classifications on different sub-datasets of MNIST. **b** Output probability comparison on QF-FB(C), QF-F(Q)-ideal which assumes perfect qubits, QF-F(Q)-noise which applies noise model to "ibm_armonk" backend of IBM quantum processor, and results of circuit design ("design 4") in subfigure (i) on the same backend. **c** Demonstration of Quantum Advantage achieved by U-LYR in QuantumFlow: comparison is conducted by using 50 random generated weights for each input size. **d** Design 1 for a two-input neural computation, which follows the original neural computation design. **e–g** Three optimized designs (design 2–4), based on design 1. **h** The deviation of design 1 and design 4 obtained from "ibm_velencia" backend IBM quantum processor, using QF-FB(C) as golden results.

computer, denoted as binMLP(C); and (iii) a state-of-the-art quantum-aware neural network with binary inputs and weights[31], denoted as FFNN(Q).

Before reporting the detailed results, we first discuss the experimental setting. In this experiment, we extract sub-datasets from MNIST, which originally include 10 classes. For instance, {3, 6} indicates the sub-datasets with two classes (i.e., digits 3 and 6), which are commonly used in quantum machine learning (e.g., Tensorflow-Quantum[29]). To evaluate the advantages of the proposed QF-Nets, we further include more complicated sub-datasets, {3,8}, {3,9}, {1,5} for two classes. In addition, we show that QF-Nets can work well on larger datasets, including {0,3,6} and {1,3,6} for three classes, and {0,3,6,9}, {0,1,3,6,9}, {0,1,2,3,4} for four and five classes. For the datasets with two or three classes, the original image is downsampled from the resolution of 28 × 28 to 4 × 4, while it is downsampled to 8 × 8 for datasets with four or five classes. All original images in MNIST and the downsampled images are with gray levels. For all involved datasets, we employ a two-layer neural network, where the first layer contains 4 neurons for two-class datasets, 8 neurons for three-class datasets, and 16 neurons for four- and five-class datasets. The second layer contains the same number of neurons as the number of classes in datasets. Kindly note that these architectures are manually tuned for higher accuracy, the neural architecture search (NAS) will be our future work.

In the experiments, for each network, we have two implementations: one with batch normalization (w/BN) and one without batch normalization (w/o BN). Kindly note that FFNN[31]

does not consider batch normalization between layers. To show the benefits and generality of our newly proposed BN for improving the quantum circuits' accuracy, we add that same functionality to FFNN for comparison. From the results in Fig. 2a, we can see that the proposed "QF-hNetw/ BN" (abbr. QF-hNet-BN) achieves the highest accuracy among all networks (even higher than MLP running on classical computers). Specifically, for the dataset of {3, 6}, the accuracy of QF-hNet-BN is 98.27%, achieving 3.01% and 15.27% accuracy gain against MLP(C) and FFNN(Q), respectively. It even achieves a 1.17% accuracy gain compared to QF-pNet-BN. An interesting observation attained from this result is that with the increasing number of classes in the dataset, QF-hNet-BN can maintain the accuracy to be larger than 90%, while other competitors suffer an accuracy loss. Specifically, for dataset {0,3,6} (input resolution of 4 × 4), {0,3,6,9} (input resolution of 8 × 8), {0,1,3,6,9} (input resolution of 8 × 8), the accuracy of QF-hNet-BN are 90.40, 93.63, and 92.62%; however, for MLP(C), these figures are 75.37, 82.89, and 70.19%. This is achieved by the hybrid use of two types of neural computation in QF-hNet to better extract features in images. The above results validate that the proposed QF-hNet has a great potential in solving machine learning problems and our co-design framework is effective to design a quantum neural network with high accuracy.

Furthermore, we have an observation for our proposed batch normalization (BN). For almost all test cases, BN helps to improve the accuracy of QF-pNet and QF-hNet, and the most significant improvement is observed at dataset {1, 5}, from <70%

to 84.56% for QF-pNet and 90.33% to 96.60% for QF-hNet. Interestingly, BN also helps to improve MLP(C) accuracy significantly for dataset {1, 3, 6} (from <60% to 81.99%), with a slight accuracy improvement for dataset {3, 6} and a slight accuracy drop for dataset {3, 8}. This shows that the importance of batch normalization in improving model performance and the proposed BN is useful for quantum neural networks.

**QF-FB(C) and QF-F(Q) are consistent.** Next, we evaluate the results of QF-FB(C) for both QF-pNet and QF-hNet on classical computers, and that of QF-F(Q) simulation on classical computers for the quantum circuits QF-Circ built upon QF-Nets. Supplementary Table 2 reports the comparison results in the usage of qubits in QF-F(Q) and the inference accuracy, where results under Column QF-FB(C) are the golden results for accuracy comparison. Because of the limitation of Qiskit Aer (whose backend is "ibmq_qasm_simulator") used in QF-F(Q) that can maximally support 32 qubits, we measure the results after each neuron. This indicates that the outputs of a layer are independent of each other. However, if a quantum design is based on the same input qubits, it will involve correlation among neurons which may introduce error. In "Methods" section, we give two designs to eliminate such correlation to guarantee the consistency between the quantum implementation and the simulation. We select three datasets, including {3,6}, {3,8}, and {1,3,6}, for evaluation. Datasets with more classes (e.g., {0,3,6,9}) are based on larger inputs, which will lead to the usage of qubits in neuron computation to exceed the limitation (i.e., 32 qubits).

In Supplementary Table 2, columns "Layer Structure" gives the structure of a layer of neuron computation, based on which we can calculate the number of qubits used for each layer. Let $n_1 \rightarrow n_2$ be a layer with $n_1$ input neurons and $n_2$ output neurons. We first assume $n_2 = 1$, and we have the following formula for calculating the qubits used in three types of layer: P-LYR, U-LYR, and N-LYR.

$$Q_P(n_1) = n_1 + \log n_1 + (\log n_1 - 1), \quad (1)$$

where it has $n_1$ input qubits, $\log n_1$ encoding qbits, and $\log n_1 - 1$ ancillary qubits.

$$Q_U(n_1) = \log n_1, \quad (2)$$

where all $n_1$ inputs are encoded to $\log n_1$ qubits.

$$Q_N = \begin{cases} 0 & \text{if } \theta = 0 \text{ and } \gamma = 0 \\ 2 & \text{else if } \theta = 0 \text{ or } \gamma = 0 \text{ or } t = 0, \\ 4 & \text{otherwise} \end{cases} \quad (3)$$

where $\theta = 0$ (or $\gamma = 0$) indicates that batch_adj (or indiv_adj) is not applied, and $t = 0$ indicates that batch_adj and indiv_adj can be merged into one circuit. Finally, we can get the number of qubits used in a layer with $n_1$ inputs and $n_2$ outputs.

$$Q_{lyr}(n_1, n_2) = \begin{cases} (Q_P(n_1) + Q_N + 1) \times n_2 & \text{P} - \text{LYR} \\ (Q_U(n_1) + Q_N + 1) \times n_2 & \text{U} - \text{LYR} \end{cases}, \quad (4)$$

where "1" represents the output qubit, and we multiply $n_2$ to make outputs are independent to each other. Kindly note that we can further optimize U-LYR to encode $n_2 \times n_1$ inputs into $\log(n_2 \times n_1)$ qubits (see the last subsection in "Methods" section).

Based on these understandings, we obtain the number of qubits used in each layer as reported in Supplementary Table 2 under columns "qubits". For example, for L1 under QF-pNet in {3, 6} with 16 inputs and 4 outputs, it applies P-LYR and N-LYR with $t \neq 0$. According to Formula (4), the qubits are $Q_{lyr}(16, 4) = ((16 + 4 + 3) + 4 + 1) \times 4 = 28 \times 4$. From the results, it is clear to see that the U-LYR, which applied in L1 of QF-hNet, can significantly

reduce the number of required qubits. For dataset {3, 6}, it achieves 4× reduction, compared with P-LYR in L1 of QF-pNet.

Column "Accuracy" in Supplementary Table 2 reports the accuracy comparison. For QF-FB(C), there will be no difference in accuracy among different executions. For QF-F(Q), we implement the obtained QF-Circ from QF-Nets on Qiskit Aer simulation with 8192 shots. We have the following two observations from these results: (1) There exist accuracy differences between QF-FB(C) and QF-F(Q). This is because Qiskit Aer simulation used in QF-F(Q) is based on the Monte–Carlo method, leading to the variation. In addition, since the output probability of different neurons may quite close in some cases, it will easily result in different classification results for small variations. (2) Such accuracy differences for QF-hNet is much less than that of QF-pNet, because QF-pNet utilizes much more qubits, which leads to the accumulation of errors. In QF-hNet, we can see that there is a small difference between QF-FB(C) and QF-F(Q). For the dataset {3,8}, QF-F(Q) can even achieve higher accuracy. The above results demonstrate both QF-pNet and QF-hNet can be consistently implemented on classical and quantum computers.

In Fig. 2b, we further verify the accuracy of QF-FB by conducting a comparison for the optimized design with 1 qubit (will be introduced later) on IBM quantum processor with "ibm_armonk" backend. Kindly note that the quantum processor backend is selected by QF-Map. In this experiment, the result of QF-FB(C) is taken as a baseline. In the figure, the x- and y-axis represent the inputs and deviation, respectively. The deviation indicates the difference between the baseline and the results obtained by Qiskit Aer simulation or that by executing on IBM quantum processor. For comparison, we involve two configurations for QF-F(Q): (1) QF-F(Q)-ideal assuming perfect qubits; (2) QF-F(Q)-noise with error models derived from "ibm_armonk". We launch either simulation or execution for respective approaches 10 times, each of which is represented by a dot in Fig. 2b. We observe that the results of QF-F(Q)-ideal are distributed around that generated by QF-FB(C) within 1% deviation; while QF-F(Q)-noise obtains similar results of that on the IBM quantum processor. These results verify that the QF-Nets on the classical computer can achieve consistent results with that of QF-Circ deployed on a quantum computer with perfect qubits.

**QF-Map is the key to achieve quantum advantage.** Two sets of experiments are conducted to demonstrate the quantum advantage achieved by QuantumFlow. First, we conduct an ablation study to compare the operator/gate usage of the core computation component, neural computation layer. Then, the comparison of gate usage is further conducted on the trained neural networks for different sub-datasets from MNIST. In these experiments, we compare QuantumFlow to MLP(C) and FFNN(Q)[31]. For MLP(C), we consider the adder/multiplier as the basic operators, while for FFNN(Q) and QuantumFlow, we take the quantum logic gate (e.g., Pauli-X, Controlled Not, Toffoli) as the operators. The operator usage reflects the total cycles for neural computation. Kindly note that the results of QuantumFlow are obtained by using QF-Map on neural computation U-LYR; and that of FFNN(Q) are based on the state-of-the-art hypergraph state approach proposed in ref. [30]. For a fair comparison, QuantumFlow and FFNN(Q) are based on the same weights.

Figure 2c reports the comparison results for the core component in neural network, the neural computation layer. The x-axis represents the input size of the neural computation, and the y-axis stands for the number of basic operators used in the corresponding design. For quantum implementation (both FC

(Q)[30] in FFNN(Q)[31] and U-LYR in QuantumFlow), the value of weights will affect the gate usage, so we generate 50 sets of weights for each scale of input, and the dots on the lines in this figure represent average usage. From this figure, it clearly shows that the usage of FC(C) in MLP(C) on classical computing platforms grows exponentially along with the increase of inputs. The state-of-the-art quantum implementation FC(Q) has the similar exponentially growing trend. On the other hand, we can see that the growing trend of U-LYR is much slower. As a result, the reduction on operator usage continuously increases along with the growth of the input size of neural computation. For the input size of 16 and 32, the average reductions are 2.4× and 3.3×, compared with the implementations on classical computers. When the input size grows to 2048, the reduction increased to 64× on average. The reduction trends in operator usage in this figure clearly demonstrate the quantum advantage achieved by U-LYR. In the "Methods" section, we will demonstrate that for the neural computation with an input size of $n = 2^k$, the operator used in quantum implementation is $O(\log^2 n)$, while it is $O(n)$ on classical computers. We further investigate the commonly used time-space product complexity[2,43–46], which can better describe the tradeoff between time and space. For the quantum implementation, the circuit depth is less than the operator (i.e., gates), indicating the time complexity is $O(\log^2 n)$, while the space complexity (i.e., qubit numbers) is $O(\log n)$. Therefore, the time-space complexity is $O(\log^3 n)$, which is still lower than $O(n)$ on the classical computer.

Supplementary Table 3 reports the comparison results for the whole network. The neural network models for MNIST in Fig. 2a are deployed to quantum circuits to get the cost. In addition, to demonstrate the scalability, we further include a model for dataset "{0,1,3,6,9}*", which takes the larger sized inputs but less neurons in the first layer $L1$ and having higher accuracy over "{0,1,3,6,9}". In this table, columns $L1$, $L2$, and $Tot.$ under three approaches report the number of gates used in the first and second layers, and in the whole network. Columns "Red." represent the comparison with baseline $MLP(C)$.

From the Supplementary Table 3, it is clear to see that all cases implemented by QF-hNet can achieve cost reduction over MLP(C), while for datasets with more than 3 classes, FFNN(Q) needs more gates than MLP(C). A further observation made in the results is that QF-hNet can achieve higher cost reduction with the increase of input size. Specifically, for input size is 16, the reduction ranges from 1.05× to 1.63×. The reduction increases to 2.18× for input size is 64, and it continuously increases to 10.85× when the input size grows to 256. The above results are consistent with the results shown in Fig. 2c. It further indicates that even the second layer in QF-hNet uses the P-LYR which requires more gates for implementation, the quantum advantage can still be achieved for the whole network because the first layer using U-LYR can significantly reduce the number of gates. Above all, QuantumFlow demonstrates the quantum advantages on the MNIST dataset.

**QF-Circ on IBM quantum processor**. This subsection further evaluates the efficacy of QuantumFlow on IBM Quantum Processors. We first show the importance of quantum circuit optimization in QF-Circ to minimize the number of required qubits. Based on the optimized circuit design, we then deploy a 2-input binary classifier on IBM quantum processors.

Figure 2d–g demonstrates the optimization of a 2-input neuron step by step. All quantum circuits in Fig. 2d–g achieves the same functionality, but with a different number of required qubits. The equivalency of all designs will be demonstrated in Supplementary Note 4. Design 1 in Fig. 2d is directly derived from the design methodology presented in "Methods" section. To optimize the

circuit using fewer qubits, we first convert it to the circuit in Fig. 2e, denoted as design 2. Since there is only one controlled $Z$ gate from qubit I0 to qubit E/O, we can merge these two qubits, and obtain an optimized design in Fig. 2f with 2 qubits, denoted as design 3. The circuit can be further optimized to use 1 qubit, as shown in Fig. 2g, denoted as design 4. The function $f$ in design 4 is defined as follows:

$$f(\alpha, \beta) = 2 \cdot \arcsin(\sqrt{x + y - 2 \cdot x \cdot y}), \qquad (5)$$

where $x = \sin^2 \frac{\alpha}{2}$, $y = \sin^2 \frac{\beta}{2}$, representing input probabilities.

To compare these designs, we deploy them onto IBM Quantum Processors, where "ibm_velencia" backend is selected by QF-Map. In the experiments, we use the results from QF-FB(C) as the golden results. Figure 2h reports the deviations of designs 1 and 4 against the golden results. It clearly shows that design 4 is more robust because it uses fewer qubits. Specifically, the deviation of design 4 against golden results is always <5%, while reaching up to 13% for design 1. In the following experiments, design 4 is applied in QF-Circ.

Next, we are ready to introduce the case study on an end-to-end binary classification problem as shown in Fig. 3. In this case study, we train the QF-pNet based on QF-FB(C). Then, the tuned parameters are applied to generate QF-Circ. Finally, QF-Map optimizes the deployment of QF-Circ to IBM quantum processor, selecting the "ibmq_essex" backend.

The classification problem is illustrated in Fig. 3a, which is a binary classification problem (two classes) with two inputs: $x$ and $y$. For instance, if $x = 0.2$ and $y = 0.6$, it indicates class 0. The QF-pNet, QF-Circ, and QF-Map are demonstrated in Fig. 3b–d. First, Fig. 3b shows that QF-pNet consists of one hidden layer with one 2-input neuron and batch normalization. The output is the probability $p_0$ of class 0. Specifically, an input is recognized as class 0 if $p_0 \geq 0.5$; otherwise it is identified as class 1.

The quantum circuit QF-Circ of the above QF-pNet is shown in Fig. 3c. The circuit is composed of three parts, (1) neural computation, (2) batch_adj in batch normalization, and (3) indiv_adj in batch normalization. The neural computation is based on design 4 as shown in Fig. 2g. The parameter of $Ry$ gate in neural computation at qubit $q_0$ is determined by the inputs $x$ and $y$. Specifically, $f(x, y) = 2 \cdot \arcsin(\sqrt{x + y - 2 \cdot x \cdot y})$, as shown in Formula (5). Then, batch normalization is implemented in two steps, where qubits $q_2$ and $q_4$ are initialized according to the trained BN parameters. During the process, $q_1$ holds the intermediate results after batch_adj, and $q_3$ holds the final results after indiv_adj. Finally, we measure the output on qubit $q_3$.

After building QF-Circ, the next step is to map qubits from the designed circuit to the physical qubits on the quantum processor, and this is achieved through our QF-Map. In this experiment, QF-Map selects "ibm_essex" as backend with its physical properties shown in Fig. 3d, where error rates of each qubit and each connection are illustrated by different colors. By following the rules as defined by QF-Map (see "Methods" section), we obtain the physically mapped QF-Circ shown in Fig. 3d. For example, the input $q_0$ is mapped to the physical qubit labeled as 4.

After QuantumFlow goes through all the steps from input data to the physic quantum processor, we can perform inference on the quantum computer. In this experiment, we test 100 combinations of inputs from $\langle x, y \rangle = \langle 0.1, 0.1 \rangle$ to $\langle x, y \rangle = \langle 1.0, 1.0 \rangle$. First, we obtain the results using QF-FB(C) as golden results and QF-F(Q) as quantum simulation assuming perfect qubits, which are reported in Fig. 3e and f, achieving 100 and 98% prediction accuracy. The results verify the correctness of the proposed QF-pNet. Second, in Fig. 3g, we show the results obtained by using the default mapping algorithm in IBM Qiskit,

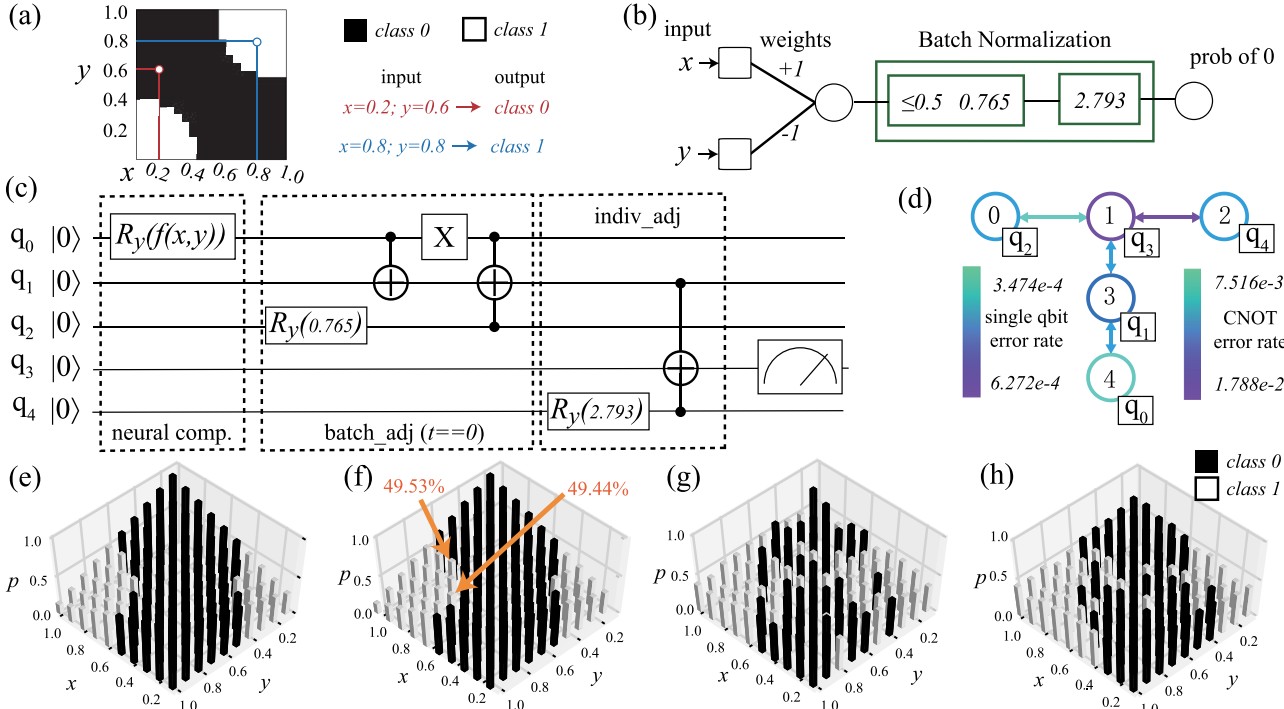

**Fig. 3 Results of a binary classification case study on IBM quantum processor of "ibmq_essex" backend. a** Binary classification with two inputs "$x$" and "$y$"; **b** QF-Nets with trained parameters; **c** QF-Circ derived from the trained QF-Nets where input is encoded by $f(x, y)$; **d** the virtual-to-physic mapping obtained by QF-Map upon "ibmq_essex" quantum processor; **e** QF-FB(C) achieves 100% accuracy; **f** QF-F(Q) achieves 98% accuracy where 2 marked error cases having probability deviation within 0.6%; **g** results on "ibmq_essex" using the default mapping, achieving 68% accuracy; **h** results obtained by "ibmq_essex" with the mapping in (**d**), achieving 82% accuracy; shots number in all tests is set as 8192.

whose accuracy is only 68%. Finally, the results obtained by QF-pNet on quantum processors are shown in Fig. 3h, which achieves 82% accuracy in prediction. The results demonstrate the value of QF-Map in improving the achievable accuracy on a physical quantum processor with errors.

## Discussion

In summary, we propose a holistic QuantumFlow framework to co-design the neural networks and quantum circuits. Novel quantum-aware QF-Nets are first designed. Then, an accurate and efficient inference engine, QF-FB, is proposed to enable the training of QF-Nets on classical computers. Based on QF-Nets and the training results, the QF-Map can automatically generate and optimize a corresponding quantum circuit, QF-Circ. Finally, QF-Map can further map QF-Circ to a quantum processor in terms of qubits' error rates.

The neural computation layer is one key component in QuantumFlow to achieve state-of-the-art accuracy and quantum advantage. We have shown in Fig. 2a that the existing quantum-aware neural network[31] that interprets inputs as the binary form will degrade the network accuracy. To address this problem, in QF-pNet, we first propose a probability-based neural computation layer, denoted as P-LYR, which interprets real number inputs as random variables following a two-point distribution. As shown in Table 1, P-LYR can represent both input and weight data using random variables, and it can directly connect layers without measurement. In summary, P-LYR provides better flexibility to perform neural computation than others; however, it suffers high complexity, i.e., the time-space product complexity of $O(n^2 \cdot \log n)$, where $n$ is the input number.

In order to acquire quantum advantages, we further propose a unitary matrix-based neural computation layer, called U-LYR. As

illustrated in Table 1, U-LYR sacrifices some degree of flexibility on data representation and nonlinear function but can significantly reduce the circuit complexity. Specifically, with the help of QF-Map, the cost complexity can be reduced from $O(n)$ on classical computing (i.e., FC(C)) to $O(\log^3 n)$, demonstrating the potential quantum advantage. Kindly note that this complexity involves neuron computation only. In addition to neuron computation, it also needs the quantum state-preparation, which can be conducted efficiently with the help of quantum random access memory (for details, see "Methods" section). Thus, we focus on the complexity of neural computation here.

Batch normalization is another key technique in improving accuracy, since the backbone of the quantum-friendly neuron computation layers (P-LYR and U-LYR) is similar to that in classical computers, using both linear and nonlinear functions. This can be seen from the results in Fig. 2a. Batch normalization can achieve better model accuracy, mainly because the data passing a nonlinear function $y^2$ will lead to outputs to be significantly shrunken to a small range around 0 for real number representation and $1/m$ for a two-point distribution representation, where $m$ is the number of inputs. Unlike straightforwardly doing normalization on classical computers, it is non-trivial to normalize a set of qubits. Innovations are made in QuantumFlow for a quantum-friendly normalization.

The philosophy of co-design is demonstrated in the design of P-LYR, U-LYR, and N-LYR. From the neural network design, we take the known operations as the backbones in P-LYR, U-LYR, and N-LYR; while from the quantum circuit design, we take full use of its ability in processing probabilistic computation and unitary matrix-based computations to make P-LYR, U-LYR, and N-LYR quantum-friendly. In addition, as will be shown in the next section, the key to achieve quantum advantage for U-LYR is that QF-Map fully considers the flexibility of the neural networks

**Table 1 Comparison of the implementation of a basic neural computation with $n$ input neurons and 1 output neuron.**

| Layers | | FC(C)[58] | FC(Q)[31] | P-LYR | U-LYR |
|---|---|---|---|---|---|
| Complexity | Space | $O(n)$ (or $O(1)$) | $O(\log n)$ | $O(n)$ | $O(\log n)$ |
| | Time | $O(1)$ (or $O(n)$) | $O(n)$ | $O(n \cdot \log n)$ | $O(\log^2 n)$ |
| | Cost | $O(n)$ | $O(n \cdot \log n)$ | $O(n^2 \cdot \log n)$ | $O(\log^3 n)$ |
| Data type | Input data | F32 | Bin | R.V. | F32 |
| | Weights | Bin (F32) | Bin | Bin (R.V.) | Bin |
| Conn. w/o measurement | | ✓ | – | ✓ | × |
| Summary | Flexibility | – | × | ✓ | × |
| | Qu. Adv. | – | × | × | ✓ |

(i.e., the order of inputs can be changed), while the requirement of continuously executing machine learning algorithms on the quantum computer leads to a hybrid neural network, QF-hNet, with both neural computation operations: P-LYR and U-LYR. Without the co-design, the previous works did not exploit quantum advantages in implementing neural networks on quantum computers, which reflects the importance of conducting co-design.

We have experimentally tested QuantumFlow on a 32-qubit Qiskit Aer simulator and a 5-qubit IBM quantum processor based on superconducting technology. We show that the proposed quantum oriented neural networks QF-Nets can obtain state-of-the-art accuracy on the MNIST dataset. It can even outperform the conventional model on a similar scale for the classical computer. For the experiments on IBM quantum processors, we demonstrate that, even with the high error rates of the current quantum processor, QF-Nets can be applied to classification tasks with high accuracy.

In order to accelerate the QF-FB on classical computers to support training, we make the assumptions that the perfect qubits are used. This enables us to apply theoretic formulations to accelerate the simulation process, while it leads to some error in predicting the outputs of its corresponding deployment on a physical quantum processor with high error rates (such as the current IBM quantum processor with error rates in the range of $10^{-2}$). However, we do not deem this as a drawback of our approach, rather this is an inherent problem of the current physical implementation of quantum processors. As the error rates get smaller in the future, it will help to narrow the gap between what QF-Nets predicts and what quantum processor delivers.

## Methods

In this section, we will first introduce the designed shallow neural network (with one hidden layer) in QuantumFlow; then, we will discuss how to extend QuantumFlow to support deep neural network. More specifically, neural computation and batch normalization are two key components in a neural network. We will first present the design and implementation of these two components in QF-Nets, QF-FB, QF-Circ, and QF-Map, respectively, for a shallow network. Then, we discuss how to utilize these components for a deep neural network.

**QF-pNet and QF-hNet**. Figure 4a–e demonstrates two different neural computation components in QuantumFlow: P-LYR and U-LYR. As stated in the "Discussion" section, P-LYR and U-LYR have their different features. Before introducing these two components, we demonstrate the common prepossessing step in Fig. 4a, which goes through the downsampling and gray level normalization to obtain a matrix with values in the range of 0–1. With the prepossessed data, we will discuss the details of each component in the following texts.

*Neural computation P-LYR*. An $m$-input neural computation component is illustrated in Fig. 4c, where $m$-input data $I_0, I_1, \cdots, I_{m-1}$ and $m$ corresponding weights $w_0, w_1, \cdots, w_{m-1}$ are given. Input data $I_i$ is a real number ranging from 0 to 1, while weight $w_i$ is a $\{-1, +1\}$ binary number. Neural computation in P-LYR is composed of 4 operations: (i) R: this operation converts a real number $p_k$ of input $I_k$ to a two-point distributed random variable $x_k$, where $P\{x_k = -1\} = p_k$ and $P\{x_k = +1\} = 1 - p_k$, as shown in Fig. 4b. For example, we treat the input $I_0$'s real value of $p_0$ as the probability of $x_0$ that outcomes $-1$ while $q_0 = 1 - p_0$ as the probability that

outcomes $+1$. (ii) C: this operation calculates $y$ as the average sum of weighted inputs, where the weighted input is the product of a converted input (say $x_k$) and its corresponding weight (i.e., $w_k$). Since $x_k$ is a two-point random variable, whose values are $-1$ and $+1$ and the weights are binary values of $-1$ and $+1$, if $w_k = -1$, $w_k \cdot x_k$ will lead to the swap of probabilities $P\{x_k = -1\}$ and $P\{x_k = +1\}$ in $x_k$. (iii) A: we consider the quadratic function as the nonlinear activation function in this work, and A operation outputs $y^2$ where $y$ is a random variable. (iv) E: this operation converts the random variable $y^2$ to 0–1 real number by taking its expectation. It will be passed to batch normalization to be further used as the input to the next layer.

*Neural computation U-LYR*. Unlike P-LYR taking advantage of the probabilistic properties of qubits to provide the maximum flexibility, U-LYR aims to minimize the gates for quantum advantage using the property of the unitary matrix. The $2^k$ input data are first converted to $2^k$ corresponding data that can be the first column of a unitary matrix, as shown in Fig. 4d. Then the linear function $C_u$ and activation quadratic function $A_u$ are conducted. U-LYR has the potential to significantly reduce the quantum gates for computation, since the $2^k$ inputs are the first column in a unitary matrix and can be encoded to $k$ qubits. But the state-of-the-art hypergraph based approach[30] needs $O(2^k)$ basic quantum gates to encode $2^k$ corresponding weights to $k$ qubits, which is the same as that of classical computer needing $O(2^k)$ operators (i.e., adder/multiplier). In the later section of QF-Map, we propose an algorithm to guarantee that the number of used basic quantum gates to be $O(k^2)$, achieving quantum advantages.

*Multiple layers*. P-LYR and U-LYR are the fundamental components in QF-Nets, which may have multiple layers. In terms of how a network is composed using these two components, we present two kinds of neural networks: QF-pNet and QF-hNet. QF-pNet is composed of multiple layers of P-LYR. For its quantum implementation, operations on random variables can be directly operated on qubits. Therefore, R operation is only conducted in the first layer. Then, C and A operations will be repeated without measurement. Finally, in the last layer, we measure the probability for output qubits, which is corresponding to the E operation. On the other hand, QF-hNet is composed of both U-LYR and P-LYR, where the first layer applies U-LYR with the converted inputs. The output of U-LYR is represented by the probability on a qubit, so it can seamlessly connect to C in P-LYR used in later layers.

*Batch normalization*. Figure 4f, g illustrates the proposed batch normalization (N-LYR) component. It can take the output of either P-LYR or U-LYR as input. N-LYR is composed of two sub-components: batch adjustment ("batch_adj") and individual adjustment ("indiv_adj"). Basically, batch_adj is proposed to avoid data to be continuously shrunken to a small range (as stated in "Discussion" section). This is achieved by normalizing the probability mean of a batch of outputs to 0.5 at the training phase, as shown in Fig. 5a, b. In the inference phase, we have two trained parameters ($t$ to indicate the movement direction and $\theta$ to control the movement distance), where $t = 0$ indicates that the mean of the probabilities for a batch of images is less than 0.5 which needs the downward movement and vice versa. Then, the output $\hat{z}$ can be computed as follows:

$$\hat{z} = (1 - z) \times \left(\sin^2 \frac{\theta}{2}\right) + z, \text{ if } t = 0$$
$$\hat{z} = z \times \left(\sin^2 \frac{\theta}{2}\right), \text{ if } t = 1$$

$$(6)$$

After batch_adj, the outputs of all neurons are normalized around 0.5. In order to increase the variety of different neurons' output for better classification, indiv_adj is proposed. It contains a trainable parameter $\lambda$ and a parameter $\gamma$ (see Fig. 5c). It is performed in two steps: (1) we get a start point of an output $p_z$ according to $\lambda$, and then moves it back to $p = 0.5$ to obtain parameter $\gamma$; (2) we move $p_z$ the angle of $\gamma$ to obtain the final output. Since different neurons have different values of $\lambda$, the variation of outputs can be obtained. In the inference

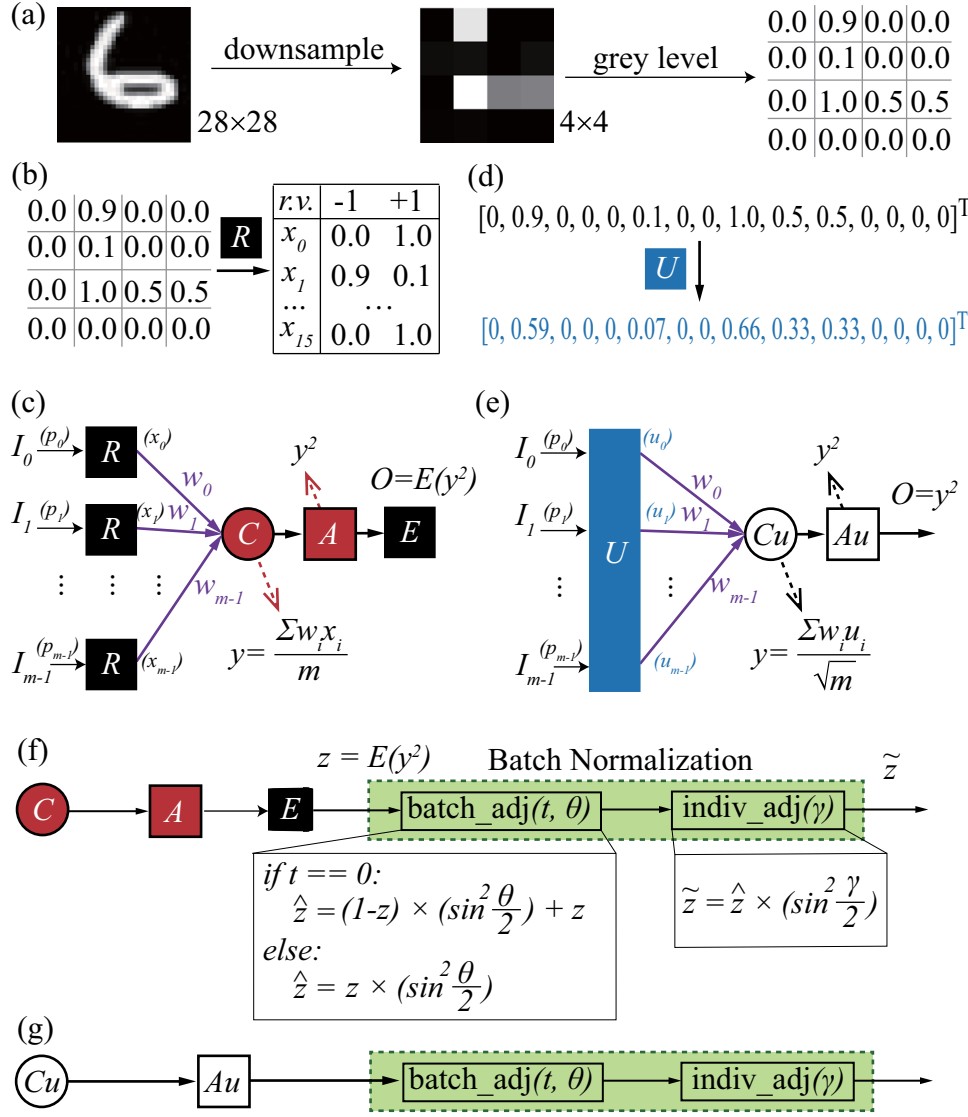

**Fig. 4 Neural computation. a** Prepossessing of inputs by (i) downsampling the original 28 × 28 image in MNIST to 4 × 4 image and (ii) get the 4 × 4 matrix and normalize gray level to [0, 1]. **b** In P-LYR, input data are converted from real number to a random variable following a two-point distribution. **c** Four operations in P-LYR, (i) R: converting a real number ranging from 0 to 1 to a random variable, (ii) C: average sum of weighted inputs, (iii) A: nonlinear activation function, (iv) E: converting random variable to a real number. **d** In U-LYR, $m$-input data are converted to a vector in the first column of a $m \times m$ unitary matrix. **e** Three operations in U-LYR, (i) U: unitary matrix converter, (ii) $C_u$: average sum of weighted inputs; (iii) $A_u$: nonlinear activation function. **f** Normalization for the connection of P-LYR. **g** Normalization for the connection of U-LYR.

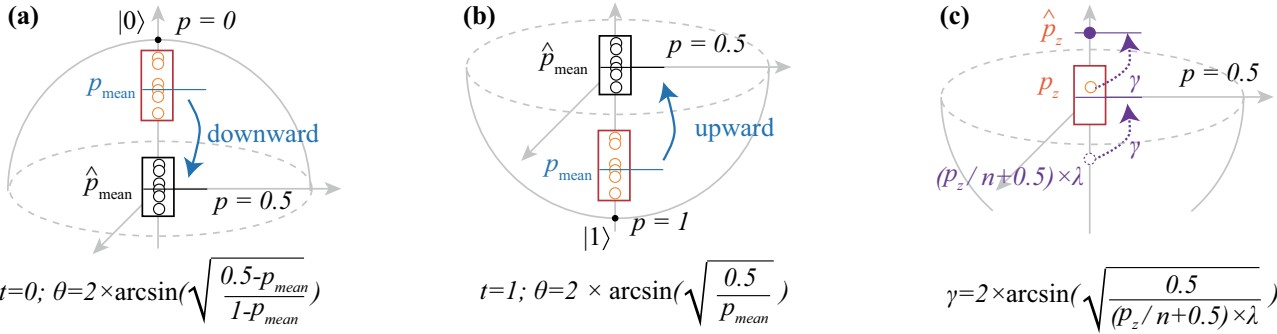

**Fig. 5 QF-FB: the determination of parameters, $t$, $\theta$, and $\gamma$, in batch normalization component of QF-Nets. a** Batch_adj when $t = 1$; **b** batch_adj when $t = 0$; **c** indiv_adj.

## Box 1 | Three algorithms in QF-FB

**Algorithm 1**: QF-FB: simulating P-LYR

**Input:** (1) number of inputs $m$; (2) $m$ probabilities $\langle p_0, \cdots, p_{m-1} \rangle$; (3) $m$ weights $\langle w_0, \cdots, w_{m-1} \rangle$.

**Output:** expectation of $y^2$

1. Expectation of random variable $x_i$: $e_i = E(x_i) = 1 - 2 \times p_i$;
2. Expectation of $w_i \times x_i$: $E(w_i \times x_i) = w_i \times e_i$;
3. Sum of pair product $sum_{pp} = \sum_{\forall i} \sum_{\forall j > i} \{E(w_i \times x_i) \times E(w_j \times x_j)\}$;
4. Expectation of $y^2$: $E(y^2) = \frac{m + 2 \times sum_{pp}}{m^2}$;
5. Return $E(y^2)$;

**Algorithm 2**: QF-FB: simulating U-LYR

**Input:** (1) number of inputs $m$; (2) $m$-input values $\langle p_0, \cdots, p_{m-1} \rangle$; (3) $m$ weights $\langle w_0, \cdots, w_{m-1} \rangle$.

**Output:** $\left[ \frac{\sum_{\forall i} \{u_i \times w_i\}}{\sqrt{(m)}} \right]^2$

1. Generating square matrix $A$ and compute $B \sum C^* = SVD(A)$
2. Calculating $MAT_u = BC^*$ and extract vector $U$ from $MAT_u$;
3. Compute $y = \frac{\sum_{\forall i} \{u_i \times w_i\}}{\sqrt{(m)}}$;
4. Return $y^2$;

**Algorithm 3**: QF-FB: simulating N-LYR

**Input:** (1) $E(y^2)$ from neural computation; (2) parameters $t$, $\theta$, $\gamma$ determined by training procedure.

**Output:** normalized output $\tilde{z}$

1. Initialize $z$: $z = E(y^2)$;
2. Calculate $\hat{z}$ according to Formula (6);
3. Calculate $\tilde{z}$ according to Formula (7);
4. Return $\tilde{z}$;

phase, its output $\tilde{z}$ can be calculated as follows.

$$\tilde{z} = \hat{z} \times \left( \sin^2 \frac{\gamma}{2} \right) \quad (7)$$

The determination of parameters $t$, $\theta$, and $\gamma$ is conducted in the training phase, which will be introduced later in QF-FB.

**QF-FB**. QF-FB involves both forward propagation and backward propagation. In forward propagation, all weights and parameters are determined, and we can conduct neural computation and batch normalization layer by layer. For P-LYR, the neural computation will compute $y = \frac{\sum_{\forall i} \{x_i \times w_i\}}{m}$ and $y^2$, where $x_i$ is a two-point random variable. The distributions of $y$ and $y^2$ are illustrated in Algorithms 1–3.

It is straightforward to get the expectation of $y^2$ by using the distribution; however, for $m$ inputs, it involves $2^m$ terms (e.g., $\prod q_i$ is one term), and leads to the time complexity to be $O(2^m)$. To reduce the time complexity, QF-FB takes advantage of independence of inputs to calculate the expectation as follows:

$$E\left( \left[ \sum_{\forall i} w_i x_i \right]^2 \right) = E\left( \sum_{\forall i} [w_i x_i]^2 + 2 \times \sum_{\forall i} \sum_{\forall j > i} [w_i x_i w_j x_j] \right)$$
$$= m + 2 \times \sum_{\forall i} \sum_{\forall j > i} E(w_i x_i) \times E(w_j x_j) \quad (8)$$

where $E(\sum_{\forall i} [w_i x_i]^2) = m$, since $[w_i x_i]^2 = 1$ and there are $m$ inputs in total. The above formula derives Algorithm 1 with time complexity of $O(m^2)$ to simulate the neural computation P-LYR.

For U-LYR, the neural computation will first convert inputs $I = \{i_0, i_1, \cdots, i_{m-1}\}$ to a vector $U = \{u_0, u_1, \cdots, u_{m-1}\}$ who can be the first column of a unitary matrix $MAT_u$. By operating $MAT_u$ on $K = \log_2 m$ qubits with initial state (i.e., $|0\rangle$), we can encode $U$ to $2^K = m$ states. The generating of unitary matrix $MAT_u$ is equivalent to the problem of identifying the nearest orthogonal matrix given a square matrix $A$. Here, matrix $A$ is created by using $I$ as the first column, and 0 for all other elements. Then, we apply singular value decomposition (SVD) to obtain $B \sum C^* = SVD(A)$, and we can obtain $MAT_u = BC^*$. Based on the obtained vector $U$ in $MAT_u$, U-LYR computes $y = \frac{\sum_{\forall i} \{u_i \times w_i\}}{\sqrt{(m)}}$ and $y^2$, as shown in Algorithm 2.

The forward propagation for batch normalization can be efficiently implemented based on the output of the neural computation. A code snippet is given in Algorithm 3.

For the backward propagation, we need to determine weights and parameters (e.g., $\theta$ in N-LYR). The typically used optimization method (e.g., stochastic gradient

descent[47]) is applied to determine weights. In the following, we will discuss the determination of N-LYR parameters $t$, $\theta$, $\gamma$.

The batch_adj sub-component involves two parameters, $t$ and $\theta$. During the training phase, a batch of outputs are generated for each neuron. Details are demonstrated in Fig. 5a, b with 6 outputs. In terms of the mean of outputs in a batch $p_{mean}$, there are two possible cases: (1) $p_{mean} \leq 0.5$ and (2) $p_{mean} > 0.5$. For the first case, $t$ is set to 0 and $\theta = 2 \times \arcsin\left( \sqrt{\frac{0.5 - p_{mean}}{1 - p_{mean}}} \right)$ can be derived from Formula (6) by setting $\hat{z}$ to 0.5; similarly, for the second case, $t$ is set to 1 and $\theta = 2 \times \arcsin\left( \sqrt{\frac{0.5}{p_{mean}}} \right)$. Kindly note that the training procedure will be conducted in multiple iterations of batches. As with the method for batch normalization in the conventional neural network, we employ moving average to record parameters. Let $x_i$ be the parameter of $x$ (e.g., $\theta$) at the $i$th iteration, and $x_{cur}$ be the value obtained in the current iteration. For $x_i$, it can be calculated as $x_i = m \times x_{i-1} + (1 - m) \times x_{cur}$, where $m$ is the momentum which is set to 0.1 by default in the experiments.

In forward propagation, the sub-module indiv_adj is almost the same as batch_adj for $t = 0$; however, the determination of its parameter $\gamma$ is slightly different from $\theta$ for batch_adj. As shown in Fig. 5c, the initial probability of $\hat{z}$ after batch_adj is $p_z$. The basic idea of indiv_adj is to move $\hat{z}$ by an angle, $\gamma$. It will be conducted in three steps: (1) we move the start point at $p_z$ to point $A$ with the probability of $(p_z/n + 0.5) \times \lambda$, where $n$ is the batch size and $\lambda$ is a trainable variable; (2) we obtain $\gamma$ by moving point $A$ to $p = 0.5$; (3) we finally move solution at $p_z$ by the angle of $\gamma$ to obtain the final result. By replacing $P_{mean}$ by $(p_z/n + 0.5) \times \lambda$ in batch_adj when $t = 1$, we can calculate $\gamma$. For each batch, we calculate the mean of $\gamma$, and we also employ the moving average to record $\gamma$.

**QF-Circ**. We now discuss the corresponding circuit design for components in QF-Nets, including P-LYR, U-LYR, and N-LYR. Figure 6a, b demonstrates the circuit design for P-LYR (see Fig. 4c) and U-LYR (see Fig. 4e), respectively; Fig. 6c–f demonstrates the N-LYR in Fig. 4.

*Implementing P-LYR on quantum circuit.* For an $m$-input neural computation, the quantum circuit for P-LYR is composed of $m$-input qubits (I), and $k = \log_2 m$ encoding qubits (E), and 1 output qubit (O).

In accordance with the operations in P-LYR, the circuit is composed of four parts. In the first part, the circuit is initialized to perform R operation. For qubits $I$, we apply $m$ Ry gates with parameter $\theta = 2 \times \arcsin(\sqrt{p_k})$ to initialize the input qubit $I_k$ in terms of the input real value $p_k$, such that the input of $I_k$ is changed from $|0\rangle$ to $\sqrt{q_k}|0\rangle + \sqrt{p_k}|1\rangle$. For encoding qubits $E$ and output qubit $O$, they are initialized as $|0\rangle$. The second part completes the average sum function, i.e., C operation. It further includes three steps: (1) dot product of inputs and weights on qubits $I$, (2) make encoding qubits $E$ into superposition, (3) encode $m$ probabilities in qubits $I$ to $2^k = m$ states in qubits $E$. The third part implements the quadratic activation function, that is the A operation. It applies the control gate to extract the amplitudes in states $|I_0 I_1 \cdots I_{m-1}\rangle \otimes |00 \cdots 0\rangle$ to qubit $O$. As we know that the probability is the square of the amplitude, the quadratic activation function can be naturally implemented. Finally, E operations correspond to the fourth part that measures qubit $O$ to obtain the output real number $E(y^2)$, where the state of $O$ is $|O\rangle = \sqrt{1 - E(y^2)}|0\rangle + \sqrt{E(y^2)}|1\rangle$. A detailed demonstration of the equivalency between QF-Circ and P-LYR can be found in Supplementary Note 3.

Kindly note that for a multi-layer network composed of P-LYR, namely QF-pNet, there is no need to have a measurement at interfaces, because the converting operation R initializes a qubit to the state exactly the same with $|O\rangle$. In addition, the batch normalization can also take $|O\rangle$ as input.

*Implementing U-LYR on quantum circuit.* For an $m$-input neural computation, the quantum circuit for U-LYR contains $k = \log_2 m$ encoding qubits $E$ and 1 output qubit $O$.

According to U-LYR, the circuit in turn performs U, $C_u$, $A_u$ operations, and finally obtains the result by a measurement. In the first operation, unlike the circuit for P-LYR using R operation to initialize circuits using $m$ qubits; for U-LYR, we using the matrix $MAT_u$ to initialize circuits on $k = \log_2 m$ qubits $E$. Recalling that the first column of $MAT_u$ is vector $V$, after this step, $m$ elements in vector $V$ will be encoded to $2^k = m$ states represented by qubits $E$. The second operation is to perform the dot product between all states in qubits $E$ and weights $W$, which is implemented by control Z gates and will be introduced in QF-Map. Finally, like the circuit for P-LYR, the quadratic activation and measurement are implemented. Kindly note that, in addition to quadratic activation, we can also implement higher orders of non-linearity by duplicating the circuit to perform U, $C_u$, and $A_u$ to achieve multiple outputs. Then, we can use control NOT gate on the outputs to achieve higher orders of non-linearity. For example, using a Toffoli gate on two outputs can realize $y^4$. Let the nonlinear function be $y^k$ and the cost complexity of U-LYR using quadratic activation be $O(N)$, then the cost complexity of U-LYR using $y^k$ as the nonlinear function will be $O(kN)$.

There are two main steps in U-LYR: the state-preparation (i.e., U or encoding $MAT_u$) and the weighted sum computation (i.e., $C_u$ and $A_u$). For the state-preparation, since the inputs are given, we apply the same approach in works[48,49] by using quantum random access memory (qRAM)[50]. Specifically, the vector in

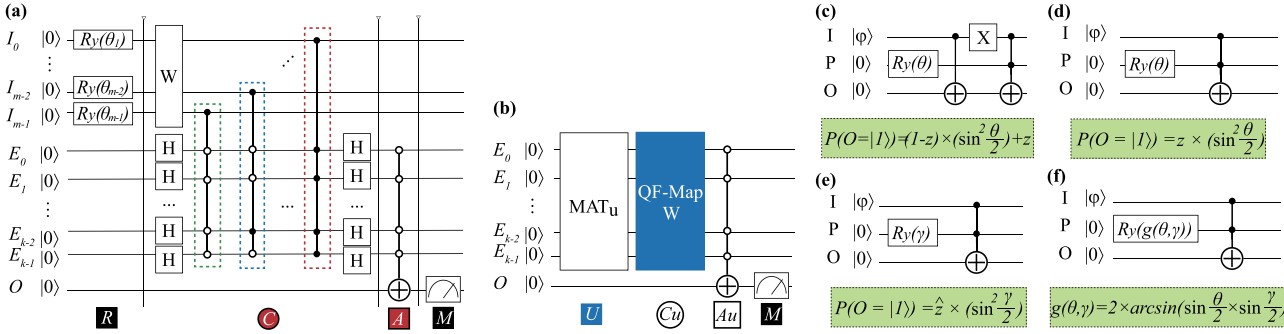

**Fig. 6 The corresponding implementation of operations (e.g., R, C, A) in QF-Circ. a** Quantum circuit designs for QF-pNet; **b** quantum circuit design for QF-hNet; **c–f** batch normalization, quantum circuit designs for different cases; **c** design of "batch_adj" for the case of $t = 0$; **d** design of "batch_adj" for the case of $t = 1$; **e** design of "indiv_adj"; **f** optimized design for a specific case when $t = 1$ in "batch_adj".

---

### Box 2 | Algorithm in QF-Map

**Algorithm 4**: Weights mapping with adjustable inputs.

**Input:** (1) An integer $R \in (0, 2^{k-1}]$; (2) number of qubits $k$;
**Output:** A set of applied gate $G$

```
void recursive(G,R,k) {
  if (R < 2^{k-2}) {
    recursive(G,R,k − 1); // Case 1 in the third step
  }
  else if (R == 2^{k-1}) {
    G.append(PG_{2^{k-1}}); // Case 2 in the third step
    return;
  } else {
    G.append(PG_{2^{k-1}});
    recursive(G,2^{k-1} − R,k − 1); // Case 3 in the third step
  }
}
// Entry of weight mapping algorithm
set main(R,k){
  Initialize empty set G;
  recursive(G,R,k);
  return G
}
```

MAT$_u$ will be stored in a binary-tree based structure in qRAM, which can be queried in quantum superposition and can generate the states efficiently. Therefore, the key to obtain the quantum advantage in a U-LYR neuron computation is to reduce the used number of gates. We will present an algorithm in QF-Map for U-LYR to achieve this goal.

*Implementing N-LYR on quantum circuit.* Now, we discuss the implementation of N-LYR in quantum circuits. In these circuits, three qubits are involved: (1) qubit $I$ for input, which can be the output of qubit $O$ in circuit without measurement, or initialized using a Ry gate according to the measurement of qubit $O$ in circuit; (2) qubit $P$ conveys the parameter, which is obtained via training procedure, see details in QF-FB; (3) output qubits $O$, which can be directly used for the next layer or be measured to convert to a real number.

Figure 6c, d shows the circuit design for two cases in batch_adj. Since parameters in batch_adj are determined in the inference phase, if $t = 0$, we will adopt the circuit in Fig. 6c, otherwise, we adopt that in Fig. 6d. Then, Fig. 6e shows the circuit for indiv_adj. We can see that circuits in Fig. 6d, e are the same, except the initialization of parameters, $\theta$ and $\gamma$. For circuit optimization, we can merge the above two circuits into one by changing the input parameters to $g(\theta, \gamma)$, as shown in Fig. 6f. In this circuit, $\tilde{z}' = z \times \sin^2 \frac{g(\theta, \gamma)}{2}$, while for applying circuits in Fig. 6d, e, we will have $\tilde{z} = z \times \sin^2 \frac{\theta}{2} \times \sin^2 \frac{\gamma}{2}$. To guarantee the consistent function, we can derive that $g(\theta, \gamma) = 2 \times \arcsin(\sin \frac{\theta}{2} \times \sin \frac{\gamma}{2})$.

**QF-Map**. QF-Map is an automatic tool to map QF-Nets to the quantum processor through two steps: network-to-circuit mapping, which maps QF-Nets to QF-Circ; and virtual-to-physic mapping, which maps QF-Circ to physical qubits.

*Mapping QF-Nets to QF-Circ*. The first step of QF-Map is to map three kinds of layers (i.e., P-LYR, U-LYR, and N-LYR) to QF-Circ. The mappings of P-LYR and N-LYR are straightforward. Specifically, for P-LYR in Fig. 6a, the circuit for weight $W$ is determined using the following rule: for a qubit $I_k$, an X gate is placed if and only if $W_k = -1$. Let the probability $P(x_k = -1) = P(I_k = |0\rangle) = q_k$, after the X gate, the probability becomes $1 - q_k$. Since the values of random variable $x_k$ are $-1$ and $+1$, such an operation computes $-x_k$. For N-LYR, let $O_1$ be the output qubit of the first layer. It can be directly connected to the qubit $I$ in Fig. 6c–f, according to the type of batch normalization, which is determined by the training phase. The mapping of U-LYR to quantum circuits is the key to achieve quantum advantages. In the following texts, we will first formulate the problem, and then introduce the proposed algorithm to guarantee the cost for a neural computation with $2^k$ inputs to be $O(k^2)$.

Before formally introducing the problem, we first give some fundamental definitions that will be used. We first define the quantum state and the relationship between states as follows. Let the computational basis be composed of $k$ qubits, as in Fig. 6b. Define $|x_i\rangle = \left| b_{k-1}^i, \cdots, b_j^i, \cdots, b_0^i \right\rangle$ to be the $i^{th}$ state, where $b_j$ is a binary number and $x_i = \sum_{\forall j} \{ b_j^i \cdot 2^j \}$. For two states $\left| x_p \right\rangle$ and $\left| x_q \right\rangle$, we define $\left| x_p \right\rangle \subseteq \left| x_q \right\rangle$ if $\forall b_j^p = 1$, we have $b_j^q = 1$. We define $sign(x_i)$ to be the sign of $x_i$.

Next, we define the gates to flip the sign of states. The controlled $Z$ operation among $K$ qubits (e.g., $C^K Z$) is a quantum gate to flip the sign of states[28,30]. Define $FG_{x_i}$ to be a $C^K Z$ gate to flip the state $x_i$ only. It can be implemented as follows: if $b_j^i = 1$, the control signal of the $j$th qubit is enabled by $|1\rangle$, otherwise if $b_j^i = 0$, it is enabled by $|0\rangle$. Define $PG_{x_i}$ to be a controlled $Z$ gate to flip all states $\forall x_m$ if $x_m \subseteq x_i$. It can be implemented as follows: if $b_j^i = 1$, there is a control signal of the $j$th qubit, enabled by $|1\rangle$, otherwise, it is not a control qubit. Specifically, if there is only $b_m^i = 1$ for all $b_k^i \in x_i$, we put a $Z$ gate on the $m$th qubit. Supplementary Figure 3 illustrates $FG_6$, $PG_6$, and $PG_4$. We define cost function $\mathbb{C}$ to be the number of basic gates (e.g., Pauli-X, Toffoli) used in a control $Z$ gate.

Now, we formally define the weight mapping problem as follows: Given (1) a vector $W = \{ w_{m-1}, \cdots, w_0 \}$ with $m$ binary weights (i.e., $-1$ or $+1$) and (2) a computational basis of $k = \log_2 m$ qubits that include $m$ states $X = \{ x_{m-1}, \cdots, x_0 \}$ and $\forall x_j \in X$, $sign(x_j)$ is $+$, the problem is to determine a set of gates $G$ in either FG or PG, such that the circuit cost is minimized, while the sign of each state is the same with the corresponding weight; i.e., $\forall x_j \in X$, $sign_G(x_j) = sign(w_j)$ and $min = \sum_{g \in G}(\mathbb{C}(g))$, where $sign_G(x_j)$ is the sign of state $x_j$ obtained by a sequence of quantum gates in $G$.

A straightforward way to satisfy the sign flip requirement without considering cost is to apply FG for all states whose corresponding weights are $-1$. A better solution for cost minimization is to use hypergraph states[30], which starts from the states with less $|1\rangle$, and apply PG to reduce the cost. However, as shown in the previous work, both methods have the cost complexity of $O(2^k)$, which is the same as classical computers and no quantum advantage can be achieved.

Toward the quantum advantage, we made the following important observation: the order of weights can be adjusted, since matrix MAT$_u'$ obtained by switching rows in the unitary matrix MAT$_u$ will still be a unitary matrix. Kindly note that all weights are determined in the inference phase, indicating that MAT$_u'$ can be obtained in the pre-processing phase. Therefore, we can efficiently conduct state-preparation (i.e., $U$ operation) in U-LYR, similarly to MAT$_u$. Based on the above property, we can simplify the weight mapping problem to determine a set of gates, such that the number of gates is minimized while the number of states with sign flip is the same as the number of $-1$ in weight $W$. On top of this, we propose an algorithm to guarantee the number of used gates and the depth of circuit (time complexity) for the computation (i.e., $C_u$ and $A_u$ operations) in U-LYR to be $O(k^2)$. To demonstrate this, we first give the following theorem.

**Theorem 1** *For an integer number $R$ where $R > 0$ and $R \le 2^{k-1}$, the number $R$ can be expressed by a sequence of addition ($+$) or subtraction ($-$) of a subset of $S_k = \{2^i | 0 \le i < k\}$; when the terms of $S_k$ are sorted in a descending order, the sign of the expression (addition and subtraction) are alternative with the leading sign being addition ($+$).*

**Proof** The above theorem can be proved by induction. First, for $k = 2$, the possible values of $R$ are $\{1, 2\}$, the set $S_2 = \{2, 1\}$. The theorem is obviously true. Second, for $k = 3$, the possible values of $R$ are $\{1, 2, 3, 4\}$, and the set $S_3 = \{4, 2, 1\}$. In this case, only $R = 3$ needs to involve 2 numbers from $S_3$ using the expression $3 = 4 - 1$; other numbers can be directly expressed by themselves. So, the theorem is true.

Third, assuming the theorem is true for $k = n - 1$, we can prove that for $k = n$ the theorem is true for the following three cases. Case 1: For $R < 2^{n-2}$, since the theorem is true for $k = n - 1$, based on the assumption, all numbers $<2^{n-2}$ can be expressed by using set $S_{n-1}$ and thus we can also express them by using set $S_n$ because $S_{n-1} \subseteq S_n$; Case 2: For $R = 2^{n-1}$, itself is in set $S_n$; Case 3: For $R > 2^{n-2}$, we can express $R = 2^{n-1} - T$, where $T = 2^{n-1} - R < 2^{n-2}$. Since the theorem is true for $k = n - 1$, we can express $T$ by using set $S_n - 2^{n-1} = \{2^i | 0 \le i < n - 1\} = S_{n-1}$; hence, $R$ can be expressed using set $S_n$.

Above all, the theorem is correct.

We propose to only use PG gate in a set $\{PG_{x(j)} | x(j) = 2^j - 1 \ \& \ j \in [1, k]\}$ for any required number $R \in (0, 2^k)$ of sign flips on states; for instance, if $k = 4$, the gate set is $\{PG_{0001}, PG_{0011}, PG_{0111}, PG_{1111}\}$ and $R \in (0, 16)$. This can be demonstrated using the above theorem and properties of the problem: (1) the problem has symmetric property due to the quadratic activation function. Therefore, the weight mapping problem can be reduced to find a set of gates leading to the number of $-1$ no larger than $2^{k-1}$; i.e., $R \in (0, 2^{k-1}]$. (2) for $PG_{x(j)}$, it will flip the sign of $2^{k-j}$ states; since $j \in [1, k]$, the numbers of the flipped sign by these gates belong to a set $S_k = \{2^i | 0 \le i < k\}$; These two properties make the problem in accordance with that in Theorem 1. The weight mapping problem is also consistent with three rules in the theorem. (1) A gate can be selected or not, indicating the finally determined gate set is the subset of $S_k$; (2) all states at the beginning have the positive sign, and therefore, the first gate will increase sign flips, indicating the leading sign is addition ($+$); (3) $\forall p < q$, $x(q) \subseteq x(p)$; it indicates that among the $2^{k-p}$ states whose signs are flipped by $x(p)$, there are $2^{k-q}$ states signs are flipped back; this is in accordance to alternatively use $+$ and $-$ in the expression in Theorem 1. Followed by the proof, we devise a recursive algorithm to decide which gates to be employed.

In the above algorithm, the worst case for the cost is that we apply all gates in $\{PG_{x(j)} | x(j) = 2^j - 1 \ \& \ j \in [1, k]\}$. Let the state $x(j)$ has $y | 1\rangle$ states, if $y > 2$ the $PG_{x(j)}$ can be implemented using $2y - 1$ basic gates, including $y - 1$ Toffoli gates for controlling, 1 control Z gate, and $y - 1$ Toffoli gates for resetting; otherwise, it uses 1 basic gates. Based on these understandings, we can calculate the cost complexity in the worst case, which is $1 + 1 + 3 + \cdots + (2 \times k - 1) = k^2 + 1$. Therefore, the cost complexity of linear function computation is $O(k^2)$. The quadratic activation function is implemented by a $C^k Z$ gate, whose cost is $O(k)$. Thus, the cost complexity for neural computation U-LYR is $O(k^2)$.

Finally, to make the functional correctness, in generating the inputs unitary matrix, we swap rows in it in terms of the weights, and store the generated results in quantum memory.

*Mapping QF-Circ to physical qubits.* After QF-Circ is generated, the second step is to map QF-Circ to quantum processors, called virtual-to-physic mapping. We deploy QF-Circ to various IBM quantum processors. Virtual-to-physic mapping in QF-Map has two tasks: (1) select a suitable quantum backend, and (2) map qubits in QF-Nets to physical qubits in the selected backend. For the first task, QF-Map will (i) check the number of qubits needed; (ii) find the backend with the smallest number of qubit to accommodate QF-Circ; (iii) for the backends with the same number of qubits, QF-Map will select a backend for the minimum average error rate. The second task in QF-Map is to map qubits in QF-Nets to physical qubits. The mapping follows two rules: (1) the qubit in QF-Nets with more gates is mapped to the physic qubit with a lower error rate, and (2) qubits in QF-Nets with connections are mapped to the physical qubits with the smallest distance.

**Applying U-LRY in deep neural networks**. This section will discuss how QuantumFlow can be used for deep neural networks. The inherent property of correlation among qubits in quantum computing and the requirement of independence requirement in neural networks will lead to high space complexity along with the increase of the network depth. An extension of QF-hNet and its complexity analysis are presented in Supplementary Note 5, where the full network will be implemented on quantum circuits without measurement. An alternative approach is to apply the hybrid quantum-classical scheme and measure the outputs at the end of each layer, so that the correlation can be eliminated. Such a hybrid quantum-classical scheme has been widely used in quantum machine learning[29,51–54]. In the following texts, we will discuss how to apply U-LRY in the hybrid quantum-classical scheme and analyze its cost complexity.

*Construction.* The extended design using U-LYR on hybrid quantum-classical computing iteratively conducts 4 steps for layers in sequence: (1) encoding the input neurons to input qbits; (2) do neural computation with pre-trained weights; (3) measurement the output qbits to generate output neurons; (4) translate the

**Table 2 Complexity of each step in hybrid quantum-classical computing for deep neural network with U-LYR.**

| Complexity | State-preparation | Computation | Measurement |
|---|---|---|---|
| Depth (T) | $O(d \cdot \sqrt{n})$ | $O(d \cdot \log^2 n)$ | $O(d)$ |
| Qubits (S) | $O(n)$ | $O(n \cdot \log n)$ | $O(n \cdot \log n)$ |
| Cost (TS) | $O(d \cdot n^{\frac{3}{2}})$ | $O(d \cdot n \cdot \log^3 n)$ | $O(d \cdot n \cdot \log n)$ |
| Total (TS) | $O(d \cdot n^{\frac{3}{2}})$ | | |

output neurons according to the pre-determined weights in the next layer. In the above procedure, steps (1) and (3) are the interface between classical and quantum computing, while step (2) is on quantum computing, and step (4) is conducted classical computing. Step (1) is the well-known quantum state-preparation. It can be implemented in different ways, such as qRAM based approach[48,49] and computing-based approach[55–57]. In step (2), we apply U-LYR for the computation. Step (3) will measure the output qubits and send the results to step (4) for the data preparation according to the weight in the next layer and the results of Algorithm 4. These 4 steps will be iteratively conducted $d$ times to reach the output neurons for a $d$-layer network.

*Cost complexity.* We adopt the widely used time-space product complexity[2,43–46] as the cost complexity. In quantum computing, time and space correspond to the circuit depth and the number of qubits. For a fair comparison, in classical computing, time and space correspond to the computing latency and computing storage. We consider computing storage instead of total storage because there is no need to load all data to on-chip storage during computation.

In the complexity analysis, we focus on the cost of steps (1)–(3). Note that the weights are given at the inference phase. This indicates a fixed mapping in step (4) from outputs in $i$th layer to inputs $(i+1)$th layer, which can be implemented by classical hardware at a constant cost. Let $d$ be the number of layers and let the number of input and output neurons for each layer is $O(n)$. Table 2 gives the detailed complexity analysis for each step. In step (1), we apply the computation-based approach[57] for state-preparation, where the time complexity (i.e., circuit depth) for each layer is $O(\sqrt{n})$ and it is $O(d \cdot \sqrt{n})$ for $d$ layers. Its space complexity (i.e., qubits) is $O(n \cdot \log g)$ for $O(n)$ outputs, where $g$ is related to the precision of amplitudes. In analyzing the quantum advantage, we use the same finite precision with classical computer; therefore, $g$ can be regarded as a constant, leading the space complexity to be $O(n)$. In step (2), we apply the proposed U-LYR for the computation. Its time complexity is $O(\log^2 n)$ for each layer, and $O(d \cdot \log^2 n)$ for $d$ layers. According to Formula (2), the space complexity is $O(\log n)$ for 1 output, and $O(n \cdot \log n)$ for $O(n)$ outputs. Finally, the measurement takes $O(1)$ time complexity for each layer and $O(d)$ for $d$ layers. Each output corresponds to a qubit, and therefore the space complexity is $O(n)$. Overall, we can obtain the cost time-space product complexity for the whole system to be $O(d \cdot n^{\frac{3}{2}})$.

It is obvious that the state-preparation dominates the time complexity, but it still demonstrates that the potential advantage can be achieved for executing neural networks on quantum computing against classical one. For the classical computing, the time-space product complexity is $O(d \cdot n^2)$ (for details, see Supplementary Note 6), which is obviously higher than $O(d \cdot n^{\frac{3}{2}})$ on hybrid quantum-classical computing. With the improvement of the state-preparation protocols, the overall complexity can be further reduced.

## Data availability

The authors declare that all data supporting the findings of this study are available within the article and its Supplementary Information files. Source data can be accessed via https://github.com/weiwenjiang/QuantumFlow/blob/master/Quantumflow_Data.xlsx.

## Code availability

All code scripts have been made available at https://github.com/weiwenjiang/QuantumFlow.

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

## Acknowledgements

This work is partially supported by IBM and University of Notre Dame (IBM-ND) Quantum program, and in part by the IBM-ILLINOIS Center for Cognitive Computing Systems Research.

## Author contributions

W.J. conceived the idea and performed quantum evaluations; J.X. and Y.S. supervised the work and improved the idea and experiment design. All authors contributed to manuscript writing and discussions about the results.

## Competing interests

The authors declare no competing interests.
