## [Peer Review File · Nature Communications]

Reviewers' Comments:

Reviewer #1:

Remarks to the Author:

The manuscript "Can Quantum Computers Learn Like Classical Computers? A Co-Design Framework for Machine Learning and Quantum Circuits" has the ambitious goal of creating a framework to unify classical machine learning and quantum circuits. The authors introduce QuantumFlow (QF) and its various parts: a network based on stochastic neurons (QF-Net), a tool to convert it in an equivalent quantum circuit (QF-Circ), an implementation of network optimization (QF-FB) and finally a tool to map the circuit on IBM quantum devices (QF-Map). Results from small classification tasks are provided, together with illustration of the various quantum circuits.

While the authors' effort to integrate the various components is surely remarkable, QuantumFlow has a fundamental flaw: there is no hint of any quantum advantage. In fact, the "quantum-friendly" network is simply a network of stochastic neurons: the output statistics can be reproduced using qubits, but there is no advantage in doing so.

The proposed construction shows this by suggesting two equivalent approaches in which intermediate qubits/neurons are measured and their expectation value used as a classical variable substituting the effect of the corresponding quantum circuit. At each stage the information is substantially classical.

When joining ML and QC, it is important to ask: What is the quantum advantage?

- QF-FB(C) is used as reference to evaluate QF-FB(Q) and in fact performs in general better even of the ideal quantum implementation. The classical implementation is much faster (see table 2) and the authors provide no evidence of a better scaling for the quantum version compared to the classical one.
- The quantum version seems to require a large number of gates when the NC gates are decomposed in 1- and 2-qubit gates. One has m NC gates, each controlled on m qubits (even including extra $m-2$ ancillas, one needs $O(m)$ 2-qubit gates per NC gate). Therefore, one needs at least $O(m^2)$ gates to update a single neuron. Classically the sum in eq.(4) also takes $O(m^2)$ operations if performed exhaustively, but could be done in $O(m)$ by grouping the terms with index 'i' and 'j' separately. In addition, eq.(4) provides the expectation value while quantum outputs needs to be averaged to estimate the expectation value.
- From the QF-Circ construction it is unclear whether the same input qubits can be used to update multiple neurons in the first hidden layer. Is the probability of observing $|1\rangle$ in input qubit 0 still p_0 at the end of circuit in Fig.8(a)? If not, then one needs fresh input qubits for each hidden neuron. I believe this is the case by looking at Fig.9(c).
- Referring to Fig.9(c), it is unclear to me why the outputs of the first hidden layer (qubits O2 and O3) can be used to update both neurons in the second hidden layer. Despite the measurement of O4 is delayed, it will still influence that of O5. The authors argue in the Methods:QF-Map section that "there is no such independent requirement for the output of the second layer and therefore 2 NC components use the same inputs", but the argument is unclear since the statistics of outcomes O4 and O5 would not be as intended (e.g. that from Fig.7(b) applied to the neurons in the second hidden layer).

There are a few additional points that the authors should consider:

1. The results in Fig.2 refer to MNIST. Are the authors using the 28x28 input with grey levels? Or are they simplifying the dataset? All methods apart from QF have binary weights: is it possible that their performance is negatively affected by this constraint?
2. The experiments reported in Fig.5 are hardly conclusive. First of all, the classification function is very simple and can be intuitively seen that input (x,y) should be classified "1" when both x and y

are small or large. It seems easy for a quadratic activation function do achieve it. Second, batch normalization helps by providing 2 variational parameters in addition to the two networks weights.

3. The classification function is symmetric in the input exchange $x \leftrightarrow y$, but the network in Fig.5(b) is not. Any thoughts?

4. On QF-MAP: what part of the transformation in Fig.4 are automatic with QF-Map? Can it derive circuit (d) from (a)? From the derivation in the SI, it seems that the derivation is based on a visual analysis of all $2^3=8$ amplitudes and done "by-hand". Is this approach scalable, for example when $m>40$?

5. If the design in Fig.4(d) was obtained "by hand", what feature of QF-Map distinguish it from a regular hardware-aware mapper required to run any quantum circuit?

6. Referring to the batch normalization: does the decision of normalizing the probability mean of a batch of outputs to 0.5 implies that intermediate neurons have activation values concentrated around 0.5? Does this negatively affect the number of samples required to estimate it (considering that Monte-Carlo error scales as $1/\sqrt{K}$ with K being the number of samples)?

Overall, I believe that the framework presented in this manuscript does not support the claim of integrating ML and QC in a meaningful way. In my opinion, this requires the identification of a clear path to achieve quantum advantage. Path that is completely neglected in this work. I cannot recommend publication in Nature Communications.

Reviewer #2:

Remarks to the Author:

In this manuscript, the authors propose a novel framework for training machine learning models on quantum circuits and demonstrate superior experimental results to current State of the Art in quantum machine learning. I see the main innovation of this work in an effective design of the mapping function that translates the machine learning network to a quantum circuit, and the first existing implementation of batch normalization operation.

Overall, my impression is that the paper is well-written, experimental design and theoretical formulations are appropriate and sound. However, I am not entirely convinced that the claim of "co-design" in the title is justified. The authors present QF-Net as a quantum friendly network, but it seems that the quantum circuit QF-Circ is constructed specially to reproduce the behavior of the given QF-Net network, therefore, the term "co-design" might not be appropriate. There should be some evidence of co-design between QF-Net and QF-Circ. I would like to ask the authors to address this issue before being considered for possible publication.

Below I list some comments that require revision:

1) In Introduction, the caption to Figure 1(b) is hard to understand

2) In Results - QF-FB(C) and QF-FB(Q) are consistent, the second paragraph states that: "We can see that there is a small difference between QF-FB(C) and QF-FB(Q)". However, it is questionable that Accuracy values in Table 1 can be considered small.

Further questions arise about the experiments:

How many times the experiments were repeated?

What was the variance?

Why the datasets were matched in such way?

3) In the third paragraph, does "This verifies that QF-FB can provide..." refer to QF-FB(C) or (QF-FB(Q)?

4) In QF-Circ and QF-Map on IBM Quantum Processor, the definition of "deviation" mentioned in the third paragraph and in Figure 4, should be made clear.

5) In Figure 4, subplots (a)-(d) present different designs to explore relations between CNOT numbers and the error rates. It is clear that design in (d) is superior, and it seems unnecessary to present the results from all of the designs in the Figure 4(e). We feel it is sufficient to show only (a) and (d) for comparison, to present the effect of CNOT numbers more clearly.

6) In Figure 4 and other places, component x of Pauli matrices (σ_x) is denoted as X, but denoting the rotation of y, R_y as Y is confusing.

7) In Discussion, the first paragraph, the sentence “..the QF-Circ can automatically generate and optimize a corresponding quantum circuit” implies that QF-Circ is a function that generates quantum circuit for QF-Net. On the other hand, in the Introduction, second paragraph from the end (“..QF-Map is an automatic tool to conduct (1) network-to-circuit mapping (from QF-Net to QF-Circ)..”), it appears that QF-Circ is generated from QF-Net by the QF-Map. To avoid confusion, it should be clearly distinguished throughout the paper, what is the function, and what is the product generated by the function.

8) In Methods - QF-Net, second paragraph, iv) “ E: this operation converts the random variable y 2 to 0-1 real number by taking its expectation” – is the operation $R \rightarrow C \rightarrow A$ repeated multiple times to obtain a mean value?

9) In QF-Map, figure 9(b), the random values generated from the inputs from I_k ($k_0, \dots, 3$) are fed into two C/A. Are these random values different for each C/A input? If so, there is no problem, but if these are the same random variables, it is strange that they would output different results, as presented in Figure 9(b) and Figure 9(c).

10) In Supplementary Information, 1.3 CNOT gate on two initialized qubits, “...corresponding to states $[|00\rangle + |01\rangle + |10\rangle + |11\rangle]^T$ – such notation may lead to misunderstanding. Should instead be written as $A|00\rangle + B|01\rangle + C|10\rangle + D|11\rangle$ ”

11) Lastly, the manuscript would benefit from proofreading, as there are some minor language mistakes, for example in the abstract: “In this article we present... .. to fixed the missing link”.

Reviewer #3:

Remarks to the Author:

This manuscript presents a novel framework for machine learning architecture for circuit model quantum computing. Authors consider a set of transformations native to quantum dynamics to introduce nonlinearity required for machine learning tasks. The proposed ML model is then mapped directly to a quantum circuit through a set of developed tools. Some proof-of-concept experiments have been tested on IBM 5-qubit machines. Overall, I found the work an interesting contribution to the active research area of QML. I have few questions before recommending for publication:

1. Authors refer to batch normalization as a motivation for the introduced framework. My question is that why is batch normalization needed for QML circuit models? In neural network literature BN was introduced to alleviate the covariate shift problem which was due to the nonlinearity of neurons. Why do you think similar problem would arise in the quantum domain?

2. It would be nice to discuss possibility of extension of your model to higher orders of non-linearity beyond the quadratic terms. How does circuit complexity increase?

3. The first stage of the model where you map data into quantum random variables in a trick similar the one introduced in the IBM paper on SVMs. You should cite that paper: V Havlicek,

"Supervised learning with quantum-enhanced feature spaces", Nature. vol. 567, pp. 209-212 (2019).

Thanks,
Alireza Shabani

Response to Referees

***Original Title:* Can Quantum Computers Learn Like Classical Computers? A Co-Design Framework for Machine Learning and Quantum Circuits**

***Modified Title:* A Co-Design Framework of Machine Learning and Quantum Circuits Towards Quantum Advantage**

***Type:* Revision**

We would like to express our sincere thanks to the reviewers for their valuable time and constructive comments to enable this paper to become a top-quality paper. We are glad to report that this revision has complied with all the review comments.

In this report, we have listed the detailed actions for each comment. The review comments are shown in *italic*, our detailed responses are shown in **bold and upright**, and the modifications in the revised paper are placed within rectangles.

Outline:

- Summary of changes (Pages 2-3)
- Actions for the comments from the 1st reviewer (Pages 4-21)
- Actions for the comments from the 2nd reviewer (Pages 22-32)
- Actions for the comments from the 3rd reviewer (Pages 33-35)

Summary of Changes:

- We have demonstrated that Quantum Advantages can be achieved via the co-design framework:*
 - (Hardware-Aware Neural Network Design) In “QF-Net and QF-Net+” in the Methods Section, first, we have proposed a new quantum-aware neural computation layer, namely U-LYR, **such that 2^N data can be expressed and operated on N qbits simultaneously**. Compared to the probabilistic model-based neural computation using 2^N qbits to encode data, U-LYR using N qbits provides the potential to achieve quantum advantage. Second, we have integrated the proposed two neural computations in a hybrid neural network, called QF-Net+, **such that the entire network can be executed on the quantum circuit without measurement**.
 - (Software-Aware Quantum Circuit Design) In “QF-Map” in the Methods Section, we have devised a novel mapping algorithm for U-LYR. Even using N qbits to operate 2^N data, state-of-the-art hypergraph state approach-based mapping generates the quantum circuit with cost complexity of $O(2^N)$ in terms of the basic quantum logic gates (e.g., Hadamard, CNOT, Toffoli). We observed that the order of inputs in machine learning algorithm can be changed, and in turn, the positions of weights can be changed. **Based on this property, we have developed the novel mapping algorithm to guarantee the cost complexity to be $O(N^2)$** . Considering that the classical computer requires $O(2^N)$ basic computing units, where each unit takes 1 clock cycle, it implies that we can achieve quantum advantage for neural computation layer, U-LYR.
- We have added or modified five sets of experimental results in the revision, including:*
 - In “QF-Nets Achieve High Accuracy on MNIST” in the Results Section, we have added 6 more sub-datasets in MNIST with a number of classes ranging from 2 to 5 in Figure 2. **Results demonstrate that the proposed QF-Nets can consistently achieve state-of-the-art accuracy in comparison with the existing neural networks on datasets with different scales**.
 - In “QF-FB(C) and QF-FB(Q) are Consistent” in the Results Section, we have added new experimental results of simulating neural networks on classical computers and quantum computers in Table 1. Compared with the original QF-Net whose accuracy loss on QF-FB(Q) can reach up to 5.92%, **the accuracy difference between QF-FB(C) and QF-FB(Q) for QF-Net+ is merely 0.81%**. Furthermore, quantum simulation QF-FB(Q) can even achieve 0.54% accuracy gain compared with QF-FB(C).
 - In “QF-Map is the Key to Achieve Quantum Advantage” in the Results Section, we have added two set of new experiments on a comparison of the number of basic gates used in neural computation (Figure 4) and in the neural network for MNIST dataset (Table 2). From the first set of results, it is clear to see that U-LYR can achieve quantum advantage. With the increasing input sizes from 16 to 2048, the cost reduction of U-LYR over classical **computer grows from 2.4 times to 64 times**. Furthermore, benefiting from quantum advantages achieved by U-LYR, the second set of results show that the proposed hybrid network for quantum computing can

also achieve **more than 10 times cost reduction** compared with that on classical computer. As far as we knew, this is the first work to demonstrate quantum advantages for machine learning.

- In QF-Circ on IBM Quantum Processor Section, we have followed the reviewers' comments to simplify the results in Figure 5. The new results on the original design and the optimized design clearly demonstrate the importance of conducting circuit optimization.

3. *We have improved the paper according to the review suggestions. The major improvements include:*

- In the Introduction Section, we have made the following five modifications: (1) we have discussed the emerging problem on heavy computation load of AI in specific field; (2) we have simplified Figure 1 to make it more concise; (3) we have added new references to show the importance of using continuous values for inputs; (4) we have demonstrated the co-design philosophy; and (4) we have added the discussion of QF-Net+ toward quantum advantage.
- In the Discussion Section, we have added Table 3 to make a comparison with cost complexity and flexibility among (1) the classical computing platform, (2) state-of-the-art quantum implementation of machine learning algorithms, and (3) two proposed neural computation layers (i.e., P-LYR and U-LYR) in this work. From the comparison, it is clear to see that P-LYR provides better flexibility but higher cost, while U-LYR can achieve cost reduction from $O(2^k)$ to $O(k^2)$, achieving quantum advantage but has limitations on binary representation for weights and difficult to connect multi-layer without the help of P-LYR.
- In the Methods Section, for all components (QF-Nets, QF-FB, QF-Circ, QF-Map), we have discussed how to support new neural computation U-LYR. In QF-Nets, we have further discussed how to implement a network with multiple layers. In QF-Map, we have formulated the problem of mapping neural computation to QF-Circ, and proved that the proposed algorithm can guarantee the cost complexity of U-LYR with input size of 2^k to be $O(k^2)$.

4. *We have added the following eight new references:*

- [15] Bernard, O. et al. Deep learning techniques for automatic MRI cardiac multi-structures segmentation and diagnosis: is the problem solved? IEEE transactions on medical imaging 37, 2514–2525 (2018).
- [16] Bonaldi, A. & Braun, R. Square kilometre array science data challenge 1. arXiv preprint arXiv:1811.10454(2018).
- [17] Lukic, V., de Gasperin, F. & Brüggen, M. ConvoSource: Radio-Astronomical Source-Finding with Convolutional Neural Networks. Galaxies 8, 3 (2020).
- [31] Havlicek, V. et al. Supervised learning with quantum-enhanced feature spaces. Nature 567, 209–212 (2019).
- [34] Courbariaux, M., Bengio, Y. & David, J.-P. Binaryconnect: Training deep neural networks with binary weights during propagations. In Advances in neural information processing systems, 3123–3131 (2015)
- [36] Google. TensorFlow Quantum. <https://www.tensorflow.org/quantum/tutorials/mnist> (2020). Accessed: 08-18-20
- [37] Rosenblatt, F. The perceptron, a perceiving and recognizing automaton Project Para (Cornell Aeronautical Laboratory, 1957).
- [39] Lvovsky, A. I., Sanders, B. C. & Tittel, W. Optical quantum memory. Nat. photonics 3, 706–714 (2009).

5. *We have made **demons using quirk with 16 qbits** (i.e., maximum supported by quirk) for the proposed QF-Net+ for {3,6} sub-dataset of MNIST dataset, which can achieve accuracy of 95.73%, which can be accessed via the following link: <http://wjiang.nd.edu/categories/QF>.*

Comments by the 1st Reviewer:

The manuscript “Can Quantum Computers Learn Like Classical Computers? A Co-Design Framework for Machine Learning and Quantum Circuits” has the ambitious goal of creating a framework to unify classical machine learning and quantum circuits. The authors introduce QuantumFlow (QF) and its various parts: a network based on stochastic neurons (QF-Net), a tool to convert it in an equivalent quantum circuit (QF-Circ), an implementation of network optimization (QF-FB) and finally a tool to map the circuit on IBM quantum devices (QF-Map). Results from small classification tasks are provided, together with illustration of the various quantum circuits.

[Response]

We appreciate your valuable time to give us helpful comments. In the revision, we have followed all your suggestions and taken actions on every comment. Specifically, your comments motivate us to add new components in the co-design framework to make it completed. Thanks so much.

[Comment 1] While the authors’ effort to integrate the various components is surely remarkable, QuantumFlow has a fundamental flaw: there is no hint of any quantum advantage. In fact, the “quantum-friendly” network is simply a network of stochastic neurons: the output statistics can be reproduced using qubits, but there is no advantage in doing so. The proposed construction shows this by suggesting two equivalent approaches in which intermediate qubits/neurons are measured and their expectation value used as a classical variable substituting the effect of the corresponding quantum circuit. At each stage the information is substantially classical.

When joining ML and QC, it is important to ask: What is the quantum advantage?

[Action]

Thanks for the helpful comments. We fully agree that demonstrating quantum advantage is one of the most important motivations for joining machine learning and quantum computing, which lacks discussion in the original manuscript. In this revision, we have re-investigated the co-design procedure to demonstrate quantum advantage. Specifically, we have added a new neural computation layer (U-LYR) into QuantumFlow, coupled with a novel algorithm in QF-Map.

For a neural computation with 2^k input data, we have demonstrated that U-LYR and QF-Map can work together to successfully reduce the number of basic gates used in the quantum circuit to $O(k^2)$, which is $O(2^k)$ for classical computing and the existing quantum computing implementations. This demonstrate that QuantumFlow can achieve quantum advantages.

In the Results Section of the revision, experimental results further verify this. Compared with the classical computing platforms, for input sizes of neural computations range from 16 to 2048, the cost reduction achieved by U-LYR increased from 2.4× to 64×. Furthermore, the outputs of the new proposed component U-LYR can seamlessly be connected to the previous proposed

probabilistic neural computation layer (P-LYR) to support a multi-layer neural network, namely QF-Net+. We demonstrate that QF-Net+ on a MNIST sub-dataset with 5 classes can achieve accuracy of 94.09%, while the cost reduction against the classical computer is 10.85×.

On Page 1, in Abstract:

We discover that, in order to make full use of the strength of quantum representation, it is best to represent data in a neural network as random variables or numbers in unitary matrices, such that they can be directly operated by the basic quantum logical gates. Based on these data representations, we propose two quantum friendly neural networks, QF-Net and QF-Net+ in QuantumFlow. QF-Net using the probabilistic model has better flexibility, which can seamlessly connect two layers without measurement but requires more qubits and logical gates. On the other hand, the unitary matrix applied in QF-Net+ can help to encode 2^k data into k qubits, and a novel algorithm can guarantee the cost complexity (i.e., logical gates) to be $O(k^2)$. Compared to the cost of $O(2^k)$ in classical computing and the existing quantum implementations, QF-Net+ demonstrates the quantum advantages. ... Results further show that for input sizes of neural computation grow from 16 to 2048, the cost reduction of QuantumFlow increased from 2.4× to 64×. Furthermore, on MNIST dataset, QF-Net+ can achieve accuracy of 94.09%, while the cost reduction against the classical computer reaches 10.85×.

2nd Paragraph on Right-Hand Page 2, in Introduction Section:

Towards achieving the quantum advantage, we propose QF-Net+ and a novel neural computation (denoted as U-LYR). U-LYR is based on the unitary matrix, where 2^k input data are converted to a vector in the unitary matrix, such that all data can be represented by the amplitudes of states in a quantum circuit with k qubits. The reduction in input qubits provides the possibility to achieve quantum advantage; however, the state-of-the-art implementation²⁴ using hypergraph state for computation still has the cost complexity of $O(2^k)$. In this work, we devise a novel optimization algorithm to guarantee the cost complexity of U-LYR to be $O(k^2)$, which takes full use of the properties of machine learning models and quantum logic gates. Compared with the complexity of $O(2^k)$ on classical computing platforms, U-LYR demonstrates the quantum advantages of executing machine learning operations.

3rd Paragraph on Page 5, in Results Section:

Figure 4 reports the comparison results for the core component in neural network, the neural computation layer. The x-axis represents the input size of the neural computation, and they-axis stands for the cost, that is, the number of operators used in the corresponding design. For quantum implementation (both FC(Q)²⁶ in FFNN(Q)²⁷ and U-LYR in QuantumFlow), the value of weights will affect the gate usage, so we generate 50 sets of weights for each scale of input, and the dots on the lines in this figure represent average cost. From this figure, it clearly shows that the cost of FC(C)

Figure 4. Demonstration of Quantum Advantage Achieved by U-LYR in QuantumFlow: comparison is conducted by using 50 random generated weights for each input size.

in MLP(C) on classical computing platforms grows exponentially along with the increase of inputs. The state-of-the-art quantum implementation FC(Q) has the similar exponentially growing trend. On the other hand, we can see that the growing trend of U-LYR is much slower. As a result, the cost reduction continuously increases along with the growing of input size of neural computation. For input size of 16 and 32, the average cost reductions are 2.4× and 3.3×, compared with the implementations on classical computers. When the input size grows to 2,048, the cost reduction increased to 64× on average. The cost reduction trends in this figure clearly demonstrate the quantum advantage achieved by U-LYR. In the Methods section, for the neural computation with an input size of 2k, we will show that the complexity for quantum implementation is $O(k^2)$, while it is $O(2^k)$ for classical computers.

Table 2. QuantumFlow demonstrates quantum advantages on neural networks for MNIST datasets with the increasing model sizes: comparison on the number of used gates.

Dataset	Structure		MLP(C)			FFNN(Q)			QF-Net+(Q)				
	In	L1 L2	L1	L2	Tot.	L1	L2	Tot.	Red.	L1	L2	Tot.	Red.
{1,5}	16	4 2				80	38	118	1.27×	74	38	112	1.34×
{3,6}	16	4 2				96	38	134	1.12×	58	38	96	1.56×
{3,8}	16	4 2	132	18	150	76	34	110	1.36×	58	34	92	1.63×
{3,9}	16	4 2				98	42	140	1.07×	68	42	110	1.36×
{0,3,6}	16	8 3	264	51	315	173	175	348	0.91×	106	175	281	1.12×
{1,3,6}	16	8 3				209	161	370	0.85×	139	161	300	1.05×
{0,3,6,9}	64	16 4	2064	132	2196	1893	572	2465	0.89×	434	572	1006	2.18×
{0,1,3,6,9}	64	16 5	2064	165	2229	1809	645	2454	0.91×	437	645	1082	2.06×
{0,1,2,3,4}	64	16 5				1677	669	2346	0.95×	445	669	1114	2.00×
{0,1,3,6,9}*	256	8 5	4104	85	4189	5030	251	5281	0.79×	135	251	386	10.85×

*: Model with 16×16 resolution input for dataset {0,1,3,6,9} to test scalability, whose accuracy is 94.09%, which is higher than 8×8 input with accuracy of 92.62%.

Table 2 reports the comparison results for the whole network. The neural network models for MNIST in Figure 2 are deployed to quantum circuits to get the cost. ...

From the table, it is clear to see that all cases implemented by QF-Net+ can achieve cost reduction over MLP(C), while for datasets with more than 3 classes, FFNN(Q) needs more gates than MLP(C). A further observation made in the results is that QF-Net+ can achieve higher cost reduction with the increasing of input size. Specifically, for input size is 16, the reduction ranges from 1.05× to 1.63×. The reduction increases to 2.18× for input size is 64, and it continuously increases to 10.85× when the input size grows to 256. The above results are consistent to the results shown in Figure 4. It further indicates that even the second layer in QF-Net+ uses the P-LYR which requires more gates for implementation, the quantum advantage can still be achieved for the whole network because the first layer using U-LYR can significantly reduce the number of gates. Above all, QuantumFlow demonstrates the quantum advantages on MNIST dataset.

[Comment 2] QF-FB(C) is used as reference to evaluate QF-FB(Q) and in fact performs in general better even of the ideal quantum implementation. The classical implementation is much faster (see table 2) and the authors provide no evidence of a better scaling for the quantum version compared to the classical one.

[Action]

Thanks for the comment. In the Results Section of the revision, we have added two sets of experiments to show the better scaling for the quantum implementation over the classical one.

First, we have clarified that the QF-FB(Q) is based on the simulation on classical computers, that is, IBM Qiskit Aer. We also show that the accuracy loss for QF-Net+ on a quantum computer is within 0.81% against that on classical computers, and for some cases, quantum implementation can even achieve higher accuracy against classical implementation.

Last Paragraph on Left-Hand Page 4, in Results Section:

Table 1. Inference accuracy and efficiency comparison between QF-FB(C) and QF-FB(Q) on both QF-Net and QF-Net+ using MNIST dataset to show the consistency of implementations of QF-Nets on classical computers and quantum computers.

dataset	QF-Net Qbits (Neurons)		Accuracy			Elapsed CPU Time		QF-Net+ Qbits (Neurons)		Accuracy			Elapsed CPU Time	
	L1	L2	QF-FB(C)	QF-FB(Q)	Diff.	QF-FB(C)	QF-FB(Q)	L1	L2	QF-FB(C)	QF-FB(Q)	Diff.	QF-FB(C)	QF-FB(Q)
{3,6}	28(4)	12(2)	97.10%	95.53%	-1.57%	5.13S	2,555H	7(4)	5(2)	98.27%	97.46%	-0.81%	4.30S	16.57H
{3,8}	28(4)	12(2)	86.84%	83.59%	-3.25%	5.59S	2,631H	7(4)	5(2)	87.40%	88.06%	+0.54%	4.05S	16.56H
{1,3,6}	28(8)	18(3)	87.91%	81.99%	-5.92%	15.89S	14,650H	7(8)	8(3)	88.53%	88.14%	-0.39%	6.96S	47.98H

Column “Accuracy” in Table 1 reports the accuracy comparison. For QF-FB(C), there will be no difference in accuracy among different executions. For QF-FB(Q), we implement the obtained QF-

Circ from QF-Nets on Qiskit Aer simulation with 8,192 shots. We have the following two observations from these results: (1) There exist accuracy differences between QF-FB(C) and QF-FB(Q). This is because Qiskit Aer simulation used in QF-FB(Q) is based on the Monte Carlo method, leading to the variation. In addition, since the output probability of different neurons may quite close in some cases, it will easily result in different classification results for small variations. (2) Such accuracy differences for QF-Net+ is much less than that of QF-Net, because QF-Net utilizes much more qubits, which leads to the accumulation of errors. In QF-Net+, we can see that there is a small difference between QF-FB(C) and QF-FB(Q). For the dataset {3,8}, QF-FB(Q) can even achieve higher accuracy. The above results demonstrate both QF-Net and QF-Net+ can be consistently implemented on classical and quantum computers.

Second, in the **Results Section** of the revision, we compare the basic operators used in the key computation component: neural computation, on both classical computer (i.e., adder and multiplier) and quantum computer (e.g., Pauli-X, Toffoli). With the help of QF-Map, in the new results, we observe that QuantumFlow can achieve 2X operator reduction for the network with 16 neurons and over 4X operator reduction for that with 64 neurons. In the **Methods Section** of the revision, we further demonstrate that QF-Map can reduce the complexity of neural computation from $O(2^k)$ to $O(k^2)$, where N is the number of input neurons in a layer.

2nd Paragraph on Right-Hand Page 5, in Results Section:

Table 2. QuantumFlow demonstrates quantum advantages on neural networks for MNIST datasets with the increasing model sizes: comparison on the number of used gates.

Dataset	Structure			MLP(C)			FFNN(Q)			QF-Net+(Q)				
	In	L1	L2	L1	L2	Tot.	L1	L2	Tot.	Red.	L1	L2	Tot.	Red.
{1,5}	16	4	2				80	38	118	1.27×	74	38	112	1.34×
{3,6}	16	4	2				96	38	134	1.12×	58	38	96	1.56×
{3,8}	16	4	2	132	18	150	76	34	110	1.36×	58	34	92	1.63×
{3,9}	16	4	2				98	42	140	1.07×	68	42	110	1.36×
{0,3,6}	16	8	3				173	175	348	0.91×	106	175	281	1.12×
{1,3,6}	16	8	3	264	51	315	209	161	370	0.85×	139	161	300	1.05×
{0,3,6,9}	64	16	4	2064	132	2196	1893	572	2465	0.89×	434	572	1006	2.18×
{0,1,3,6,9}	64	16	5				1809	645	2454	0.91×	437	645	1082	2.06×
{0,1,2,3,4}	64	16	5	2064	165	2229	1677	669	2346	0.95×	445	669	1114	2.00×
{0,1,3,6,9}*	256	8	5	4104	85	4189	5030	251	5281	0.79×	135	251	386	10.85×

*: Model with 16×16 resolution input for dataset {0,1,3,6,9} to test scalability, whose accuracy is 94.09%, which is higher than 8×8 input with accuracy of 92.62%.

Table 2 reports the comparison results for the whole network. The neural network models for MNIST in Figure 2 are deployed to quantum circuits to get the cost. ...

From the table, it is clear to see that all cases implemented by QF-Net+ can achieve cost reduction over MLP(C), while for datasets with more than 3 classes, FFNN(Q) needs more gates than MLP(C). A further observation made in the results is that QF-Net+ can achieve higher cost reduction with the increasing of input size. Specifically, for input size is 16, the reduction ranges from 1.05× to 1.63×. The reduction increases to 2.18× for input size is 64, and it continuously increases to 10.85× when the input size grows to 256. The above results are consistent to the results shown in Figure 4. It further indicates that even the second layer in QF-Net+ uses the P-LYR which requires more gates for implementation, the quantum advantage can still be achieved for the whole network because the first layer using U-LYR can significantly reduce the number of gates. Above all, QuantumFlow demonstrates the quantum advantages on MNIST dataset.

Last Paragraph on Left-Hand Page 13, in Methods Section:

Algorithm 4: QF-Map: weight mapping algorithm

```

Input: (1) An integer  $R \in (0, 2^{k-1}]$ ; (2) number of qubits  $k$ ;
Output: A set of applied gate  $G$ 
void recursive( $G, R, k$ ){
    if ( $R < 2^{k-2}$ ){
        recursive( $G, R, k - 1$ ); // Case 1 in the third step
    }
    else if ( $R == 2^{k-1}$ ){
         $G.append(PG_{2^{k-1}})$ ; // Case 2 in the third step
        return;
    }else{
         $G.append(PG_{2^{k-1}})$ ;
        recursive( $G, 2^{k-1} - R, k - 1$ ); // Case 3 in the third step
    }
}
// Entry of weight mapping algorithm
set main( $R, k$ ){
    Initialize empty set  $G$ ;
    recursive( $G, R, k$ );
    return  $G$ 
}

```

In the above algorithm, the worst case for the cost is that we apply all gates in $\{PG_{x(j)} \mid x(j) = 2j-1 \ \& \ j \in [1, k]\}$. Let the state $x(j)$ has $y \mid 1\rangle$ states, if $y > 2$ the $PG_{x(j)}$ can be implemented using $2y-1$ basic gates, including $y-1$ Toffoli gates for controlling, 1 control Z gate, and $y-1$ Toffoli gates for resetting; otherwise, it uses 1 basic gates. Based on these understandings, we can calculate the cost complexity in the worst case, which is $1+1+3+\dots+(2 \times k-1) = k^2+1$. Therefore, the cost complexity of linear function computation is $O(k^2)$. The quadratic activation function is implemented by a C^kZ gate, whose cost is $O(k)$. Thus, the cost complexity for neural computation U-LYR is $O(k^2)$.

[Comment 3] The quantum version seems to require a large number of gates when the NC gates are decomposed in 1- and 2-qubit gates. One has m NC gates, each controlled on m qubits (even including extra $m-2$ ancillas, one needs $O(m)$ 2-qubit gates per NC gate). Therefore, one needs at least $O(m^2)$

gates to update a single neuron. Classically the sum in eq.(4) also takes $O(m^2)$ operations if performed exhaustively, but could be done in $O(m)$ by grouping the terms with index 'i' and 'j' separately.

[Action]

Thanks for the comment. In the Discussion Section of the revision, we have added discussions on the complexity in qbits and cost for the proposed designs. Specifically, for the original probabilistic design, namely P-LYR, it has the most flexibility for data representation and supporting different non-linear functions; however, as pointed out by this comment, for a neural computation with $m=2^k$ inputs and 1 output, the required number of qbits is $O(2^k)$ and the number of basic gates are $O(k*2^k)$. Therefore, P-LYR is suitable for small scale neural computation. On the other hand, the newly added neural computation layer, namely U-LYR, is designed for larger-scale neural computation. For the neural computation with $m=2^k$ inputs and 1 output, it only requires $O(k)$ qbits and $O(k^2)$ basic gates. For the comparison, the classical computer requires $O(2^k)$ operations. Therefore, it demonstrates that U-LYR achieves quantum advantages. Taking the advantages of each design, we construct a hybrid neural network, called QC-Net+, where the first layer is based on U-LYR and later layers are based on P-LYR.

Last Paragraph on Left-Hand Page 7, in Discussion Section:

Table 3. Comparison of the implementation of Neural Computation with $m = 2^k$ input neurons.

Metrics		MLP(C)	FFNN(Q)	QF-NC _p	QF-NC _u
Complexity	# Bits/Qbits	$O(2^k)$	$O(k)$	$O(2^k)$	$O(k)$
	# Operators	$O(2^k)$	$O(2^k)$	$O(k \cdot 2^k)$	$O(k^2)$
Data Representation	Input Data	F32	Bin	R.V.	F32
	Weights	Bin (F32)	Bin	Bin (R.V.)	Bin
Connect Layers w/o Measurement		✓	-	✓	×
Summary	Flexibility	-	×	✓	×
	Qu. Adv.	-	×	×	✓

Neural computation layer is one key component in QuantumFlow to achieve state-of-the-art accuracy and quantum advantage. We have shown in Figure 2 that the existing quantum-aware neural network²⁵ that interprets inputs as the binary form will degrade the net-work accuracy. To address this problem, in QF-Net, we first propose a probability-based neural computation layer, denoted as P-LYR, which interprets real number inputs as random variables following a two-point distribution. As shown in Table 3, P-LYR can represent both input and weight data using random variables, and it can directly connect layers without measurement. In summary, P-LYR provides better flexibility to perform neural computation than others; however, it suffers high complexity, i.e., $O(2^k)$ for the usage of qbits and $O(k*2^k)$ for the usage of operators (basic quantum gates).

In order to acquire quantum advantages, we further propose a unitary matrix based neural computation layer, called U-LYR. As illustrated in Table 3, U-LYR sacrifices some degree of flexibility on data representation and non-linear function but can significantly reduce the circuit complexity. Specifically, with the help of QF-Map, the number of basic operators used by U-LYR can be reduced from $O(2^k)$ to $O(k^2)$, compared to MLP(C) and FFNN(Q). Kindly note that this work does not take the cost of inputs encoding into consideration in demonstrating quantum advantage; instead, we focus on the speedup of the commonly used computation component layer, that is, the neural computation layer. The cost of encoding inputs can be reduced to $O(1)$ by preprocessing data and storing them into quantum memory, or approximating the quantum states by using basic gate (e.g., Ry). For neural computation, we demonstrated that U-LYR can successfully achieve quantum advantage in the next section.

[Comment 4] In addition, eq.(4) provides the expectation value while quantum outputs needs to be averaged to estimate the expectation value.

[Action]

Thank you for the comment. In the Method Section of the revision, we have clarified that in the quantum circuit design, there is no need to measure the quantum outputs after each layer. As stated in the paper, the neural network, QF-Net, has four operations R, C, A, E. We can sequentially execute these operations in each layer, but measuring the expectation of the i^{th} layer and converting the expectation value to the random variable in the $i+1^{\text{th}}$ are not necessary. Specifically, we convert the data using R in the first layer; then, repeat C, A in the following layer; finally, we conduct E operation in the last layer to extract the expectation value.

1st Paragraph Right-Hand on Page 9, in Methods Section:

Multiple Layers: P-LYR and U-LYR are the fundamental components in QF-Nets, which may have multiple layers. In terms of how a network is composed using these two components, we present two kinds of neural networks: QF-Net and QF-Net+. QF-Net is composed of multiple layers of P-LYR. For the quantum implementation, operations on random variable can be directly operated on qbits. Therefore, **R** operation is only conducted in the first layer. Then, **C** and **A** operations will be repeated without measurement. Finally, at the last layer, we measure the probability for output qbits, which is corresponding to the **E** operation. On the other hand, QF-Net+ is composed of both U-LYR and P-LYR, where the first layer applies U-LYR with the converted inputs. The output of U-LYR is directly represented by the probability form on a qbit, and it can seamlessly connect to C in P-LYR used in later layers.

[Comment 5] (1) From the QF-Circ construction it is unclear whether the same input qubits can be used to update multiple neurons in the first hidden layer. Is the probability of observing $|1\rangle$ in input

qubit 0 still p_0 at the end of circuit in Fig.8(a)? If not, then one needs fresh input qubits for each hidden neuron. I believe this is the case by looking at Fig.9(c).

(2) Referring to Fig.9(c), it is unclear to me why the outputs of the first hidden layer (qubits O2 and O3) can be used to update both neurons in the second hidden layer. Despite the measurement of O4 is delayed, it will still influence that of O5. The authors argue in the Methods: QF-Map section that “there is no such independent requirement for the output of the second layer and therefore 2 NC components use the same inputs”, but the argument is unclear since the statistics of outcomes O4 and O5 would not be as intended (e.g. that from Fig.7(b) applied to the neurons in the second hidden layer).

[Action]

Thanks for the comment. For question (1), in **Section 2.1 of the supplementary information** of the revision, we have clarified that the probability of observing $|1\rangle$ in input qubit 0 will be maintained as p_0 at the end of the circuit in Fig.9(a) (original Fig. 8(a)). Intuitively, since all input qubits act as the controller, their probabilities will not be changed. Like a CNOT(I, E) gate where qbit I is $|1\rangle$ then qbit E flips, the probability of the control qbit I will not be changed. Formally, in our design, as shown in Figure 2 in the appendix, each group corresponds to one state in the space of all input qubits. As shown in the figure, from step S_0 to S_1 , it only “averages” amplitudes in each group. Therefore, it will not change the probability of input qubits. From S_1 to S_2 , it only changes the sign of amplitudes. This will also not change the probability since the probability is the square of amplitudes. In consequence, at state S_2 , the probability of all input qubits will not be changed. After S_2 , there are no more operations on input qubits. Thus, the probability of all input qubits will not be changed. As a result, if there is only one layer, we do not need to fresh input qubits for each hidden neuron.

On Page 3, in Section 2.1 in Supplementary Information:

Figure 2. Illustration of computation at a neuron in QF-Net with the square non-linear function: (a) translation to random variables and neural computation; (b)-(d) probability distributions of random variables x , y , and y^2 ; (e) quantum circuit implementation of neural computation; (f) amplitude of states at different time steps.

Kindly note that the probability of input qubits at step S_4 is exactly the same as that at step S_1 . This is because of all H gates applied to the encoding qubits, which will not change the sum of squares of

amplitude in a group (i.e., probability). From S1 to S2, it only changes the sign of amplitudes. This will also not change the probability since the probability is the sum of the square of amplitudes. In consequence, at state S2, the probability of all input qubits will not be changed. After S2, there are no more operations on input qubits, and therefore, the probability of every input qubit at S4 is the same with that at S1.

For question (2), according to the above discussion that the probability of input qubits will not be changed throughout the neural computation process, making it possible to use the same inputs (i.e., the outputs of layer 1, O_2 and O_3) to calculate multiple neurons in a layer (i.e., layer 2). However, since different neurons have different weights, we need to make sure the weighted inputs to be resume after the computation of one neuron. In our design, since the weight is implemented by the X gate and I (identity) gate, we can put the same weight component after the neural computation to resume the probability of all input qubits. We have revised the figure to make it clearer.

In addition, in Section 4 of the supplementary information, we have revised the independent statement to make it clear. Specifically, in QF-Net, operation R convert real numbers (inputs) to random variables, to support the efficient simulation on classical computers, we utilize Eq.(4) for calculation, which requires the inputs are independent. The requirement of independent inputs is not conflicting to that the calculations of neurons can share the same inputs. But using the shared inputs at the i^{th} layer will lead the outputs at the i^{th} layer to be dependent. There are two ways to solve this problem. (1) If we do not want any measurement during the processing, we need multiple inputs for each layer; however, if we are processing the last layer, there is no need to duplicate multiple inputs, since the outputs will no longer be used as inputs and therefore no independent requirement is needed. (2) If we allow the circuit to do measurement during the processing, we can use the same inputs at each layer with a measurement at the end of this layer.

On Page 7, in Section 4 in Supplementary Information:

QF-Net in Figure 5(a) is a neural network with 2 hidden layers, designed based on neural computation P-LYR and batch normalization N-LYR sub-components. In QF-Circ, we notice that the results are stored in state $|1\rangle$ of the output qubits for both P-LYR and N-LYR; while the initialization operation R (using Y gate) is to encode the previous results to state $|1\rangle$. As a result, P-LYR and N-LYR circuits in QF-Circ can directly take the output qubit from the previous circuit without measurement as input. As such, Figure 5(b) shows a corresponding network to QF-Circ, where internal data type conversions in Figure 5(a) are removed. Of course, alternatively, BN can still take a given real number (e.g., measured results from the previous circuit) as input, but it would need a Y gate to initialize qubit I. In this example, we demonstrate the circuits without internal measurement, and we remove the internal data type conversion in Figure 5(a) to obtain the network in Figure 5(b).

Figure 5. Network-to-circuit mapping in QF-Map: (a-c) QF-Net; (d-f) QF-Net+. (a) QF-Net with 2 hidden layers for classical computer where batch normalization is applied in the first layer; (b) the corresponding QF-Net without internal measurement (i.e., no data type conversion); (c) QF-Circ mapped from QF-Net in (b); (d) QF-Net+ with 2 hidden layers for classical computer where batch normalization is applied in the first layer; (e) the corresponding QF-Net without internal measurement (i.e., no data type conversion); (f) QF-Circ mapped from QF-Net+ in (e).

In QF-Circ, operation **R** converts real numbers (inputs) into random variables. To support the efficient simulation on classical computers, we calculate the expectation of the weighted sum of random variables, which requires the inputs are independent. The requirement of independent inputs is not conflicting to that the calculations of neurons in a layer can share the same inputs. However, using the shared inputs at the i^{th} layer will lead the outputs at the i^{th} layer to be dependent. If no measurement can be conducted between layers, we need to duplicate inputs (qubits) for each output neuron in a layer. There is one exception, that is, the last layer. Since the output of the last layer will no longer be used as inputs, there are no requirements of independence for the last layer, and therefore, the last layer can use the shared inputs. As shown in Figure 5(c), we utilize two independent sets of qubits ($\{I_0, \dots, I_3\}$ and $\{I_0', \dots, I_3'\}$) for computing two output neurons using two different weights (W_0 and W_1) in the first layer; while the output qubits of the first layer (i.e., O_2 and O_3) are shared for the calculation of different neurons in the second/last layer.

There are a few additional points that the authors should consider:

[Comment 7] The results in Fig.2 refer to MNIST. (1) Are the authors using the 28x28 input with grey levels? Or are they simplifying the dataset? (2) All methods apart from QF have binary weights: is it possible that their performance is negatively affected by this constraint?

[Action]

Thank you for these comments. For question (1), in **“QF-Nets Achieve High Accuracy on MNIST” of the Results Section** of the revision, we have added the experimental settings before introducing the results, in which we have clarified that the simplifications are made on the datasets. First, we extract different sub-datasets from MNIST to compose a testbed with 2-5 classes. For different sub-datasets, we made different simplifications on them. For the sub-datasets with 2-3 classes, we downsample the image with resolutions of 28*28 to 4*4, while for sub-datasets with 4-5 classes, the input images are downsampled to 8*8. All original images and downsampled images are with grey levels.

Last Paragraph On Left-Hand Page 3, in Results Section:

Before reporting the detailed results, we first discuss the experimental setting. In this experiment, we extract sub-datasets from MNIST, which originally include 10 classes. For instance, {3,6} indicates the sub-datasets with two classes (i.e., digits 3 and 6), which are commonly used in quantum machine learning (e.g., Tensorflow-Quantum³⁶). To evaluate the advantages of the proposed QF-Nets, we further include more complicated sub-datasets, {3,8}, {3,9}, {1,5} for two classes. In addition, we show that QF-Nets can work well on larger datasets, including {0,3,6} and {1,3,6} for three classes, and {0,3,6,9}, {0,1,3,6,9}, {0,1,2,3,4} for four and five classes. For the datasets with two or three classes, the original image is downsampled from the resolution of 28×28 to 4×4, while it is downsampled to 8×8 for datasets with four or five classes. All original images in MNIST and the downsampled images are with grey levels. For all involved datasets, we employ a two-layer neural network, where the first layer contains 4 neurons for two-class datasets, 8 neurons for three-class datasets, and 16 neurons for four- and five-class datasets. The second layer contains the same number of neurons as the number of classes in datasets.

References

36. Google. Tensorflow quantum. <https://www.tensorflow.org/quantum/tutorials/mnist> (2020). Accessed: 2020-08-18

For question (2), in the **“Introduction Section”** of the revision, we have clarified that QF-Net is taken binary values as weights, and we have also added a reference which demonstrates that applying binary weights can achieve high performance in deep neural network applications. For a fair comparison, all methods in the experiments employ the binary weights.

1st Paragraph on Right-Hand Page 2, in Introduction Section:

Kindly note that P-LYR can model both inputs and weight to be random variables. But because binary weights can achieve comparable high accuracy for deep neural network applications³⁴ and significantly reduce circuit complexity, we employ random variables for inputs only and binary values for weights in P-LYR.

References

34. Courbariaux, M., Bengio, Y. & David, J.-P. Binaryconnect: Training deep neural networks with binary weights during propagations. In Advances in neural information processing systems, 3123–3131 (2015).

[Comment 8] The experiments reported in Fig.5 are hardly conclusive. First of all, the classification function is very simple and can be intuitively seen that input (x,y) should be classified “1” when both x and y are small or large. It seems easy for a quadratic activation function do achieve it. Second, batch normalization helps by providing 2 variational parameters in addition to the two networks weights.

[Action]

Thank you for the comment. Experiments in Fig. 5 is not set to evaluate the performance of QF-Nets, instead, it validates the function of QF-Circ and virtual-to-physic mapping in QF-Map. Limited by the number of physical qubits and the high error rate for a larger sized quantum computer, we study a simple classification function. But we obtained two observations from the results: (1) the current quantum computer can be utilized for some simple task, providing high accuracy; (2) without the consideration of physic qubits, the classification results will be significantly affected.

To better evaluate the function of QF-Nets, in the Results Section of the revision, we have added 6 datasets from MNIST to demonstrate that the proposed QF-Nets can work on different scales of datasets.

2nd Paragraph on Right-Hand Page 3, in Results Section:

Figure 2. QF-Net+ achieves state-of-the-art accuracy in image classifications on different sub-datasets of MNIST.

From the results in Figure 2, we can see that the proposed “QF-Net+ w/ BN”(abbr. QF-Net+_BN) achieves the highest accuracy among all networks (even higher than MLP running on classical computers). Specifically, for the dataset of {3,6}, the accuracy of QF-Net+_BN is 98.27%, achieving 3.01% and 15.27% accuracy gain against MLP(C) and FFNN(Q), respectively. It even achieves a 1.17% accuracy gain compared to QF-Net_BN. An interesting observation attained from this result is that with the increasing number of classes in the dataset, QF-Net+_BN can maintain

the accuracy to be larger than 90%, while other competitors suffer an accuracy loss. Specifically, for dataset{0,3,6} (input resolution of 4×4), {0,3,6,9} (input resolution of 8×8), {0,1,3,6,9} (input resolution of 8×8), the accuracy of QF-Net+_BN are 90.40%, 93.63% and 92.62%; however, for MLP(C), these figures are 75.37%, 82.89%, and 70.19%. This is achieved by the hybrid use of two types of neural computation in QF-Net+ to better extract features in images. The above results validate that the proposed QF-Net+ has a great potential in solving machine learning problems and our co-design framework is effective to design a quantum network with high accuracy.

[Comment 9] The classification function is symmetric in the input exchange $x \leftrightarrow y$, but the network in Fig.5(b) is not. Any thoughts?

[Response]

Thank you for the comment. In fact, the network in Fig 5(b) is symmetric in terms of input x and y . Let's first see the function of neural computation without switch x and y , which can be represented as $N1 = [(+1 \times x) + (-1 \times y)]^2 = (x - y)^2$. Now, let's exchange x and y , using the same weights, and we can get $N2 = [(+1 \times y) + (-1 \times x)]^2 = (y - x)^2$. Obviously, we have $N1 = N2$. Thus, the network in Fig. 5(b) is symmetric, which is consistent with the classification function.

[Comment 10] On QF-MAP: what part of the transformation in Fig.4 are automatic with QF-Map? Can it derive circuit (d) from (a)? From the derivation in the SI, it seems that the derivation is based on a visual analysis of all $2^3=8$ amplitudes and done "by-hand". Is this approach scalable, for example when $m>40$? If the design in Fig.4(d) was obtained "by hand", what feature of QF-Map distinguish it from a regular hardware-aware mapper required to run any quantum circuit?

[Action]

Thank you for the comment. The optimizations in Fig. 4 are not automatically conducted by QF-Map. But such an optimization can be integrated into QF-Map for better results. In the Methods Section of the revision, we have clarified the unique feature of QF-Map that distinguishes it from a regular hardware-aware mapper. Specifically, QF-Map conducts the automatically mapping in two phases (1) network-to-circuit mapping; (2) virtual-to-physic mapping. The network-to-circuit mapping takes full considerations of flexibility on QF-Nets and the requirement on QF-Circ to optimize the quantum circuit, while the virtual-to-physic mapping considers the error rates on physic qbits, which will be the interface for future exploration of error-aware QF-Nets.

In this work, the main contribution of QF-Map is to optimize network-to-circuit mapping. For the basic neural computation layer (i.e., U-LYR) with the input size of 2^k , the proposed algorithm in QF-Map can guarantee the cost complexity to be $O(k^2)$, which is reduced from $O(2^k)$ needed for classical computer and the existing quantum implementation. In the Methods Section of the revision, we have added details to demonstrate how this improvement can be achieved.

On Page 12-13, in Methods Section:

Mapping QF-Nets to QF-Circ: The first task of QF-Map is to map three kinds of layers (i.e., P-LYR, U-LYR, and N-LYR) to QF-Circ. The mappings of P-LYR and N-LYR are straightforward. Specifically, for P-LYR in Figure10(a), the circuit for weight W is determined using the following rule: for a qbit I_k , an X gate is placed if and only if $W_k=-1$. Let the probability $P(x_k=-1)=P(I_k=|0\rangle)$ $=q_k$, after the X gate, the probability becomes $1-q_k$. Since the values of random variable x_k are -1 and $+1$, such an operation computes $-x_k$. For N-LYR, let O_1 be the output qbit of the first layer. It can be directly connected to the qbit I in Figure10(c)-(f), according to the type of batch normalization, which is determined by the training phase.

The mapping of U-LYR to quantum circuits is the key to achieve quantum advantages. In the following texts, we will first formulate the problem, and then introduce the proposed algorithm that can guarantee the cost for a neural computation with 2^k inputs to be $O(k^2)$.

Before formally introducing the problem, we first give fundamental definitions that need to be used. We first define the quantum state and the relationship between states as follows. Let the computational basis be composed of k qbits, as in Figure 10(b). Define $|x_i\rangle = |b_{k-1}^i, \dots, b_j^i, \dots, b_0^i\rangle$ to be the i^{th} state, where b_j is a binary number and $x_i = \sum_{\forall j} \{b_j^i \cdot 2^j\}$. For two states $|x_p\rangle$ and $|x_q\rangle$, we define $|x_p\rangle \subseteq |x_q\rangle$ if $\forall b_j^p = 1$, we have $b_j^q = 1$. We define $sign(x_i)$ to be the sign of x_i .

Figure 11. Illustration of state $|6\rangle = |0110\rangle$ and $|4\rangle = |0100\rangle$ in a $k = 4$ computation system: (a) FG_6 ; (b) PG_6 ; (c) PG_4 .

Next, we define the gates to flip the sign of states. The controlled Z operation among K qbits (e.g., C^KZ) is a quantum gate to flip the sign of states^{22, 24}. Define FG_{x_i} to be a C^KZ gate to flip the state x_i only. It can be implemented as follows: if $b_j^i = 1$, the control signal of the j^{th} qbit is enabled by $b_j^q = |1\rangle$, otherwise if $b_j^i = 0$, it is enabled by $|0\rangle$. Define PG_{x_i} to be a controlled Z gate to flip all states $\forall x_m$ if $x_m \subseteq x_i$. It can be implemented as follows: if $b_j^i = 1$, there is a control signal of the j^{th} qbit, enabled by $|1\rangle$, otherwise, it is not a control qbit. Specifically, if there is only one $b_m^i = 1$ for all $b_k^i \in x_i$, we put a Z gate on the m^{th} qbit. Figure 11 illustrates FG_6 , PG_6 and PG_4 . Different control Z gates have different costs. We define C to be a cost function from control gates to basic gates (e.g., Pauli-X, Toffoli).

Now, we formally define the weight mapping problem as follows: given (1) a vector $W = \{w_{m-1}, \dots, w_0\}$ with m binary weights and (2) a computational basis of $k = \log_2 m$ qbits that include

m states $X = \{x_{m-1}, \dots, x_0\}$ and $\forall x_j \in X, \text{sign}(x_j)$ is +, the problem is to determine a set of gates (both FG and PG) to be applied, such that the cost of all involved gates to be minimized, while the sign of each state is exactly the same with the corresponding weight; i.e., $\forall x_j \in X, \text{sign}(x_j) = \text{sign}(w_j)$.

A straightforward way to satisfy the sign flip requirement without considering cost is to apply FG for all states whose corresponding weights are -1 . A better solution for cost minimization is to use hypergraph states²⁴, which starts from the states with less $|1\rangle$, and apply PG to reduce the cost. However, as shown in the previous work²⁴, both methods have the cost complexity of $O(2^k)$, which is the same as classical computers and no quantum advantage can be achieved.

Toward the quantum advantage, we made the following important observation: the order of weights can be adjusted, since matrix MAT'_u obtained by switching two rows in the unitary matrix MAT_u will still be unitary matrix. Based on this property, we can simplify the weight mapping problem to determine a set of gates, such that the cost is minimized while the number of states with sign flip is the same with the number of -1 in weight W . On top of this, we proposed an algorithm to guarantee the cost complexity to be $O(k^2)$. Compared to $O(2^k)$ needed for classical computers, we can achieve quantum advantage. To demonstrate how to guarantee the cost complexity to be $O(k^2)$, we first have the following theorem.

Theorem 1. For an integer number R where $R > 0$ and $R \leq 2^k - 1$, it can be expressed by a set of numbers $S = \{2^i | 0 \leq i < k\}$ and signs $\{-, +\}$ with the following rules: (1) A number can be selected or not; (2) The sign of the maximum in the selected numbers is +; (3) The signs are alternatively changed for the selected numbers in order.

(Proof. Please see the manuscript.)

We propose to only use PG gate in a set $\{PG_{x(j)} | x(j) = 2^j - 1 \ \& \ j \in [1, k]\}$ for any required number $R \in (0, 2^k)$ of sign flips on states; for instance, if $k=4$, the gate set is $\{PG_{0001}, PG_{0011}, PG_{0111}, PG_{1111}\}$ and $R \in (0, 16)$. This can be demonstrated using the above theorem and properties of the problem: (1) the problem has symmetric property due to the quadratic activation function. Therefore, the weight mapping problem can be reduced to find a set of gates leading to the number of -1 no larger than $2^k - 1$; i.e., $R \in (0, 2^k - 1]$. (2) for $PG_{x(j)}$, it will flip the sign of $2^k - j$ states; since $j \in [1, k]$, the numbers of flipped sign by these gates belong to a set $S = \{2^i | 0 \leq i < k\}$; These two properties make the problem in accordance with that in Theorem 1. The weight mapping problem is also consistent with three rules in the theorem. (1) A gate can be selected or not; (2) all states at the beginning have the positive sign, and therefore, the first gate will increase sign flips; (3) $\forall p < q, x(q) \subseteq x(p)$; it indicates that among the $2^k - p$ states whose signs are flipped by $x(p)$, there are $2^k - q$ states signs are flipped back; this is in accordance to alternatively use + and - in the expression in

Theorem 1. Followed by the proof procedure, we devise the following recursive algorithm to decide which gates to be employed.

(Algorithm 4: QF-Map: weight mapping algorithm, please see the manuscript.)

In the above algorithm, the worst case for the cost is that we apply all gates in $\{PG_{x(j)} \mid x(j) = 2j-1 \ \& \ j \in [1, k]\}$. Let the state $x(j)$ has $y \mid 1 \rangle$ states, if $y > 2$ the $PG_{x(j)}$ can be implemented using $2y-1$ basic gates, including $y-1$ Toffoli gates for controlling, 1 control Z gate, and $y-1$ Toffoli gates for resetting; otherwise, it uses 1 basic gates. Based on these understandings, we can calculate the cost complexity in the worst case, which is $1+1+3+\dots+(2 \times k-1) = k^2+1$. Therefore, the cost complexity of linear function computation is $O(k^2)$. The quadratic activation function is implemented by a $C^k Z$ gate, whose cost is $O(k)$. Thus, the cost complexity for neural computation U-LYR is $O(k^2)$.

[Comment 11] Referring to the batch normalization: does the decision of normalizing the probability mean of a batch of outputs to 0.5 implies that intermediate neurons have activation values concentrated around 0.5? Does this negatively affect the number of samples required to estimate it (considering that Monte-Carlo error scales as $1/\sqrt{K}$ with K being the number of samples)?

[Action]

Thank you for the comment. Batch normalization has two steps. As stated in the comments, the first step is “normalizing the probability mean of a batch of outputs to 0.5”. But the second step makes differences for different neurons in order to make the classification easier. In consequence, there are no negative effects on the number of samples required. In the Methods Section of the revision, we have modified the illustration figure of indiv_adj to make it clear.

3rd Paragraph on Right-Hand Page 9, in Methods Section:

Figure 9(e) determination of parameter γ in batch normalization component of QF-Net

After batch_adj, the outputs of all neurons are normalized around 0.5. In order to increase the variety of different neurons' output for better classification, indiv_adj is proposed. It contains a trainable parameter λ and a parameter γ (see Figure 9(e)). It is performed in two steps: (1) we get a start point of an output p_z according to λ , and then moves it back to $p=0.5$ to obtain parameter γ ; (2) we move

p_z the angle of γ to obtain the final output. Since different neurons have different values of λ , the variation of outputs can be obtained.

[Comment 12] Overall, I believe that the framework presented in this manuscript does not support the claim of integrating ML and QC in a meaningful way. In my opinion, this requires the identification of a clear path to achieve quantum advantage. Path that is completely neglected in this work.

[Action]

Thank you again for your valuable comments. These comments help us to significantly push forward this work. We have throughout revised this paper through the following aspects: (1) adding a new design of neural computation toward the quantum advantage, (2) devising a new mapping algorithm to optimize the quantum circuit for the new neural computation. With the consideration of machine learning algorithm properties, the proposed algorithm can guarantee the gate cost to be $O(k^2)$ for the neural computation with the input size of 2^k . For comparison, the cost of the same operation on the classical computer is $O(2^k)$. This demonstrates that the quantum advantages are achieved in the proposed framework.

Thanks again for your great comments. We hope that our revision can address all your concerns.

Comments by the 2nd Reviewer:

In this manuscript, the authors propose a novel framework for training machine learning models on quantum circuits and demonstrate superior experimental results to current State of the Art in quantum machine learning. I see the main innovation of this work in an effective design of the mapping function that translates the machine learning network to a quantum circuit, and the first existing implementation of batch normalization operation.

Overall, my impression is that the paper is well-written, experimental design and theoretical formulations are appropriate and sound.

[Response]

Thanks for your encouragement. We appreciate your valuable time and helpful comments. In the revision, we have followed all the suggestions and taken actions on every comment.

[Comment 1] However, I am not entirely convinced that the claim of “co-design” in the title is justified. The authors present QF-Net as a quantum friendly network, but it seems that the quantum circuit QF-Circ is constructed specially to reproduce the behavior of the given QF-Net network, therefore, the term “co-design” might not be appropriate. There should be some evidence of co-design between QF-Net and QF-Circ. I would like to ask the authors to address this issue before being considered for possible publication.

[Action]

Thanks for the helpful comment. In the Discussion Section of the revision, we have clarified that the philosophy of co-design is demonstrated in the design of fundamental computing components in the quantum circuit to support the neural network. Based on the co-design philosophy, in the Methods Section of the revision, we have developed the new neural network QF-Net+ and the mapping algorithm QF-Map to achieve quantum advantage. Details are discussed as follow.

(Hardware-Aware Neural Network Design) For QF-Net+, first, we have proposed a new quantum-aware neural computation layer, namely U-LYR, such that 2^N data can be expressed and operated on N qbits simultaneously. Compared to the probabilistic model-based neural computation using 2^N qbits to encode data, U-LYR using N qbits provides the potential to achieve quantum advantage. Second, we have integrated the proposed two neural computations in a hybrid neural network, called QF-Net+, such that the entire network can be executed on the quantum circuit without measurement.

(Software-Aware Quantum Circuit Design) For QF-Map, we have devised a novel mapping algorithm for U-LYR. Even using N qbits to operate 2^N data, state-of-the-art hypergraph state approach-based mapping generates the quantum circuit with cost complexity of $O(2^N)$ in terms of the basic quantum logic gates (e.g., Hadamard, CNOT, Toffoli). We observed that the order

of inputs in machine learning algorithm can be changed, and in turn, the positions of weights can be changed. Based on this property, we have developed the novel mapping algorithm to guarantee the cost complexity to be $O(N^2)$. Considering that the classical computer requires $O(2^N)$ basic computing units, where each unit takes 1 clock cycle, it implies that we can achieve quantum advantage for neural computation layer, U-LYR.

3rd Paragraph on Page 8, in Discussion Section:

The philosophy of co-design is demonstrated in the design of P-LYR, U-LYR, and N-LYR. From the machine learning side, we take the known operations as the backbones in P-LYR, U-LYR, and N-LYR; while from the quantum computing side, we take full use of its ability in processing probabilistic computation and unitary matrix based computations to make P-LYR, U-LYR, and N-LYR quantum-friendly. In addition, as will be shown in the next section, the key to achieve quantum advantage for U-LYR is that QF-Map fully considers the flexibility on machine learning side (i.e., the order of inputs can be changed), while the requirement of continuously executing machine learning algorithms on the quantum computer leads to a hybrid neural network architecture with both neural computation operations: P-LYR and U-LYR. Without a co-design procedure, the previous works did not exploit quantum advantages in implementing neural networks on quantum computers, which reflects the importance of conducting co-design.

Last Paragraph on Left-Hand Page 9, in Methods Section:

Neural Computation U-LYR: Unlike P-LYR taking advantage of the probabilistic properties of qubits to provide the maximum flexibility, U-LYR aims to minimize the gates for quantum advantage using the property of the unitary matrix. The 2^k input data are first converted to 2^k corresponding data that can be the first column of a unitary matrix, as shown in Figure 7(d). Then the linear function C_u and activation quadratic function A_u are conducted. U-LYR has the potential to significantly reduce the quantum gates for computation, since the 2^k inputs are the first column in a unitary matrix and can be encoded to k qubits. But the state-of-the-art hyper-graph based approach²⁴ needs $O(2^k)$ basic quantum gates to encode 2^k corresponding weights to k qubits, which is the same with that of classical computer needing $O(2^k)$ operators (i.e., adder/multiplier). In the later section of QF-Map, we propose an algorithm to guarantee that the number of used basic quantum gates to be $O(k^2)$, achieving quantum advantages.

4th Paragraph on Right-Hand Page 12, in Methods Section:

Toward the quantum advantage, we made the following important observation: the order of weights can be adjusted, since matrix MAT'_u obtained by switching two rows in the unitary matrix MAT_u will still be unitary matrix. Based on this property, we can simplify the weight mapping problem to determine a set of gates, such that the cost is minimized while the number of states with sign flip is the same with the number of -1 in weight W . By this simplification, we proposed an algorithm to

guarantee the cost complexity to be $O(k^2)$. Compared to $O(2^k)$ needed for classical computers, we can achieve quantum advantage. To demonstrate how to guarantee the cost complexity to be $O(k^2)$, we first have the following theorem. ...

Algorithm 4: QF-Map: weight mapping algorithm

```

Input: (1) An integer  $R \in (0, 2^{k-1}]$ ; (2) number of qubits  $k$ ;
Output: A set of applied gate  $G$ 
void recursive( $G, R, k$ ){
    if ( $R < 2^{k-2}$ ){
        recursive( $G, R, k - 1$ ); // Case 1 in the third step
    }
    else if ( $R == 2^{k-1}$ ){
         $G.append(PG_{2^{k-1}})$ ; // Case 2 in the third step
        return;
    }else{
         $G.append(PG_{2^{k-1}})$ ;
        recursive( $G, 2^{k-1} - R, k - 1$ ); // Case 3 in the third step
    }
}
// Entry of weight mapping algorithm
set main( $R, k$ ){
    Initialize empty set  $G$ ;
    recursive( $G, R, k$ );
    return  $G$ 
}

```

In the above algorithm, the worst case for the cost is that we apply all gates in $\{PG_{x(j)} \mid x(j) = 2j-1 \ \& \ j \in [1, k]\}$. Let the state $x(j)$ has y $|1\rangle$ states, if $y > 2$ the $PG_{x(j)}$ can be implemented using $2y-1$ basic gates, including $y-1$ Toffoli gates for controlling, 1 control Z gate, and $y-1$ Toffoli gates for resetting; otherwise, it uses 1 basic gates. Based on these understandings, we can calculate the cost complexity in the worst case, which is $1+1+3+\dots+(2 \times k-1) = k^2+1$. Therefore, the cost complexity of linear function computation is $O(k^2)$. The quadratic activation function is implemented by a C^kZ gate, whose cost is $O(k)$. Thus, the cost complexity for neural computation U-LYR is $O(k^2)$.

Below I list some comments that require revision:

[Comment 2] In Introduction, the caption to Figure 1(b) is hard to understand

[Action]

Thanks for pointing it out. In the revision, we have moved the mentioned sub-figure Figure 1(b) to Figure 7(a) to make Figure 1 more concise. In the Methods Section of the revision, we have rewritten the caption of this figure to make it clear. Specifically, the caption is written as follows: “prepossessing input data by *i*) down-sampling the original 28×28 image in MNIST to 4×4 image and *ii*) converting each data in the down-sampled image from real number to a random variable following a two-point distribution”.

On Page 2, in Introduction Section:

On Page 8, in Methods Section:

Figure 1. QuantumFlow, an end-to-end co-design framework, provides a missing link between machine learning and quantum circuit designs, which is composed of QF-Nets, QF-Net+, QF-FB, QF-Circ, QF-Map that work collaboratively to design neural networks and their quantum implementations.

Figure 6. Neural Computation: (a) preprocessing of inputs by *i*) down-sampling the original 28×28 image in MNIST to 4×4 image and *ii*) get the 4×4 matrix with grey level normalized to $[0, 1]$. (b-c) QF-NC_p: (b) input data are converted from real number to a random variable following a two-point distribution; (c) four operations in QF-NC_p, *i*) R: converting a real number ranging from 0 to 1 to a random variable, *ii*) C: average sum of weighted inputs, *iii*) A: non-linear activation function, *iv*) E: converting random variable to a real number. (d-e) QF-NC_u: (d) input data are converted to become the first column of a unitary matrix; (e) three operations in QF-NC_u, *i*) U: unitary matrix converter, *ii*) C_u: average sum of weighted inputs; *iii*) A_u: non-linear activation function.

[Comment 3] In Results - QF-FB(C) and QF-FB(Q) are consistent, the second paragraph states that: “We can see that there is a small difference between QF-FB(C) and QF-FB(Q)”. (1) However, it is questionable that Accuracy values in Table 1 can be considered small. Further questions arise about the experiments: (2) How many times the experiments were repeated? (3) What was the variance? (4) Why the datasets were matched in such way?

[Action]

Thanks for pointing this out. In the **Results Section** of the revision, we have conducted more results for the neural network QF-Net+ designed by fully considering quantum circuit implementation. It better demonstrates that the consistency between QF-FB(C) and QF-FB(Q). In addition, we have added experimental details to respond to questions (2) and (3). For question (4), we have added more sub-datasets from MNIST for comparison of accuracy in Figure 2, and we have also added the explanations on the reason that we select 3 representative datasets in Table 1. Detailed actions are listed as follows one by one.

(1) In the new experiments on QF-Net+, the difference between QF-FB(C) and QF-FB(Q) is less than 0.9%. The new results verify the statement that the accuracy difference is small.

(2) In the revision, we have discussed that the results for QF-FB(C) in different runs are the same. For QF-FB(Q), we launch 8,192 shots for each input image.

(3) In the revision, we have discussed that the variance comes from the fundamental Monte Carlo method used in the simulator of Qiskit Aer simulation, which is the base of QF-FB(Q).

Last Paragraph on Left-Hand Page 4, in Results Section:

Table 1. Inference accuracy and efficiency comparison between QF-FB(C) and QF-FB(Q) on both QF-Net and QF-Net+ using MNIST dataset to show the consistency of implementations of QF-Nets on classical computers and quantum computers.

dataset	QF-Net Qbits (Neurons)		Accuracy			Elapsed CPU Time		QF-Net+ Qbits (Neurons)		Accuracy			Elapsed CPU Time	
	L1	L2	QF-FB(C)	QF-FB(Q)	Diff.	QF-FB(C)	QF-FB(Q)	L1	L2	QF-FB(C)	QF-FB(Q)	Diff.	QF-FB(C)	QF-FB(Q)
{3,6}	28(4)	12(2)	97.10%	95.53%	-1.57%	5.13S	2,555H	7(4)	5(2)	98.27%	97.46%	-0.81%	4.30S	16.57H
{3,8}	28(4)	12(2)	86.84%	83.59%	-3.25%	5.59S	2,631H	7(4)	5(2)	87.40%	88.06%	+0.54%	4.05S	16.56H
{1,3,6}	28(8)	18(3)	87.91%	81.99%	-5.92%	15.89S	14,650H	7(8)	8(3)	88.53%	88.14%	-0.39%	6.96S	47.98H

Column “Accuracy” in Table 1 reports the accuracy comparison. For QF-FB(C), there will be no difference in accuracy among different executions. For QF-FB(Q), we implement the obtained QF-Circ from QF-Nets on Qiskit Aer simulation with 8,192 shots. We have the following two observations from these results: (1) There exist accuracy differences between QF-FB(C) and QF-FB(Q). This is because Qiskit Aer simulation used in QF-FB(Q) is based on the Monte Carlo method, leading to the variation. In addition, since the out-put probability of different neurons may quite close in some cases, it will easily result in different classification results for small variations. (2) Such accuracy differences for QF-Net+ is much less than that of QF-Net, because QF-Net utilizes much more qbits, which leads to the accumulation of errors. In QF-Net+, we can see that there is a small difference between QF-FB(C) and QF-FB(Q). For the dataset {3,8}, QF-FB(Q) can even achieve higher accuracy. The above results demonstrate both QF-Net and QF-Net+ can be consistently implemented on classical and quantum computers.

(4) In QF-Nets evaluations in the revision, we have added sub-datasets in MNIST to show that the proposed QF-Nets can work for different scales. In addition, we have discussed why we choose these datasets from MNIST. Specifically, we have added {1,5}, {3,9}, {0,3,6}, {0,3,6,9}, {0,1,3,6,9}, {0,1,2,3,4} for the evaluation. Results consistently show that the proposed QF-Nets can achieve state-of-the-art accuracy compared with the competitors. We have also discussed that dataset {3,6} is simpler than others for classification and typically used in quantum machine learning studies.

In QF-FB evaluations in the revision, we select three sub-datasets {3,6}, {3,8}, {1,3,6}. We have discussed that this is because the limitation of the Qiskit Aer simulator can support the simulation of 32 qbits at most.

Last Paragraph on Left-Hand Page 3, in Results Section:

... Before reporting the detailed results, we first discuss the experimental setting. In this experiment, we extract sub-datasets from MNIST, which originally include 10 classes. For instance, {3,6}

indicates the sub-datasets with two classes (i.e., digits 3 and 6), which are commonly used in quantum machine learning (e.g., Tensorflow-Quantum³⁶). To evaluate the advantages of the proposed QF-Nets, we further include more complicated sub-datasets, {3,8}, {3,9}, {1,5} for two classes. In addition, we show that QF-Nets can work well on larger datasets, including {0,3,6} and {1,3,6} for three classes, and {0,3,6,9}, {0,1,3,6,9}, {0,1,2,3,4} for four and five classes. For the datasets with two or three classes, the original image is downsampled from the resolution of 28×28 to 4×4, while it is downsampled to 8×8 for datasets with four or five classes. All original images in MNIST and the downsampled images are with grey levels. For all involved datasets, we employ a two-layer neural network, where the first layer contains 4 neurons for two-class datasets, 8 neurons for three-class datasets, and 16 neurons for four- and five-class datasets. The second layer contains the same number of neurons as the number of classes in datasets. ...

Figure 2. QF-Net+ achieves state-of-the-art accuracy in image classifications on different sub-datasets of MNIST.

2nd Paragraph on Page 4, in Results Section:

QF-FB(C) and QF-FB(Q) are Consistent

... We select three datasets, including {3,6}, {3,8}, and {1,3,6}, for evaluation. Datasets with more classes (e.g., {0,3,6,9}) are based on larger inputs, which will lead to the usage of qubits in QF-Net to exceed the limitation (i.e., 32 qubits). ...

New References

36. Google. Tensorflow quantum. <https://www.tensorflow.org/quantum/tutorials/mnist> (2020). Accessed: 2020-08-18

[Comment 4] In the third paragraph, does “This verifies that QF-FB can provide...” refer to QF-FB(C) or (QF-FB(Q)?

[Action]

Thanks for pointing this out. In the **Results Section** of the revision, we have clarified that it refers to QF-FB(C).

2nd Paragraph on Right-Hand Page 4, in Results Section:

The speedup is more than six orders of magnitude larger (i.e., $10^6\times$) for QF-Net, and more than four orders of magnitude larger (i.e., $10^4\times$) for QF-Net+. This verifies that QF-FB(C) can provide an efficient forward propagation procedure to support the lengthy training of QF-Net.

[Comment 5] In QF-Circ and QF-Map on IBM Quantum Processor, the definition of “deviation” mentioned in the third paragraph and in Figure 5, should be made clear.

[Action]

Thanks for your suggestion. In the **Results Section** of the revision, we have added discussions on the deviation. Specifically, we take the results of QF-Sim as the baseline. Then, the deviation is the difference of results between the baseline and results obtained by other approaches.

1st Paragraph on Page 5, in Results Section:

In this experiment, the result of QF-FB(C) is taken as a baseline. In the figure, the x-axis and y-axis represent the inputs and deviation, respectively. The deviation indicates the difference between the baseline and the results obtained by Qiskit Aer simulation or that by executing on IBM quantum processor.

[Comment 6] In Figure 5, subplots (a)-(d) present different designs to explore relations between CNOT numbers and the error rates. It is clear that design in (d) is superior, and it seems unnecessary to present the results from all of the designs in the Figure 5(e). We feel it is sufficient to show only (a) and (d) for comparison, to present the effect of CNOT numbers more clearly.

[Action]

Thanks for the helpful suggestion. In the **Results Section** of the revision, per your suggestion, we have removed the results of design 2 (i.e., subplot (b)) and design 3 (i.e., subplot (c)) to make the comparison concise.

4th Paragraph on Page 6, in Results Section:

Figure 4. Evaluation of the quantum circuits for a two-input neural computation, where weights are $\{-1,+1\}$: (a) design 1: original neural computation design; (b-d) three optimized designs (design 2-4), based on design 1; (e) the deviation of design 1 and design 4 obtained from “ibm_velencia” backend IBM quantum processor, using QF-FB(C) as golden results.

... Figure 5(e) reports the deviations of design 1 and design 4 against the golden results. The results clearly show that design 4 is more robust because it uses fewer qubits in the circuit. ...

[Comment 7] In Figure 5 and other places, component x of Pauli matrices (σ_x) is denoted as X , but denoting the rotation of y , R_y as Y is confusing.

[Action]

Thanks for the helpful comment to make the presentation clear. Per your suggestions, in the revision, we have revised all figures in the paper to denote the rotation of y as R_y , instead of Y .

Figure 5 on Page 6, in Results Section:

[Comment 8] In Discussion, the first paragraph, the sentence “..the QF-Circ can automatically generate and optimize a corresponding quantum circuit” implies that QF-Circ is a function that generates quantum circuit for QF-Net. On the other hand, in the Introduction, second paragraph from the end (“..QF-Map is an automatic tool to conduct (1) network-to-circuit mapping (from QF-Net to QF-Circ)..”), it appears that QF-Circ is generated from QF-Net by the QF-Map. To avoid confusion, it should be clearly distinguished throughout the paper, what is the function, and what it the product generated by the function.

[Action]

Thanks for the comment. In the Discussion Section of the revision, we have revised the inconsistent statements. Specifically, we have clarified that QF-Map is the function for generating quantum circuits for QF-Nets, and QF-Circ is the generated quantum circuit. We have also gone through the whole paper to make it consistent.

2nd Paragraph on Page 7, in Discussion Section:

In summary, we propose an integrated QuantumFlow framework to co-design the machine learning models and quantum circuits. Novel quantum-aware QF-Nets are first designed. Then, an accurate and efficient inference engine, QF-FB, is proposed to enable the training of QF-Nets on classical computers. Based on QF-Nets and the training results, the QF-Map can automatically generate and optimize a corresponding quantum circuit, QF-Circ. Finally, QF-Map can further map QF-Circ to a quantum processor in terms of qubits’ error rates.

[Comment 9] In *Methods - QF-Net*, second paragraph, iv) “ E: this operation converts the random variable y_2 to 0-1 real number by taking its expectation” – is the operation $R \rightarrow C \rightarrow A$ repeated multiple times to obtain a mean value?

[Action]

Thanks for the comment. In the **Methods Section** of the revision, we have clarified that the differences in applying these operations on classical computers and quantum computers. Specifically, for the quantum computer, **R** operations are only conducted in the first layer. Then, **C** and **A** operations are repeatedly for multiple times without measurement because operations on a random variable can be directly operated on qbits. Finally, the measurement is conducted in the last layer, which is corresponding to **E** operation. However, for the classical computer, **R**, **C**, **A**, **E** operations are sequentially conducted in each layer.

1st Paragraph on Right-Hand Page 9, in Methods Section:

QF-Net is composed of multiple layers of P-LYR. For its quantum implementation, operations on random variables can be directly operated on qbits. Therefore, **R** operation is only conducted in the first layer. Then, **C** and **A** operations will be repeated without measurement. Finally, at the last layer, we measure the probability for output qbits, which is corresponding to the **E** operation.

[Comment 10] In *QF-Map*, figure 10(b), the random values generated from the inputs from I_k ($k_0, \dots, 3$) are fed into two C/A. Are these random values different for each C/A input? If so, there is no problem, but if these are the same random variables, it is strange that they would output different results, as presented in Figure 10(b) and Figure 10(c).

[Action]

Thanks for the comment. In **Subsection QF-Map of Section Methods** of the revision, we have revised the Figure 10(b) (which is Figure 5(b) in the supplementary information in the revision) to make the neural computation clear. Specifically, the inputs will be operated on weights in the neural computation, where inputs are the same, but the weights are different. As a result, the output of C/A will be different.

On Page 7, in Section 4 in Supplementary Information:

As shown in Figure 5(c), we utilize two independent sets of qbits ($\{I_0, \dots, I_3\}$ and $\{I_0', \dots, I_3'\}$) for computing two output neurons using two different weights (W_0 and W_1) in the first layer; while the output qbits of the first layer (i.e., O_2 and O_3) are shared for the calculation of different neurons in the second/last layer.

[Comment 11] In Supplementary Information, 1.3 CNOT gate on two initialized qbits, “...corresponding to states $[|00\rangle + |01\rangle + |10\rangle + |11\rangle]^T$ – such notation may lead to misunderstanding. Should instead be written as $A|00\rangle + B|01\rangle + C|10\rangle + D|11\rangle$ ”

[Action]

Thanks for the comment. In Section 1.3 in Supplementary Information of the revision, we have revised the misleading presentation per your suggestions.

On Page 1, in Section 1.3 of Supplementary Information:

$$\dots \text{ we have } |q_0, q_1\rangle = |q_0\rangle \otimes |q_1\rangle = \begin{bmatrix} \cos \frac{\alpha}{2} \times \cos \frac{\beta}{2} \\ \cos \frac{\alpha}{2} \times \sin \frac{\beta}{2} \\ \sin \frac{\alpha}{2} \times \cos \frac{\beta}{2} \\ \sin \frac{\alpha}{2} \times \sin \frac{\beta}{2} \end{bmatrix}, \text{ denoted as } \begin{bmatrix} A \\ B \\ C \\ D \end{bmatrix}, \text{ where } A = \cos \frac{\alpha}{2} \times \cos \frac{\beta}{2},$$

$B = \cos \frac{\alpha}{2} \times \sin \frac{\beta}{2}, C = \sin \frac{\alpha}{2} \times \cos \frac{\beta}{2},$ and $D = \sin \frac{\alpha}{2} \times \sin \frac{\beta}{2}.$ Then we can represent the states of qbits q_0 and q_1 as $|q_0, q_1\rangle = A|00\rangle + B|01\rangle + C|10\rangle + D|11\rangle.$...

[Comment 12] Lastly, the manuscript would benefit from proofreading, as there are some minor language mistakes, for example in the abstract: “In this article we present... .. to fixed the missing link”.

[Action]

Thanks for pointing out the typos. In this revision, we have fixed them. In addition, we have gone through the whole paper to fix typos and grammars to improve the quality of this paper.

Comments by the 3rd Reviewer:

This manuscript presents a novel framework for machine learning architecture for circuit model quantum computing. Authors consider a set of transformations native to quantum dynamics to introduce nonlinearity required for machine learning tasks. The proposed ML model is then mapped directly to a quantum circuit through a set of developed tools. Some proof-of-concept experiments have been tested on IBM 5-qubit machines. Overall, I found the work an interesting contribution to the active research area of QML. I have few questions before recommending for publication:

[Response]

Thanks very much for your encouragement. We appreciate your valuable time to give us helpful comments. In the revision, we have carefully considered and taken actions to your comments. The detailed actions are reported as follows.

[Comment 1] Authors refer to batch normalization as a motivation for the introduced framework. My question is that why is batch normalization needed for QML circuit models? In neural network literature BN was introduced to alleviate the covariate shift problem which was due to the nonlinearity of neurons. Why do you think similar problem would arise in the quantum domain?

[Action]

Thanks for the comment. First, in the Abstract and Introduction Section of the revision, we have clarified that the main contribution and motivation of this paper is to achieve quantum advantages via the co-design of neural network and quantum circuits. We demonstrate that by fully consider the properties of neural network and quantum circuit designs, we are able to reduce the cost complexity from $O(2^k)$ on a classical computer to $O(k^2)$ on the quantum circuit built by the QuantumFlow framework.

Second, in Discussion Section, we have discussed that why batch normalization is required in QML circuit models. Specifically, the backbone of QF-Nets has similar structure to the neural network designed for classical computer, which also involves the nonlinearity into neurons. Taking quadratic function as an example, if no normalization is applied, the original data under several rounds of quadratic will be significantly shrunken to a small range around $1/m$, where m is the number of inputs. As a result, it is hard to classify different neurons without normalization.

Third, in the Results Section, we have shown that the developed batch normalization can indeed be helpful in improving model accuracy.

On Page 1, in Abstract:

We discover that, in order to make full use of the strength of quantum representation, it is best to represent data in a neural network as random variables or numbers in unitary matrices, such that they can be directly operated by the basic quantum logical gates. Based on these data representations,

we propose two quantum friendly neural networks, QF-Net and QF-Net+ in QuantumFlow. QF-Net using the probabilistic model has better flexibility, which can seamlessly connect two layers without measurement but requires more qubits and logical gates. On the other hand, the unitary matrix applied in QF-Net+ can help to encode 2^k data into k qubits, and a novel algorithm can guarantee the cost complexity (i.e., logical gates) to be $O(k^2)$. Compared to the cost of $O(2^k)$ in classical computing and the existing quantum implementations, QF-Net+ demonstrates the quantum advantages. ... Results further show that for input sizes of neural computation grow from 16 to 2048, the cost reduction of QuantumFlow increased from $2.4\times$ to $64\times$. Furthermore, on MNIST dataset, QF-Net+ can achieve accuracy of 94.09%, while the cost reduction against the classical computer reaches $10.85\times$...

2nd Paragraph on Page 8, in Discussion Section:

Batch normalization is another key technique in improving accuracy, since the backbone of the quantum-friendly neuron computation layers (P-LYR and U-LYR) is similar to that in classical computers, using both linear and non-linear functions. This can be seen from the results in Figure 2. Batch normalization can achieve better model accuracy, mainly because the data passing a nonlinear function y^2 will lead to outputs to be significantly shrunken to a small range around 0 for real number representation and $1/m$ for a two-point distribution representation, where m is the number of inputs.

[Comment 2] It would be nice to discuss possibility of extension of your model to higher orders of non-linearity beyond the quadratic terms. How does circuit complexity increase?

[Action]

Thanks for the comment. In the Methods Section of the revision, we have discussed the extension of non-linear function on a new neural computation component, U-LYR, which has demonstrated incomparable cost complexity over other designs and demonstrate quantum advantages. Since the implementation of higher orders of non-linearity needs the duplication of circuits, it is more reasonable to integrate the higher order of non-linear functions into U-LYR.

Specifically, according to the required non-linear function, say y^k , we can first obtain $k/2$ outputs. Then, we use controlled NOT gate on these outputs. For the case that k is odd number, we can perform conduct square root on inputs. Let $O(N)$ be the cost complexity of U-LYR using quadratic function, then the cost of U-LYR using y^k will be $O(kN)$.

1st Paragraph on Right-Hand Page 11, in Methods Section:

Kindly note that, in addition to quadratic activation, we can also implement higher orders of non-linearity by duplicate the circuit to perform U, Cu, and Au to achieve multiple outputs. Then, we can use control NOT gate on the outputs to achieve higher orders of non-linearity. For example, using a Toffoli gate on two outputs can realize y^4 . Let the non-linear function be y^k and the cost

complexity of U-LYR using quadratic activation be $O(N)$, then the cost complexity of U-LYR using y^k as the non-linear function will be $O(kN)$.

[Comment 3] The first stage of the model where you map data into quantum random variables in a trick similar the one introduced in the IBM paper on SVMs. You should cite that paper: V Havlicek, “Supervised learning with quantum-enhanced feature spaces”, *Nature*. vol. 567, pp. 209-212 (2019).

[Action]

Thanks for the helpful comment. In the Introduction Section of the revision, we have added the reference. The mentioned reference is a good example to demonstrate the importance of data representation (instead of simple binary representation) in machine learning algorithms.

2nd Paragraph on Page 1, in Introduction Section:

To overcome this problem, some existing works^{26–29} assume binary representation (i.e., “-1” and “+1”) of activation, which cannot well represent data as seen in modern machine learning applications. For instance, in computer vision related applications, data in images are commonly represented as real numbers. This has also been demonstrated in work³¹, where data in the interval of $(0, 2\pi]$ instead of binary representation are mapped onto the Bloch sphere to achieve high accuracy for support vector machines (SVMs).

New References

31. Havlicek, V. et al. Supervised learning with quantum-enhanced feature spaces. *Nature* 567, 209–212 (2019)

Reviewers' Comments:

Reviewer #1:

Remarks to the Author:

I would like to acknowledge the work made by the authors to reply to all points I raised in the previous review round. It is evident that there was genuine effort to improve the quality of the manuscript. Consequently, the manuscript changed substantially. At the same time, I feel that a few important points are still unclear to me.

I have two main concerns. The first one relates to the quantum advantage; the second one relates to the confidence in the simulation results. A third issue refers to the quantum resources required to implement the QF-Net.

1. Quantum advantages has limitations not properly presented to the reader.

- In U-LYR, the operation MATu in Fig. 10(b) is not included in the $O(k^2)$ cost. While the development of quantum memory is a hard problem to solve, it is a common requirement in many quantum protocols, and it is fine to exclude its cost.

- The implementation of "QF-MAP W" in Fig.10 (b) is very interesting. However, it implies that, given a number R of negative weights, those negative weights have specific indices. Therefore MATu has to be changed with MATu', which has a different permutation of the 2^k rows. This is mentioned in the manuscript.

- While one can assume MATu to be available (meaning that the order of the input is fixed), MATu' must be obtained from MATu with $O(k^2)$ quantum gates or it will dominate the cost. This is not discussed in the paper and it is hard to believe that a generic permutation of the 2^k rows can be implemented with $O(k^2)$ quantum gates and classical processing.

- One cannot expect MATu' to be provided since the weights change during training and therefore the permutation changes as well.

- A separate problem with U-LYR is that it can be used as only the first layer. One has then to continue with P-LYR layers. The quantum advantage is retained only if the number of neurons per layer is reduced exponentially from the first layer to the second (for example from 2^k inputs, to $\text{poly}(k)$ neurons in the first intermediate layer).

- In addition, while commenting on Table 1, the authors write: "The speedup of QF-FB(C) over QF-FB(Q) is more than six orders of magnitude larger [...]. This verifies that QF-FB(C) can provide an efficient forward propagation procedure to support the lengthy training of QF-pNET.". Does the author intend to use the quantum network for inference and perform the training on classical networks? If yes, the quantum advantage is confined to the inference (and not the learning).

2. My concern relates to how the simulations were performed. It seems to me that the circuit has been simulated piecewise (see sentence in page 4: "Because of the limitation of Qiskit Aer [...], we measure the results after each neuron."), but that this procedure may neglect correlations building throughout the network among neurons even in the same layer.

My understanding is that simulations (for example of the first part of the circuit) were used to compute the probability of some intermediate qubit (let us say qubit k) to be in $|0\rangle$ (probability denoted qk) or $|1\rangle$ (probability denoted $p_k=1-q_k$). Then, this classical information was used to initialize qubit k for the next piecewise simulation to: $\sqrt{q_k}|0\rangle + \sqrt{p_k}|1\rangle$. The other qubits involved in the first piecewise simulation (and that are now entangled with qubit k) were simply disregarded.

The issue with this approach is that correlations between intermediate qubits (even in the same

layer) are not accounted in that kind of simulations. This is the same effect mentioned in my previous report (comment 5, part 2):

"(2) Referring to Fig.9(c), it is unclear to me why the outputs of the first hidden layer (qubits O2 and O3) can be used to update both neurons in the second hidden layer. Despite the measurement of O4 is delayed, it will still influence that of O5. [...] the statistics of outcomes O4 and O5 would not be as intended [...]"

The authors' response:

"For question (2), according to the above discussion that the probability of input qubits will not be changed throughout the neural computation process, making it possible to use the same inputs (i.e., the outputs of layer 1, O2 and O3) to calculate multiple neurons in a layer (i.e., layer 2). [...]"

focuses on the (unconditional) probability of the inputs. My point is about the correlation between outputs. To clarify, consider the simple circuit below with one input and two outputs:

qubit 0 = In0 : $|0\rangle$ --H--*--o--

| |

qubit 1 = Out0: $|0\rangle$ -----X--|--

|

qubit 2 = Out1: $|0\rangle$ -----X--

There are three gates: Hadamard, CNOT, and the X gate conditioned on the control qubit being in $|0\rangle$. The input state (after the Hadamard gate) corresponds to a random variable with $p_0=q_0=0.5$. Each output qubit has balanced probabilities, corresponding to random variables with: $p_1=q_1=p_2=q_2=0.5$.

But let us look at the joint probability of the outputs. If the outputs were independent (as assumed in QF-Net), the probability of measuring them both in state $|0\rangle$ should be q_1*q_2 . For the above circuit, it is 0. This is because using the same input qubits to update two output qubits in the next layer correlates them (in the example above and in the QF-Net, actually entangles them).

The binary classifier, discussed as an actual implementation on IBM chips, goes around this problem since has a single neuron update.

To summarize point 2, results from QF-FB(Q) may not correspond to a quantum implementation of QF-FB(C) since correlations between intermediate neurons are neglected. However, the piecewise simulations do correspond to a quantum version of QF-FB(C) since correlations are discarded.

3. The authors should clarify how many qubits are required in their construction. For example, a generic formula should be provided when discussing special cases in page 4 (left column).

For the batch normalization, does QF_FC(Q) requires 4 extra qubits per neuron? Two for "indiv_adj" and two for "batch_adj"?

From the considerations in point 2 above, it seems to me that (to exactly reproduce QF-FB(C) without intermediate measurements) one needs a number of input registers that grows as the product of the total number of qubits in subsequent layers (i.e. if we have N_0 inputs, N_1 neurons in the first layer, N_2 in the second layer, ..., then one may need $N_0*N_1*N_2*...$ input qubits). In general, one seems to need a number of qubits exponential in the network depth. Isn't it the case?

In addition, Table 1 should not report the time elapsed on CPU to simulate QF-FB(Q), but the expected time to run it with actual quantum devices. For example, the authors can consider a simplified quantum device with all-to-all connectivity and a single duration for every gate.

Notation and terminology:

- Usually, quantum bits are spelled "qubits" and not "qbites".
- The common term is "physical qubits" and not "physic qbites".
- The first time "batch normalization" is mentioned, a definition should be provided. Possibly together an intuitive explanation of why it is important for classical NN.
- For the introduction, other versions of quantum neural networks exist. For example:
 - o Wan, K. H., Dahlsten, O., Kristjánsson, H., Gardner, R., & Kim, M. S. (2017). Quantum generalisation of feedforward neural networks. *Npj Quantum Information*, 3, 36.
 - o Cao, Y., Guerreschi, G. G., & Aspuru-Guzik, A. (2017). Quantum Neuron: an elementary building block for machine learning on quantum computers. ArXiv:1711.11240.

Reviewer #2:

Remarks to the Author:

The authors have thoroughly responded to my comments and clarified issues found in the paper's first draft. I think the authors improved the paper enough, and I am content to recommend it as ready for publication.

Reviewer #3:

Remarks to the Author:

Authors have properly addressed my questions in the revised version and I am happy to recommend the paper for publication in its current version.

Thanks,

Alireza Shabani

Response to Referees

Title: A Co-Design Framework of Machine Learning and Quantum Circuits Towards Quantum Advantage`

Type: 2nd Round Revision

We would like to express our sincere thanks to the reviewers for their valuable time and constructive comments to enable us to continuously improve the quality of the submission. We sincerely hope that this revision has addressed all the review comments.

In this report, we have listed the detailed actions for each comment. The review comments are shown in *italic*, our detailed responses are shown in **bold and upright**, and the modifications in the revised paper are placed within rectangles.

Outline:

- Summary of changes (Page 2)
- Actions for the comments from the 1st reviewer (Pages 3-13)
- Response to the 2nd reviewer (Page 14)
- Response to the 3rd reviewer (Page 14)

Summary of Changes:

1. We have clarified the assumptions and quantitatively analyzed the conditions to achieve quantum advantages of the proposed QF-hNet.

- (Introduction) This work targets the acceleration of the inference phase on quantum circuits
- (Method Section) The QF-hNet and QF-pNet are designed as shallow networks
- (Discussion Section and Method Section) Quantum state-preparation can be efficiently conducted using a qRAM-based method or a computing-based method
- (Section 5 of Supplementary Information) A quantitative analysis on the effects of network depth and the neuron numbers to cost complexity are discussed

2. We have clarified that the simulation does not consider the correlation and the proposed quantum implementations for QF-pNet and QF-hNet are consistent to such a simulation.

- (Result Section) The simulation does not consider the correlation
- (Section 5 of Supplementary Information) For the shallow networks QF-pNet and QF-hNet, the correlation incurred by the quantum circuits can be eliminated by the circuit design.
- (Method Section) For a deep neural network, a hybrid quantum-classical computing can eliminate the correlation after the measurement.

3. We have discussed how to take advantage of the proposed U-LYR to support deep neural networks and demonstrated the potential quantum advantages in the Method Section.

4. We have added the following new references:

- [31] Cao, Y., Guerreschi, G. G. & Aspuru-Guzik, A. Quantum neuron: an elementary building block for machine learning on quantum computers. arXiv preprint arXiv:1711.11240 (2017).
- [32] Wan, K. H., Dahlsten, O., Kristjánsson, H., Gardner, R. & Kim, M. Quantum generalisation of feedforward neural networks. npj Quantum information 3, 1–8 (2017).
- [42] Allcock, J., Hsieh, C.-Y., Kerenidis, I. & Zhang, S. Quantum algorithms for feedforward neural networks. ACM Transactions on Quantum Comput. 1, 1–24 (2020).
- [43] Kerenidis, I. & Prakash, A. Quantum recommendation systems. arXiv preprint arXiv:1603.08675 (2016).
- [45] Xia, R. & Kais, S. Hybrid Quantum-Classical Neural Network for Calculating Ground State Energies of Molecules. Entropy 22, 828 (2020).
- [46] Cong, I., Choi, S. & Lukin, M. D. Quantum convolutional neural networks. Nat. Phys. 15, 1273–1278 (2019).
- [47] Otterbach, J. et al. Unsupervised machine learning on a hybrid quantum computer. arXiv preprint arXiv:1712.05771 (2017).
- [48] Perdomo-Ortiz, A., Benedetti, M., Realpe-Gómez, J. & Biswas, R. Opportunities and challenges for quantum-assisted machine learning in near-term quantum computers. Quantum Sci. Technol. 3, 030502 (2018).
- [49] Sanders, Y. R., Low, G. H., Scherer, A. & Berry, D. W. Blackbox quantum state preparation without arithmetic. Phys. review letters 122, 020502 (2019).
- [50] Grover, L. K. Synthesis of quantum superpositions by quantum computation. Phys. review letters 85, 1334 (2000).
- [51] Bausch, J. Fast Black-Box Quantum State Preparation. arXiv preprint arXiv:2009.10709 (2020).
- [52] Di Matteo, O., Gheorghiu, V. & Mosca, M. Fault-tolerant resource estimation of quantum random-access memories. IEEE Transactions on Quantum Eng. 1, 1–13 (2020).
- [53] Klauck, H., Špalek, R. & De Wolf, R. Quantum and classical strong direct product theorems and optimal time-space tradeoffs. SIAM J. on Comput. 36, 1472–1493 (2007).
- [54] Kim, P., Han, D. & Jeong, K. C. Time–space complexity of quantum search algorithms in symmetric cryptanalysis: applying to AES and SHA-2. Quantum Inf. Process. 17, 339 (2018).
- [55] Frank, M. P. & Ammer, M. J. Relativized separation of reversible and irreversible space-time complexity classes. arXiv preprint arXiv:1708.08480 (2017).

Comments by the 1st Reviewer:

I would like to acknowledge the work made by the authors to reply to all points I raised in the previous review round. It is evident that there was genuine effort to improve the quality of the manuscript. Consequently, the manuscript changed substantially. At the same time, I feel that a few important points are still unclear to me.

I have two main concerns. The first one relates to the quantum advantage; the second ones relates to the confidence in the simulation results. A third issue refers to the quantum resources required to implement the QF-Net.

[Response]

Thanks very much for your encouragement. We appreciate your valuable time to give us helpful comments. In the revision, we have carefully considered your comments and taken actions to address them. The detailed actions are reported as follows.

Quantum advantages has limitations not properly presented to the reader.

[Comment 1] In U-LYR, the operation MATu in Fig. 10(b) is not included in the $O(k^2)$ cost. While the development of quantum memory is a hard problem to solve, it is a common requirement in many quantum protocols, and it is fine to exclude its cost.

[Action]

Thanks for the comment. In Discussion Section of the revision, we have clarified that the cost for a neuron computation does not include the cost for the quantum state-preparation cost. In addition, in the Method Section of the revision, we have added discussions on how we conduct the state-preparation using an additional structure in qRAM, which follows the same methods in the existing works. In the following texts, we use n to represent the input size, instead of k^2 .

Last Paragraph on Page 7, in Discussion Section:

Specifically, with the help of QF-Map, the cost complexity can be reduced from $O(n)$ on classical computing (i.e., FC(C)) to $O(\log^3 n)$, demonstrating the potential quantum advantage. Kindly note that this complexity involves neuron computation only. In addition to neuron computation, it also needs the quantum state-preparation, which can be conducted efficiently with the help of quantum random access memory (details see Method Section). Thus, its complexity does not include.

The 4th Paragraph on Page 12, in Method Section:

There are two main steps in U-LYR: the state-preparation (i.e., U or encoding MATu) and the weighted sum computation (i.e., C_u and A_u). For the state-preparation, since the inputs are given, we apply the same approach in works^{46,47} by using quantum random access memory (qRAM)⁴⁸.

Specifically, the vector in MATu will be stored in a binary-tree based structure in qRAM, which can be queried in quantum superposition and can generate the states efficiently.

New references:

46. Allcock, J., Hsieh, C.-Y., Kerenidis, I. & Zhang, S. Quantum algorithms for feedforward neural networks. *ACM Transactions on Quantum Comput.* 1, 1–24 (2020).

47. Kerenidis, I. & Prakash, A. Quantum recommendation systems. *arXiv preprint arXiv:1603.08675*(2016).

[Comment 2] The implementation of “QF-MAP W” in Fig.10 (b) is very interesting. However, it implies that, given a number R of negative weights, those negative weights have specific indices. Therefore MATu has to be changed with MATu’, which has a different permutation of the 2^k rows. This is mentioned in the manuscript.

Q1. While one can assume MATu to be available (meaning that the order of the input is fixed), MATu’ must be obtained from MATu with $O(k^2)$ quantum gates or it will dominate the cost. This is not discussed in the paper and it is hard to believe that a generic permutation of the 2^k rows can be implemented with $O(k^2)$ quantum gates and classical processing. One cannot expect MATu’ to be provided since the weights change during training and therefore the permutation changes as well.

Q2. In addition, while commenting on Table 1, the authors write: “The speedup of QF-FB(C) over QF-FB(Q) is more than six orders of magnitude larger [...]. This verifies that QF-FB(C) can provide an efficient forward propagation procedure to support the lengthy training of QF-pNET.”. Does the author intend to use the quantum network for inference and perform the training on classical networks? If yes, the quantum advantage is confined to the inference (and not the learning).

[Action]

Thanks for the comment. Let us first clarify the question Q2 in the above comment, which will give us the insights for the question Q1.

For the question Q2, we have clarified on Page 2 of the revision that this work considers the acceleration of the inference phase (not the training phase) of a neural network. Specifically, the forward/backward propagation engine is developed and integrated into Pytorch to train QF-Net on classical computers, and then the trained network is deployed to quantum circuits for acceleration.

There are two main reasons that we consider inference on quantum computers: (1) the training process is a one-time effort while the inference process will be frequently applied, and (2) the acceleration of the inference phase will be the base for speeding up the training procedure.

Last Paragraph on Page 2, in Introduction Section:

New components in QF-Nets bring new computation paradigm. In order to support both the inference (a.k.a. testing) and training of QF-Nets, we further develop QF-FB, a forward/backward

propagation engine. In this work, we apply the classical computer to train the model, denoted as QF-FB(C). The quantum circuits with quantum advantage are designed for accelerating the inference phase, denoted as QF-F(Q). There are two main reasons that we do not consider the training procedure at this stage: first, the training process is a one-time effort while the inference process will be frequently applied; and second, the acceleration of the inference phase will be the base for speeding up the training procedure. The integration of the training procedure in QF-F(Q) will be our future work. In QF-FC, the QF-FB(C) component is integrated into PyTorch for training on classical computers; while QF-F(Q) is implemented based on Qiskit, which can be executed on either quantum processors or Qiskit Aer simulator.

Second, for question Q1, we have clarified in Method Section in the revision that the permutation of rows in MATu' is pre-determined, and therefore, the quantum states can be prepared efficiently using qRAM like MATu. As we have clarified for Q2, we consider the inference phase on quantum computers. Therefore, all weights are given and the permutation of rows in MATu' is determined.

The 4th Paragraph on Page 13, in Methods Section:

Kindly note that all weights are determined in the inference phase, indicating that MATu' can be obtained in the pre-processing phase. Therefore, we can efficiently conduct state-preparation (i.e., U operation) in U-LYR, similarly to MATu.

[Comment 3] A separate problem with U-LYR is that it can be used as only the first layer. One has then to continue with P-LYR layers. The quantum advantage is retained only if the number of neurons per layer is reduced exponentially from the first layer to the second (for example from 2^m inputs, to $\text{poly}(m)$ neurons in the first intermediate layer).

[Action]

Thanks for the comment. First, in the revision, we have clarified that the QF-hNet is designed for shallow neural networks (with one U-LYR and one P-LYR). For the deep neural network, due to the correlation issue (see Q1 in Comment 4), it will incur overhead on qubits. Such a problem fundamentally exists in designing quantum circuits for deep neural networks. A widely used and promising solution is to apply a hybrid quantum-classical computing scheme. In the revision, we have discussed how to take advantage of U-LYR in the design of deep neural networks on such a computing scheme. Details please see actions to question Q3 in Comment 4.

Second, in Section 5 of the Supplementary Information of the revision, we have added the detailed analysis of cost complexity of QF-hNet for different situations. The conclusion is that it is not required the number of neurons in the latter layer to be the logarithm of the number of inputs in the first layer, but it indeed has the constraint on the number of neurons on the second

layer to retain the quantum advantage. Specifically, for 2^m (or n) inputs, the number of neurons in the second (hidden) layer needs to be less than $2^{m/k}$ (or $n^{1/k}$) and $k > 2$. See the details as follows.

(SI) On Page 8, in Section 5 of the Supplementary Information:

Table 1. Complexity of QF-hNet with input size of $O(n)$ on classical and quantum computers.

Network		Perceptron	Shallow Network	
# Neurons in hidden or output layers		$O(1)$	$O(n^{\frac{1}{k}})$	$O(\log n)$
Classical	Time (T)	$O(1)$	$O(1)$	$O(1)$
	Space (S)	$O(n^2)$	$O(n^{\frac{1}{k}} \cdot n)$	$O(n^{\frac{k+1}{k}})$
	Cost (TS)	$O(n^2)$	$O(n^{\frac{k+1}{k}})$	$O(n \log n)$
Quantum (QF-hNet)	Time (T)	$O(\log^2 n)$	$O(n^{\frac{2}{k}} \log n)$	$O(\log^2 n)$
	Space (S)	$O(\log n)$	$O(n^{\frac{1}{k}} \log n)$	$O(\log^3 n)$
	Cost (TS)	$O(\log^3 n)$	$O(n^{\frac{3}{k}} \log^2 n)$	$O(\log^5 n)$
	Quantum Advantage	✓	✓ (when $k > 2$)	✓

Table 1 summarize the complexity for different situations, in terms of neuron numbers m and network depth d . From the table, it is clear to see that for Perceptron (i.e., $d=1$), QF-hNet can obtain quantum advantage, which is in accordance with the results in Figure 4 in the Manuscript. For the shallow network (i.e., $d=2$), the quantum advantage can be obtained when $m = n^{\frac{1}{k}}$ and $k > 2$, that is, the neurons in the hidden layer needs to be $O(n^{\frac{1}{k}})$, where n is the input size.

(for detailed analysis, please see the supplementary information)

[Comment 4] My concern relates to how the simulations were performed.

Q1: It seems to me that the circuit has been simulated piecewise (see sentence in page 4: “Because of the limitation of Qiskit Aer [...], we measure the results after each neuron.”), but that this procedure may neglect correlations building throughout the network among neurons even in the same layer.

My understanding is that simulations (for example of the first part of the circuit) were used to compute the probability of some intermediate qubit (let us say qubit k) to be in $|0\rangle$ (probability denoted q_k) or $|1\rangle$ (probability denoted $p_k=1-q_k$). Then, this classical information was used to initialize qubit k for the next piecewise simulation to: $\sqrt{q_k}|0\rangle + \sqrt{p_k}|1\rangle$. The other qubits involved in the first piecewise simulation (and that are now entangled with qubit k) were simply disregarded.

Q2: The issue with this approach is that correlations between intermediate qubits (even in the same layer) are not accounted in that kind of simulations. This is the same effect mentioned in my previous report (comment 5, part 2):

“(2) Referring to Fig.9(c), it is unclear to me why the outputs of the first hidden layer (qubits O2 and O3) can be used to update both neurons in the second hidden layer. Despite the measurement of O4 is delayed, it will still influence that of O5. [...] the statistics of outcomes O4 and O5 would not be as intended [...]”

The authors' response:

“For question (2), according to the above discussion that the probability of input qubits will not be changed throughout the neural computation process, making it possible to use the same inputs (i.e., the outputs of layer 1, O_2 and O_3) to calculate multiple neurons in a layer (i.e., layer 2). [...]”

focuses on the (unconditional) probability of the inputs. My point is about the correlation between outputs. To clarify, consider the simple circuit below with one input and two outputs:

qubit 0 = In0 : $|0\rangle$ --H--*--o--

| |

qubit 1 = Out0: $|0\rangle$ -----X--/--

|

qubit 2 = Out1: $|0\rangle$ -----X--

There are three gates: Hadamard, CNOT, and the X gate conditioned on the control qubit being in $|0\rangle$. The input state (after the Hadamard gate) corresponds to a random variable with $p_0=q_0=0.5$. Each output qubit has balanced probabilities, corresponding to random variables with: $p_1=q_1=p_2=q_2=0.5$.

But let us look at the joint probability of the outputs. If the outputs were independent (as assumed in QF-Net), the probability of measuring them both in state $|0\rangle$ should be q_1*q_2 . For the above circuit, it is 0. This is because using the same input qubits to update two output qubits in the next layer correlates them (in the example above and in the QF-Net, actually entangles them).

The binary classifier, discussed as an actual implementation on IBM chips, goes around this problem since has a single neuron update.

To summarize point 2, results from QF-FB(Q) may not correspond to a quantum implementation of QF-FB(C) since correlations between intermediate neurons are neglected. However, the piecewise simulations do correspond to a quantum version of QF-FB(C) since correlations are discarded.

Q3: From the considerations in point 2 above, it seems to me that (to exactly reproduce QF-FB(C) without intermediate measurements) one needs a number of input registers that grows as the product of the total number of qubits in subsequent layers (i.e. if we have N_0 inputs, N_1 neurons in the first layer, N_2 in the second layer, ..., then one may need $N_0*N_1*N_2*...$ input qubits). In general, one seems to need a number of qubits exponential in the network depth. Isn't it the case?

[Action]

Thanks so much for the comments and the detailed explanation. Let us divide the simulation related comments into three questions: Q1. whether the simulation neglects the correlations among neurons; Q2. whether the simulation and the quantum implementation are consistent; Q3. how to implement consistent quantum circuits to the simulation for deep neural networks. In the following, we will respond to the above three questions in order.

For the question Q1, we have clarified in the Results Section in the revision that QF-FB(C) carries out the simulation with no correlations among neurons, which is consistent with our quantum design. Specifically, the QF-hNet is designed for shallow neural networks. Due to the shallow network structure, we can eliminate the correlation in the circuit for QF-hNet with little overhead (details please see Q2). In addition, for deep neural networks, in Method Section of the revision, we have discussed how to apply U-LYR in deep neural networks on a hybrid quantum-classical computing platform (details please see Q3). As such, the correlation can also be eliminated by the measurement between two layers. Overall, we can make it consistent between the quantum system design and the simulation without the consideration of correlation.

The 2nd Paragraph on Page 4, in Results Section:

Because of the limitation of Qiskit Aer (whose backend is `ibmq_qasm_simulator`) used in QF-F(Q) that can maximally support 32 qubits, we measure the results after each neuron. This indicates that the outputs of a layer are independent to each other. However, if a quantum design is based on the same input qubits, it will involve correlation among neurons which may introduce error. In Method Section, we give two designs to eliminate such correlation to guarantee the consistency between the quantum implementation and the simulation.

For the question Q2. There are two types of potential correlations: (1) intra-layer correlation and (2) inter-layer correlation. But both correlations are eliminated in our design, and therefore, the quantum implementation and the simulation are consistent.

For the intra-layer correlation, it means that the input qubits in one layer have no correlation, but the outputs may have correlation due to they are based on the same inputs. The example in the comments of Q2 well demonstrates this case. In Section 5 in the supplementary information of the revision, we have clarified that even the output qubits of the last layer having correlation, it will not affect the classification results. This is because the classification is based on the probability of different qubits, rather than the joint states. Let's consider the example provides in the comment. If we measure qubits q_1q_2 and use the state of $|00\rangle$ for classification, it will be affected by the intra-layer correlation; however, in our design, the classification is based on the probability of qubit q_1 in the state of $|1\rangle$ and that of qubit q_2 in the state of $|1\rangle$. In the example, even $P\{q_1q_2=|00\rangle\}=0$, but $P\{q_1=|0\rangle\}$ is still 0.5, and $P\{q_2=|0\rangle\}$ is also 0.5. Therefore, we do not

need to require the final outputs have no correlation. But we indeed require the outputs of hidden layers to be independent because they will be used for the computation in the next layer.

For the inter-layer correlation, it means that if the outputs of the previous layer have correlation, the computation in the next layer will be affected. For QF-hNet, it is designed for shallow network, such correlation does not exist, because in the first layer, U-LYR, each output qubit is obtained by an independent set of input qubits, and therefore, they have no correlation. When extending QF-hNet to multiple-layers or for QF-pNet, we can eliminate the inter-layer correlation by keeping the output qubits of each layer (except the last layer) to be independent. Details please see the action to the question Q3.

(SI) On Page 7, in Section 5 of the Supplementary Information:

Let the number of inputs be $O(n)$, let the number of neurons in hidden layers and output layers be $O(m)$, and let the number of layers be d . Then, the space (i.e., qubits) complexity of QF-hNet will be $O(m^{d-1} \times m \log n) = O(m^d \log n)$, where $O(m \log n)$ is the qubits required in the first layer, where inputs are encoded into $\log n$ qubits and m outputs need m sets of inputs to conduct U-LYR. Then, for the latter layers, to guarantee the independence among outputs, the d -layer network needs to be flattened. We have $d-1$ since we relax the independent requirement for the last layer to reduce the number of qubits. Such relaxation will not affect the function correctness since the classification is based on the probability of each output qubit rather than a joint state.

For Q3, first, we have clarified that QF-pNet and QF-hNet are designed for shallow neural networks with one hidden layer. It is possible to extend them to deep neural networks. But as said in the comments, in order to eliminate the inter-layer correlation, the number of input registers grows as the product of the total number of neurons (i.e., qubits) in subsequent layers. This has been discussed in Section 5 in the supplementary information of the revision. A further question is whether we can reduce the quantum circuit complexity by reusing the same input qubits, which will bring the correlation in the latter layers. This will be our future work since it needs to further involve the fundamental changes in the training procedure to support the existence of correlation among neurons in the inference phase.

A better way to extend to deep neural networks has been discussed in Method Section of the revision. We introduce how to fully utilize U-LYR in a deep neural network to potentially achieve the quantum advantage. Specifically, like many existing works, the hybrid quantum-classical computing scheme can solve the correlation issue. In the revision, we discuss how to integrate U-LYR to deep neural networks and quantitatively analyze the cost complexity. Benefiting from the advantage of U-LYR, we can reduce the complexity from $O(d \cdot n^2)$ on classical computing to $O(d \cdot n^{3/2})$ on the hybrid quantum-classical computing scheme, given a network with d layers, $O(n)$ inputs and $O(n)$ neurons in each layer. In addition, the complexity is bounded by quantum state-preparation. With the continuous improvement of the quantum state-preparation protocol,

QuantumFlow may ultimately achieve the complexity of $O(d \cdot n \cdot \log^3 n)$. The analysis shows the potential quantum advantages to be achieved for deep neural networks.

(SI) On Page 8, in Section 5 of the Supplementary Information:

Finally, QF-hNet is designed for shallow neural networks, but it can be extended to support multiple layers. But it has additional costs. Specifically, to guarantee that the outputs of each layer except the last layer have no correlation, we need to flatten the neural network to make sure each output is computed based on an independent input. As a result, the input qubits will exponentially increase along with the network depth; this can be seen from the space complexity $O(m^d \log n)$ at the beginning of this section, where n is the inputs and m neurons in hidden and output layer, and d is the network depth. Kindly note that this is not the limitation of the proposed approach, but generally exists in implementing deep neural networks. A promising and commonly applied solution to this problem is to use the hybrid quantum-classical computing scheme. We have discussed how to apply U-LYR to such a computing scheme in the last subsection of the Method Section.

Last Paragraph on Left-Hand Page 14, in Method Section:

Construction. The extended design using U-LYR on hybrid quantum-classical computing iteratively conducts 4 steps for layers in sequence: (1) encoding the input neurons to input qubits; (2) do neural computation with pre-trained weights; (3) measurement the output qubits to generate output neurons; (4) translate the output neurons according to the pre-determined weights in the next layer. In the above procedure, steps (1) and (3) are the interface between classical and quantum computing, while step (2) is on quantum computing, and step (4) is conducted classical computing. Step (1) is the well-known quantum state-preparation. It can be implemented in different ways, such as qRAM based approach^{42,43} and computing based approach⁴⁹⁻⁵¹. In step (2), we apply U-LYR for the computation. Step (3) will measure the output qubits and send the results to step (4) for the data preparation according to the weight in the next layer and the results of Algorithm 4. These 4 steps will be iteratively conducted d times to reach the output neurons for a d -layer network.

Table 4. Complexity of each step in hybrid quantum-classical computing for deep neural network with U-LYR.

Complexity	State-Preparation	Computation	Measurement
Depth (T)	$O(d \cdot \sqrt{n})$	$O(d \cdot \log^2 n)$	$O(d)$
Qubits (S)	$O(n)$	$O(n \cdot \log n)$	$O(n \cdot \log n)$
Cost (TS)	$O(d \cdot n^{\frac{3}{2}})$	$O(d \cdot n \cdot \log^3 n)$	$O(d \cdot n \cdot \log n)$
Total (TS)	$O(d \cdot n^{\frac{3}{2}})$		

Cost complexity. We adopt the widely used time-space product complexity^{2,52-55} as the cost complexity. In quantum computing, time and space correspond to the circuit depth and the number of qubits. For a fair comparison, in classical computing, time and space correspond to the computing latency and computing storage. We consider computing storage instead of total storage because there is no need to load all data to on-chip storage during computation.

In the complexity analysis, we focus on the cost of steps (1)-(3). Note that the weights are given at the inference phase. This indicates a fixed mapping in step (4) from outputs in i^{th} layer to inputs $(i+1)^{\text{th}}$, which can be implemented by classical hardware at a constant cost. Let d be the number of layers and let the number of input and output neurons for each layer is $O(n)$. Table 4 gives the detailed complexity analysis for each step. In step (1), we apply the computation-based approach⁵¹ for state-preparation, where the time complexity (i.e., circuit depth) for each layer is $O(\sqrt{n})$ and it is $O(d \cdot \sqrt{n})$ for d layers. Its space complexity (i.e., qubits) is $O(\log g)$, where g is related to the precision of amplitudes. In analyzing the quantum advantage, we use the same finite precision with classical computer; therefore, g can be regarded as a constant, leading the space complexity to be $O(n)$. In step (2), we apply the proposed U-LYR for the computation. Its time complexity is $O(\log^2 n)$ for each layer, and $O(d \cdot \log^2 n)$ for d layers. According to Formula 2, the space complexity is $O(\log n)$ for 1 output, and $O(n \cdot \log n)$, for $O(n)$ outputs. Finally, the measurement takes $O(1)$ time complexity for each layer and $O(d)$ for d layers. Each output corresponds to a qubit, and therefore the space complexity is $O(n)$. Overall, we can obtain the cost time-space product complexity for the whole system to be $O(d \cdot n^{\frac{3}{2}} \log g)$.

It is obvious that the state-preparation dominates the time complexity, but it still demonstrates the potential advantage can be achieved for executing neural networks on quantum computing against classical one. For the classical computing, the time-space product complexity is $O(d \cdot n^2)$ (details see *Supplementary Information*). Considering that the data precision in classical computing is finite, it indicates that g is constant. Therefore, the complexity of quantum computing is $O(d \cdot n^{\frac{3}{2}})$, achieving speedup over classical computing. With the improvement of the state-preparation protocols, the overall complexity can be further reduced.

New references:

42. Allcock, J., Hsieh, C.-Y., Kerenidis, I. & Zhang, S. Quantum algorithms for feedforward neural networks. *ACM Transactions on Quantum Comput.*1, 1–24 (2020).
43. Kerenidis, I. & Prakash, A. Quantum recommendation systems. *arXiv preprint arXiv:1603.08675*(2016).
49. Sanders, Y. R., Low, G. H., Scherer, A. & Berry, D. W. Black-box quantum state preparation without arithmetic. *Phys. review letters*122, 020502 (2019).
50. Grover, L. K. Synthesis of quantum superpositions by quantum computation. *Phys. review letters* 85, 1334 (2000)
51. Bausch, J. Fast Black-Box Quantum State Preparation. *arXiv preprint arXiv:2009.10709*(2020).

[Comment 5] The authors should clarify how many qubits are required in their construction. For example, a generic formula should be provided when discussing special cases in page 4 (left column).

For the batch normalization, does QF_FC(Q) requires 4 extra qubits per neuron? Two for “indiv_adj” and two for “batch_adj”?

In addition, Table 1 should not report the time elapsed on CPU to simulate QF-FB(Q), but the expected time to run it with actual quantum devices. For example, the authors can consider a simplified quantum device with all-to-all connectivity and a single duration for every gate.

[Action]

Thanks for the comment. In Results Section of the revision, we have first given the generic formula of the number of qubits in the construction of three types of layers: P-LYR, U-LYR, and N-LYR. Specifically, for N-LYR, there are three cases to use 0, 2, and 4 extra qubits per neuron. Second, we have removed the time elapsed on CPU for the simulation, since the expected time (circuit depth) has been given in Table 2.

Last Paragraph on Left-Hand Page 4, in Results Section:

In Table 1, columns “Layer Structure” gives the structure of a layer of neuron computation, based on which we can calculate the number of qubits used for each layer. Let $n_1 \rightarrow n_2$ be a layer with n_1 input neurons and n_2 output neurons. We first assume $n_2 = 1$, and we have the following formula for calculating the qubits used in three types of layer: P-LYR, U-LYR, and N-LYR.

Table 1. Inference accuracy and efficiency comparison between QF-FB(C) and QF-F(Q) on both QF-pNet and QF-hNet using MNIST dataset to show the consistency of the inference of QF-Nets on classical computers and quantum computers.

dataset	QF-pNet				QF-hNet									
	Layer Structure		qubits		Accuracy		Layer Structure		qubits		Accuracy			
	L1	L2	L1	L2	QF-FB(C)	QF-F(Q)	Diff.	L1	L2	L1	L2	QF-FB(C)	QF-F(Q)	Diff.
{3,6}	16 → 4	4 → 2	28 × 4	12 × 2	97.10%	95.53%	-1.57%	16 → 4	4 → 2	7 × 4	8 × 2	98.27%	97.46%	-0.81%
{3,8}	16 → 4	4 → 2	28 × 4	12 × 2	86.84%	83.59%	-3.25%	16 → 4	4 → 2	7 × 4	8 × 2	87.40%	88.06%	+0.54%
{1,3,6}	16 → 8	8 → 3	28 × 8	18 × 3	87.91%	81.99%	-5.92%	16 → 8	4 → 2	7 × 8	14 × 3	88.53%	88.14%	-0.39%

$$Q_P(n_1) = n_1 + \log n_1 + (\log n_1 - 1), (1)$$

where it has n_1 input qubits, $\log n_1$ encoding qubits, and $\log n_1 - 1$ ancillary qubits.

$$Q_U(n_1) = \log n_1, (2)$$

where all n_1 inputs are encoded to $\log n_1$ qubits.

$$Q_N = \begin{cases} 0 & \text{if } \theta = 0 \text{ and } \gamma = 0 \\ 2 & \text{elseif } \theta = 0 \text{ or } \gamma = 0 \text{ or } t = 0, (3) \\ 4 & \text{otherwise} \end{cases}$$

where $\theta = 0$ ($\gamma = 0$) indicates that batch_adj (indiv_adj) is not applied, and t=0 indicates that batch_adj and indiv_adj can be merged into one circuit (see Figure 10(f)). Finally, we can get the number of qubits used in a layer with n_1 inputs and n_2 outputs.

$$Q_{try}(n_1, n_2) = \begin{cases} (Q_P(n_1) + Q_N + 1) \times n_2 & P-LYR \\ (Q_U(n_1) + Q_N + 1) \times n_2 & Q-LYR \end{cases}, (4)$$

where “1” represents the output qubit, and we multiply n_2 to make outputs are independent to each other...

Based on these understandings, we obtain the number of qubits used in each layer as reported in Table 1 under columns “qubits”. For example, for L1 under QF-pNet in {3,6} with 16 inputs and 4 outputs, it applies P-LYR and N-LYR with $t \neq 0$. According to Formula (4), the qubits are $Q_{lry}(16,4) = ((16 + 4 + 3) + 4 + 1) \times 4 = 28 \times 4$. From the results, it is clear to see that the U-LYR, which applied in L1 of QF-hNet, can significantly reduce the number of required qubits. For dataset {3,6}, it achieves 4x reduction, compared with P-LYR in L1 of QF-pNet.

[Comment 6] The first time “batch normalization” is mentioned, a definition should be provided. Possibly together an intuitive explanation of why it is important for classical NN.

[Action]

Thanks for the comment. In the Introduction Section of the revision, we have given the definition of batch normalization and explain why it is important for neural networks. Specifically, batch normalization is a normalization step to fix the distribution properties (e.g., means) of layer inputs in a training mini-batch. It is important for neural networks since it can eliminate the internal covariate shift to achieve higher accuracy and accelerate the training procedure.

The 4th Paragraph on Right-Hand Page 2, in Introduction Section:

In addition to neural computation, batch normalization is commonly employed in neural networks. It contains a normalization step to fix the distribution properties (e.g., means) of layer inputs in a training mini-batch, and thus it can eliminate the internal covariate shift to achieve higher accuracy and accelerate the training procedure. In QF-Nets, we devise a quantum-friendly batch normalization N-LYR, which can be plugged into both QF-pNet and QF-hNet. It includes additional parameters to normalize the output of a neuron, which are tuned during the training phase.

[Comment 7] Notation and terminology:

- Usually, quantum bits are spelled “qubits” and not “qbits”.
- The common term is “physical qubits” and not “physic qbits”.
- For the introduction, other versions of quantum neural networks exist. For example:
 - Wan, K. H., Dahlsten, O., Kristjánsson, H., Gardner, R., & Kim, M. S. (2017). Quantum generalisation of feedforward neural networks. *Npj Quantum Information*, 3, 36.
 - Cao, Y., Guerreschi, G. G., & Aspuru-Guzik, A. (2017). Quantum Neuron: an elementary building block for machine learning on quantum computers. *ArXiv:1711.11240*.

[Action]

Thanks for pointing these out. In the revision, we have fixed typos and added the references.

Comments by the 2nd Reviewer:

The authors have thoroughly responded to my comments and clarified issues found in the paper's first draft. I think the authors improved the paper enough, and I am content to recommend it as ready for publication.

[Response]

Thanks for your recommendation. We appreciate your valuable time and helpful comments to greatly improve the quality of this paper.

Comments by the 3rd Reviewer:

Authors have properly addressed my questions in the revised version and I am happy to recommend the paper for publication in its current version.

[Response]

Thanks for your recommendation. We appreciate your valuable time and helpful comments to greatly improve the quality of this paper.

Reviewers' Comments:

Reviewer #1:

Remarks to the Author:

The authors have responded to all my points and the manuscript has been updated accordingly. I can recommend the current version for publication.